EUROPEAN ORGANISATION FOR NUCLEAR RESEARCH (CERN)

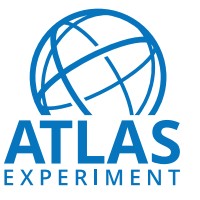

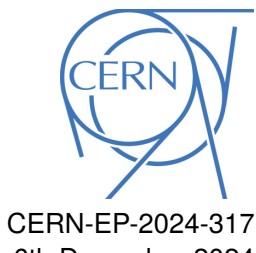

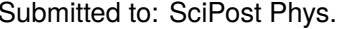

Submitted to: SciPost Phys.

CERN-EP-2024-317
6th December 2024

# Precision calibration of calorimeter signals in the ATLAS experiment using an uncertainty-aware neural network

The ATLAS Collaboration[1]

The ATLAS experiment at the Large Hadron Collider explores the use of modern neural networks for a multi-dimensional calibration of its calorimeter signal defined by clusters of topologically connected cells (topo-clusters). The Bayesian neural network (BNN) approach not only yields a continuous and smooth calibration function that improves performance relative to the standard calibration but also provides uncertainties on the calibrated energies for each topo-cluster. The results obtained by using a trained BNN are compared to the standard local hadronic calibration and to a calibration provided by training a deep neural network. The uncertainties predicted by the BNN are interpreted in the context of a fractional contribution to the systematic uncertainties of the trained calibration. They are also compared to uncertainty predictions obtained from an alternative estimator employing repulsive ensembles.

---

[1] The full author list can be found at:
https://atlas.web.cern.ch/Atlas/PUBNOTES/ATL-PHYS-PUB-2024-XXX/authorlist.pdf

# 1 Introduction

The principal signals of the non-compensating calorimeters employed by the ATLAS experiment [1] at the Large Hadron Collider (LHC) are clusters of topologically connected cell signals (topo-clusters) that reconstruct locally deposited energy into a three-dimensional signal object [2]. The calorimeter signals are calibrated to correctly measure the energy deposited by electromagnetic showers, thus providing no compensation for the fraction of deposited energy not contributing to the signal in the complex development of hadronic showers. This leads to on average smaller observed signals for hadrons depositing the same energy as electrons, positrons, or photons.

In ATLAS, the standard approach to compensate for the hadronic signal deficiencies at the level of individual topo-clusters is to apply a *local hadronic calibration* as part of a full calibration chain that also includes corrections for other effects. This chain is implemented as a four-step sequence of (1) a cluster-by-cluster signal classification determining the likelihood of the topo-cluster to be of electromagnetic origin, (2) a hadronic calibration employing a local cell-signal weighting technique giving the chain its name (LCW), (3) an out-of-cluster correction, and (4) a correction for energy losses in inactive material in the proximity of the topo-cluster, as detailed in Ref. [2]. All classification probabilities and calibration and correction scale factors used in this sequence are retrieved from multi-dimensional lookup tables that are calculated from single particle response simulations. These tables provide averages in bins of observables sensitive to the nature of the energy deposit and to effects on the signal introduced by the detector geometry, and its signal formation and extraction characteristics at the topo-cluster location. In addition to bin edge effects potentially introducing discontinuities of scale factors, the correlations between the observables are reduced to average relations in this approach. This is expected to introduce limitations to the precision of the LCW calibration, as the specific relation between the observables used as inputs is lost for any given topo-cluster.

Modern machine-learning (ML) methods [3] can be used to learn multi-dimensional, continuous calibration functions without step-like transitions. These functions preserve, and even use, the underlying correlations between observables, thus promising a more accurate topo-cluster calibration at higher precision. A set of topo-cluster observables commonly referred to as *features* provides the inputs to the corresponding training, and to the inference of the trained model. This set includes physics-motivated data representations encompassing measures of the topo-cluster signal characteristics, the underlying nature of the energy deposit, and the signal environment at the LHC.

A one-step training procedure omitting the dedicated topo-cluster classification step was designed in a previous approach to determine such a function. The results obtained by training a deep neural network (DNN) to learn the response of individual topo-clusters from which the signal calibration is derived are presented in Ref. [4]. The performance of the DNN-trained calibration with respect to signal linearity and resolution was found to yield significant improvements when compared with the standard hadronic calibration provided in the LCW sequence. Other machine-learning-based approaches to calibrate topo-clusters use cells in topo-clusters in deep learning networks employing images, graphs, and point clouds. Those were successfully trained to calibrate single particles in full Monte Carlo (MC) simulations of the ATLAS detector [5, 6].

In this paper, results from the application of a Bayesian neural network (BNN) [7–13] for the calibration of the topo-clusters obtained with full MC simulations of multijet final states of the proton–proton (*pp*) collisions at the LHC Run 2 (2015-2018) centre-of-mass energy of $\sqrt{s} = 13$ TeV are presented. These simulations encompass physics modelling of the particle-level final state and a detailed detector simulation. The detector simulation models the signal contributions from pile-up introduced by the high inelastic *pp* cross-section, in combination with the high proton beam intensities observed in Run 2.

The BNN is trained to not only predict the response and thus the calibration function cluster-by-cluster, but to also learn a statistical and a systematic uncertainty component associated with this prediction. The statistical uncertainty captures the dependence of the prediction on the size of the dataset used for training the BNN, while the systematic uncertainty component provides a measure for the accuracy of the prediction that cannot be improved by a larger training dataset or more training cycles (*epochs*). These uncertainties can provide important contributions to the overall systematic uncertainty associated with the application of the trained BNN model for the local hadronic calibration of individual topo-clusters. The uncertainties allow, for example, additional selections of topo-clusters based on the accuracy of their calibrated signal when reconstructing physics objects like individual particles and jets from them.

The individual uncertainty predictions are available for all topo-clusters with a calibration learned by the BNN, including those that are not part of the signal of a calibrated physics object like jets or particles. Similar to their use for reconstructing jets and particles, topo-cluster selections and uncertainty propagation are possible in the context of the reconstruction of more complex and softer energy flows that are potentially needed for the measurements of the missing transverse momentum, the soft hadronic recoil in vector boson production, and other hadronic event shapes.

To increase confidence in the uncertainty predictions from the BNN, they are compared to an alternative uncertainty estimate determined with *repulsive ensembles* (RE) [14, 15]. Observed large predicted uncertainties for certain topo-clusters are interpreted in the context of the signal characteristics associated with complex transition regions in the ATLAS calorimeter system.

The structure of this paper is as follows. In Section 2, the ATLAS detector and the used datasets are presented. Introduction to the neural-network-based calibration, the features it uses, and the BNN and RE network architecture and configurations are given in Section 3. In Section 4, the performance of the calibration is compared to the LCW and the DNN-derived calibrations. Section 5 discusses the validity of the uncertainties predicted by the BNN in a comparison with the corresponding RE predictions and in the context of detector signal features. The paper concludes with Section 6.

## 2 Experimental setup and simulations

The results presented here are solely obtained with MC simulations that model the setup of the ATLAS experiment and the signal formation and extraction for the various sub-detectors in great detail. The emphasis is on the central region of the calorimeter system. It is the most relevant detector component for the studies presented in this paper, as the topo-clusters used to learn the response and validate the obtained calibration are collected from calorimeter jets in this region. The only other detector systems contributing here are the inner tracking detector for the reconstruction of the collision vertices and the dedicated forward detectors comprising the luminosity measurement system [16] used for the reconstruction of the number of pile-up collisions.

## 2.1 The ATLAS detector

The ATLAS detector at the LHC covers nearly the entire solid angle around the *pp* collision vertex.[2] It consists of an inner tracking detector system (ID) surrounded by a thin superconducting solenoid, electromagnetic and hadronic calorimeters, and a muon spectrometer (MS) incorporating three large superconducting air-core toroidal magnets.

The calorimeter system provides near full absorption coverage in the pseudorapidity range $|\eta_{\text{det}}| < 4.9$. Electromagnetic calorimetry is provided up to $|\eta_{\text{det}}| < 3.2$ by highly granular barrel (EMB) and endcap (EMEC) lead/liquid-argon (LAr) calorimeters. An additional thin LAr presampler covers $|\eta_{\text{det}}| < 1.8$ to correct for energy losses in material in front of the calorimeters.

Hadron calorimetry up to $|\eta_{\text{det}}| < 1.7$ is provided by the steel/scintillator-tile (Tile) calorimeter with three barrel structures (one barrel Tile and two extended barrel Tile calorimeters). The two hadronic endcap (HEC) copper/LAr calorimeters provide coverage for $1.5 < |\eta_{\text{det}}| < 3.2$. Scintillating counters are installed between the barrel and the extended Tile calorimeters to measure energy losses in this transition region within $0.8 < |\eta_{\text{det}}| < 1.6$ (Tile gap scintillators).

The two forward calorimeters (FCal) cover $3.1 < |\eta_{\text{det}}| < 4.9$. Each consists of a copper/LAr module followed by two tungsten/LAr modules for a total of three longitudinal segments, a setup optimised for electromagnetic and hadronic energy measurements.

The ATLAS calorimeters feature highly granular readout, with a presampler followed by six longitudinal readout segments in the barrel region $|\eta_{\text{det}}| < 1.5$, a presampler up to $|\eta_{\text{det}}| = 1.8$ and six to seven longitudinal segments in the endcap region, and three longitudinal segments in the forward region. The calorimeter has a depth of at least $10\,\lambda$ everywhere, with $\lambda$ being the hadronic absorption length. Overall, it has nearly $200\,000$ independent readout cells, with the highest lateral granularities found in the EMB and the EMEC. The software suite described in Ref. [17] is used for the reconstruction and analysis of the MC simulations. It is in large parts identical to the one used for experimental data.

## 2.2 Monte Carlo simulations

The fully simulated jet samples were produced according to the description given in Ref. [18]. The hard-scatter *pp* interactions were generated at $\sqrt{s} = 13\,\text{TeV}$ using PYTHIA 8.230 [19] with the A14 [20] set of tuned parton shower and underlying event parameters and the NNPDF2.3LO [21] set of parton distribution functions (PDF).

To simulate the detector signals, the stable particles[3] from the generated final state were tracked through the ATLAS detector using the GEANT4 [22] toolkit. The energy deposited in the calorimeters was collected in the detector simulation and is available for each calorimeter cell and in regions of inactive material. The signal formation in the various detector systems is modelled following the respective signal extractions

---

[2] ATLAS uses a right-handed coordinate system with its origin at the nominal interaction point (IP) in the centre of the detector and the $z$-axis along the beam pipe. The $x$-axis points from the IP to the centre of the LHC ring, and the $y$-axis points upwards. Polar coordinates $(r, \phi)$ are used in the transverse plane, $\phi$ being the azimuthal angle around the $z$-axis. The pseudorapidity is defined in terms of the polar angle $\theta$ as $\eta = -\ln \tan(\theta/2)$ and is equal to the rapidity $y = \frac{1}{2} \ln \left( \frac{E + p_z}{E - p_z} \right)$ in the relativistic limit. The pseudorapidity $\eta_{\text{det}}$ in the detector frame of reference follows the definition of $\eta$ with reference to the nominal collision vertex at $(x, y, z) = (0, 0, 0)$. Angular distance is measured in units of $\Delta R \equiv \sqrt{(\Delta y)^2 + (\Delta \phi)^2}$.

[3] Here particles are considered stable if their proper lifetime $\tau$ is $\tau > 10\,\text{ps}$.

and digitisations in the experiment [23]. The effect of multiple *pp* interactions in the same (in-time pile-up) and neighbouring bunch crossings (out-of-time pile-up) was modelled by overlaying the simulated signals of the hard-scattering event with signals from inelastic *minimum bias pp* collisions generated with Pythia 8.186 [24] using the NNPDF2.3lo set of PDFs and the A3 set of tuned parameters [25]. The number of overlaid pile-up interactions is sampled from the distribution measured in the data during LHC Run 2.

## 2.3 Calorimeter signal formation and features

The formation of topo-clusters principally collects calorimeter cells starting from seed cells with highly significant signals and growing by following cell signal significance patterns within and across calorimeter modules in three dimensions. The initial clustering is followed by an algorithm splitting large topo-clusters between local signal maxima, again in three dimensions. The full procedure and the reconstructed observables associated with the final cluster are detailed in Ref. [2]. This reference also introduces the LCW calibration that uses signal characteristics, shapes and locations associated with the topo-clusters.

### 2.3.1 Topo-cluster kinematics

The reconstructed topo-clusters have kinematics that are represented at two scales, the basic electromagnetic (EM) scale and the calibrated LCW scale. At both scales, topo-clusters are considered to be massless *pseudo-particles*. The four-momentum representation $P_{clus}^{EM}$ of a topo-cluster at the EM scale is given by

$$P_{clus}^{EM} = E_{clus}^{EM} \cdot \left(1, \cos\phi_{clus}^{EM}/\cosh y_{clus}^{EM}, \sin\phi_{clus}^{EM}/\cosh y_{clus}^{EM}, \tanh y_{clus}^{EM}\right) , \quad (1)$$

where $E_{clus}^{EM}$, $y_{clus}^{EM}$ and $\phi_{clus}^{EM}$ are the cluster energy, rapidity and azimuth reconstructed at the EM scale, respectively.

The ML calibrations derived in this study are specific for each topo-cluster and focus on the energy deposit it represents at its location in the calorimeter. There is no attempt to include predictions for the correction of energy losses outside of the topo-cluster. Such corrections are included in the full LCW chain, though, see Appendix A for more details.

The *hadronically calibrated* cluster energy $E_{clus}^{had}$ is available from the LCW sequence, before the corrections for out-of-cluster and inactive material energy losses are applied. It serves as a reference for performance comparisons with the ML calibrations. The expectation is that it represents the energy $E_{clus}^{dep}$ deposited in the calorimeter cells collected into the topo-cluster,

$$E_{clus}^{EM} \xrightarrow{\substack{\text{hadronic} \\ \text{calibration}}} \underbrace{E_{clus}^{had} = E_{clus}^{dep}}_{\text{expectation/goal}} .$$

Consequently, $E_{clus}^{had}$ is referred to as the signal at the local hadronic LCW scale, as it is reconstructed in the context of this calibration but omits further corrections not related to the energy deposit within the volume of the topo-cluster. Due to the cell signal weighting applied in the hadronic calibration step of the LCW procedure, typically $E_{clus}^{had} > E_{clus}^{EM}$, $y_{clus}^{EM} \neq y_{clus}^{had}$ and $\phi_{clus}^{EM} \neq \phi_{clus}^{had}$ are expected for all topo-clusters not classified as purely electromagnetic (EM) based on an EM-likelihood $\mathcal{P}_{clus}^{EM} < 1$ measure provided in the classification step of this procedure. The machine-learning-based calibration applies a scale factor for $E_{clus}^{EM}$ to represent $E_{clus}^{dep}$, and as such does not change the topo-cluster direction reconstructed at EM scale.

### 2.3.2  Simulation data content

The detector simulation provides the relevant *truth information* that assesses the energy $E_{\text{clus}}^{\text{dep}}$ deposited inside the topo-cluster. This is the sum of all energies $E_{\text{cell}}^{\text{dep}}$ deposited in the $N_{\text{clus}}^{\text{cell}}$ clustered cells during the shower simulation of *primary particles* emerging directly from the generated hard-scatter interaction,

$$E_{\text{clus}}^{\text{dep}} = \sum_{i=1}^{N_{\text{clus}}^{\text{cell}}} w_{\text{cell},i}^{\text{geo}} E_{\text{cell},i}^{\text{dep}} \, .$$

The geometrical weight $0 < w_{\text{cell},i}^{\text{geo}} \leq 1$ reflects the fractional contribution of $E_{\text{cell},i}^{\text{dep}}$ to the topo-cluster if the cell $i$ is shared with another cluster, with $w_{\text{cell},i}^{\text{geo}} = 1$ for cells $i$ exclusively collected into one topo-cluster.

Within this approach $E_{\text{clus}}^{\text{dep}}$ is the simulated *detector truth* that is generated by the energy flow carried by primary particles into the calorimeter and collected within the constraints introduced by the topo-cluster algorithm. Thus, cells that have energy deposits from these particles, but are not clustered, do not contribute to the detector truth of a given final state. Also not included in $E_{\text{clus}}^{\text{dep}}$ are energies deposited by particles from in-time pile-up collisions, or any energies deposited by particle generated in nearby bunch crossings contributing to out-of-time pile-up. Any target for a machine-learned calibration, or any other, which is constructed from $E_{\text{clus}}^{\text{dep}}$ thus reflects only the energy deposit generated by the *pp* interaction of interest, and not from pile-up interactions. This principally allows determining calibration functions that locally, at the level of each topo-cluster, mitigate the effect of pile-up on the topo-cluster signal $E_{\text{clus}}^{\text{EM}}$ with these simulations.

Each $E_{\text{cell}}^{\text{dep}}$, and thus by extension the $E_{\text{clus}}^{\text{dep}}$ it contributes to, comprises visible and invisible energy losses within the particle shower. The visible energy deposits typically consist of losses through ionisation and other interactions that potentially contribute to the cell signal $E_{\text{cell}}$, and thus the topo-cluster signal $E_{\text{clus}}^{\text{EM}}$. Invisible losses do not have a signal contribution. They include the rest energy of stopped particles, particles absorbed or decaying without visible energy release, energy invested in (slow) nuclear processes following an inelastic interaction between incoming shower particles and the matter they traverse. In addition, all energy invested in invisible particles like neutrinos, which can escape the detector without further trace, is included.[4] The invisible losses mostly appear in the development of hadronic showers, and thus are the reason for the non-compensating signal character of the ATLAS calorimeter.

The composition of $E_{\text{cell}}^{\text{dep}}$, and therefore also $E_{\text{clus}}^{\text{dep}}$, is preserved in fractional contributions depending on the nature of the primary particles that cause the energy release into a cell volume in their interactions with the matter of the calorimeter. In particular, $E_{\text{clus}}^{\text{dep}}$ can be characterised by the fraction of energy $f_{\text{clus}}^{\text{dep,em}}$ deposited by showers from primary electrons/positrons ($e^{\pm}$) and photons ($\gamma$). Within the context of the studies presented in this paper, the following nomenclature is used to categorise topo-clusters according to $f_{\text{clus}}^{\text{dep,em}}$:

**Electromagnetic topo-clusters**
  have at least 90% of $E_{\text{clus}}^{\text{dep}}$ coming from primary $e^{\pm}$ or $\gamma$, $f_{\text{clus}}^{\text{dep,em}} > 0.9$.

---

[4] Within the ATLAS detector simulation, all energy released in the development of particle showers, and not invested into the production of secondary particles that can be further tracked through the detector setup, is collected at the location where it is produced. For the energy deposits discussed here, this is the calorimeter cell volume in which the interaction happened. The concept of stopped particles is implemented in the interaction modelling within GEANT4 for particles with energies or ranges below the corresponding configured thresholds for secondary particle tracking and thus does not necessarily reflect physical behaviour at the interaction level.

**Hadronic topo-clusters**

have less than 10% of $E_{\text{clus}}^{\text{dep}}$ deposited by primary $e^{\pm}$ and $\gamma$, $f_{\text{clus}}^{\text{dep,em}} < 0.1$.

**Composite topo-clusters**

have varying fractional contributions to $E_{\text{clus}}^{\text{dep}}$ from various primary particles such that $0.1 \leq f_{\text{clus}}^{\text{dep,em}} \leq 0.9$.

This categorisation is not directly used in the training of the machine-learning-based calibrations, as it is not available for topo-clusters in the experiment. It is useful when evaluating the sensitivity of observables to the cluster signal origin. More sensitive observables are expected to better reflect the relation between $E_{\text{clus}}^{\text{dep}}$ and the signal $E_{\text{clus}}^{\text{EM}}$ it generated, which, among other considerations, suggests to include them as inputs for the calibration training, as further discussed in Section 3.1.2.

## 2.4 Event observables

The MC simulation samples include pile-up at levels observed in LHC Run 2, which affects the topo-cluster signal $E_{\text{clus}}^{\text{EM}}$ and thus the calibration. Measurable indicators of the pile-up activity in a given event are therefore considered for inclusion into the training of the cluster calibration network. The two relevant observables in data and MC simulations are the number of reconstructed primary vertices $N_{\text{PV}}$ [26] and the (measured) actual number of pile-up interactions in the event $\mu$. Generally both $N_{\text{PV}}$ and $\mu$ are correlated, with $N_{\text{PV}}$ providing a direct measure for in-time pile-up activity of each bunch crossing while $\mu$ is also indicative of the out-of-time pile-up activity.[5]

# 3 Datasets and network setup

## 3.1 Dataset

The dataset consists of selected topo-clusters reconstructed in MC simulations of full $pp$ collision events at $\sqrt{s} = 13$ TeV with multijet final states. The events are generated and simulated as described in Section 2.2.

The topo-clusters are collected from fully reconstructed calorimeter jets in the central detector region of ATLAS, as provided by the MC simulations. Each jet establishes a calorimeter volume containing its topo-cluster constituents. With this, they not only provide a reference phase space for a group of clusters for future comparisons with experimental data, but also an environment characterised by varying topo-cluster densities covering many spatial cluster distributions also expected in data.

---

[5] For MC simulations, $\mu$ is sampled from the distribution of $\langle\mu\rangle$, which is the number of interactions measured over a luminosity block at the LHC (duration about 30 s to 60 s) divided by the number of proton bunch crossings in this interval (average over all bunches). Due to this time-integrated construction, $\langle\mu\rangle$ provides an average measure for the out-of-time pile-up activity. For data, $\mu$ is determined from the number of pile-up interactions averaged over all bunches at a fixed position in the bunch train, again in a luminosity block [16].

### 3.1.1 Final-state objects and phase space

Calorimeter jets are formed from topo-clusters with signals at the EM scale using the anti-$k_t$ algorithm [27] in FASTJET [28] with a radius parameter $R = 0.4$. They are calibrated using the procedures described in Ref. [29], including pile-up corrections, the MC-simulations-derived jet energy scale (JES) calibration, and direction corrections. The considered phase space for these jets is motivated by typical analysis selections and to restrict the data used for the machine-learning calibration to topo-clusters located in the high-granularity region $|\eta_{det}| < 2.5$ of the ATLAS calorimeters, where the *expressiveness* of their associated observables is expected to be the highest. It is defined using reconstructed calorimeter jet kinematics,

$$p_{T,jet}^{JES} > 20\,\text{GeV} \qquad \text{and} \qquad |y_{jet}^{JES}| < 2.0 \,, \tag{2}$$

where $p_{T,jet}^{JES}$ is the fully calibrated transverse momentum of the jet, and $y_{jet}^{JES}$ is its rapidity after all corrections.

In the context of this study particle jets in MC simulations are considered as the physics truth generating the detector truth introduced in Section 2.3.2. They are clustered from stable particles in the generated particle-level final state of the hard-scatter interaction, with the exception of muons and neutrinos, using the anti-$k_t$ algorithm with the same radius parameter used for the calorimeter jets. Only calorimeter jets within an angular distance $\Delta R < 0.4$ to the nearest particle jet are accepted, to assure a significant amount of true signal contribution to the topo-clusters they are clustered from. No pile-up-jet tagging or removal is applied to these simulated and matched calorimeter jets.

The topo-clusters are collected from the simulated calorimeter jets passing the selection above. Given the jet definition, they are found to be approximately within $|y_{clus}^{EM} - y_{jet}^{JES}| \lesssim 0.4$, with $y_{clus}^{EM}$ being the cluster rapidity reconstructed at the EM signal scale, as introduced in Eq. 1 of Section 2.3.1. To remove clusters generated by pile-up, all clusters considered for training, testing and validation of the calibration networks are required to have a true energy content of $E_{clus}^{dep} > 300\,\text{MeV}$. Most topo-clusters not satisfying this selection were found to have a high level of randomness in their signal and other associated observables, with only a weak or no relation to $E_{clus}^{dep}$ at all.[6] They are excluded as they are not significantly contributing to the hadronic final state (and jet) signals in the context of this study.

The collected sample of topo-clusters is randomised and all information from the jet context is dropped, as it is not relevant for training the networks and the performance evaluation of the learned calibration.

### 3.1.2 Feature selection for the calibration training

In addition to the kinematics represented by $E_{clus}^{EM}$ and $y_{clus}^{EM}$, each topo-cluster is associated with a set of observables that are canonically referred to as *cluster moments*. These are part of the general analysis data model in ATLAS. The moments were designed to provide signal characteristics including its timing and significance as well as selected measures indicative of the nature of the energy deposit in the topo-cluster, together with other measures that have impact on a cluster classification by signal sources and a local hadronic calibration.

---

[6] Per ATLAS convention, and following the discussion in Section 2.3.2, the topo-clusters generated purely by pile-up alone have $E_{clus}^{dep} = 0$. A meaningful target for calibrations requires $E_{clus}^{dep} > 0$, at least.

The topo-cluster moments that are expected to be useful for a local calibration are those that establish an observable space that represents the relation between the signal $E_{\text{clus}}^{\text{EM}}$ and its source $E_{\text{clus}}^{\text{dep}}$. This relation can only be directly established, within the limitations of the models applied, in MC simulations. When using a selection of these observables for a machine-learning-based multi-dimensional topo-cluster calibration, not only the individual observables but also the correlations between them are important.

In machine-learning, the inputs to a classification or regression network are often referred to as features. For learning the calibration, each topo-cluster is represented by a *feature set* $\mathcal{X}_{\text{clus}}$ that serves as input to a learned calibration network. There are several important aspects to be considered when collecting some of the features summarised in Table 1 into $\mathcal{X}_{\text{clus}}$. These mostly relay to

1. the expectation for the achievable accuracy of the calibration not only for the MC simulations but also for the application to experimental data;

2. the expressiveness of each feature with respect to the calorimeter signal source and its reflection in the signal definition given by the topological cell clustering;

3. and a meaningful feature reconstruction with sufficient resolution for almost all topo-clusters, it not all, reconstructed to measure the final state of the *pp* collisions in the ATLAS calorimeters.

Both aspects captured in item 1 and item 2, respectively, are tightly connected. Features with little sensitivity to the topo-cluster signal source are irrelevant as inputs and are thus omitted. Similarly, the selection of expressive but highly correlated features may lead to an unnecessarily large numerical system decreasing the computing performance and potentially introducing a higher level of *stochasticity* (noise).

The application of the topo-cluster calibration learned from MC simulations to experimental data requires that the individual features and their correlations are well modelled. This is generally achievable for individual features that are lower order topo-cluster moments, as documented in Ref. [2]. Several highly expressive second-order moments are not considered, as they are too sensitive to the (hadronic) shower modelling and the signal calculation in the simulation including pile-up. In particular, the modelling of pile-up contributions to these complex observables requires minimum bias physics and detector simulations to be accurate at often small scales that, at the present state-of-art, have not been achieved in all aspects. The second order moments that provide measurements of the longitudinal and lateral topo-cluster signal dispersion ($\langle \mathfrak{m}_{\text{long}}^2 \rangle$, $\langle \mathfrak{m}_{\text{lat}}^2 \rangle$) are included in $\mathcal{X}_{\text{clus}}$, because they were found to be largely insensitive to modelling issues and pile-up [2]. In addition, the signal compactness measure $p_{\text{T}}D$ and the energy-squared-weighted variance of the cell time distribution $\text{Var}_{\text{clus}}(t_{\text{cell}})$ inside the topo-cluster, both calculated as described in Appendix B.1, are included because by construction they are expected to have little sensitivity to potential (small scale) modelling issues.

The modelling of the feature correlations can only be evaluated in the context of a full performance study in which the trained calibration is applied to data, with well defined metrics. This study is out-of-scope for this method paper.

The final important aspect noted in item 3 in the list above, must be considered when composing $\mathcal{X}_{\text{clus}}$. All chosen features need to be reconstructed for each cluster in a large sample of topo-clusters in a meaningful way, with the highest possible resolution. This requires least impact from the topo-cluster formation algorithm and cell-level detector inefficiencies due to, e.g., insufficient granularity and noise, on their expressiveness for the calibration. For this reason, the jet phase space, and consequently the topo-cluster phase space, covered by this proof-of-principle study of the applicability of a ML calibration and the understanding of its predictions, is restricted to the ATLAS calorimeter region with the highest readout granularity, as mentioned in Section 3.1.1.

Table 1: The topo-cluster features used in the training of the topo-cluster calibration networks, as summarised in Eq. 3. The variables are defined in Ref. [2], except for $p_\mathrm{T}D$ and $\mathrm{Var}_\mathrm{clus}(t_\mathrm{cell})$ defined in Appendix B.

| Category | Symbol | Comment |
|---|---|---|
| Kinematics | $E^\mathrm{EM}_\mathrm{clus}$ | Signal at the electromagnetic energy scale |
| | $y^\mathrm{EM}_\mathrm{clus}$ | Rapidity at the electromagnetic energy scale |
| Signal strength, time structure | $\zeta^\mathrm{EM}_\mathrm{clus}$ | Signal significance |
| | $\mathrm{Var}_\mathrm{clus}(t_\mathrm{cell})$ | Variance of $t_\mathrm{cell}$ distribution |
| Shower depth, shower shape, compactness | $\lambda_\mathrm{clus}$ | Distance of the centre-of-gravity from the calorimeter front face (along principal cluster axis) |
| | $|\vec{c}_\mathrm{clus}|$ | Distance of the centre-of-gravity from the nominal vertex |
| | $f_\mathrm{emc}$ | Fraction of energy in the electromagnetic calorimeter |
| | $\langle \rho_\mathrm{cell} \rangle$ | Cluster signal density measure |
| | $\langle \mathfrak{m}^2_\mathrm{long} \rangle$ | Normalised energy dispersion along the principal cluster axis |
| | $\langle \mathfrak{m}^2_\mathrm{lat} \rangle$ | Normalised energy dispersion perpendicular to the principal cluster axis |
| | $p_\mathrm{T}D$ | Signal compactness measure (inspired by Ref. [30]) |
| Topology | $f_\mathrm{iso}$ | Cluster isolation measure |
| Pile-up | $t_\mathrm{clus}$ | Signal timing |
| | $N_\mathrm{PV}$ | Number of reconstructed primary vertices |
| | $\mu$ | Number of pile-up interactions per bunch crossing |

Even within the chosen phase space, very few topo-clusters in the used dataset may still not have enough cells or suffer from other deficiencies introduced by the detector or the collision environment such that it is not possible to reconstruct some or all of the features with the highest possible accuracy and precision. Nevertheless, and mostly to avoid selection biases and thus predict the most universal calibration function for all topo-clusters in the illuminated calorimeter region, these clusters are represented as well in both training and testing of the calibration network.

The final composition of $\mathcal{X}_\mathrm{clus}$ contains the 15 features described in Table 1,

$$
\mathcal{X}_\mathrm{clus} = \\
\left\{ \underbrace{E^\mathrm{EM}_\mathrm{clus}, y^\mathrm{EM}_\mathrm{clus}}_\text{kinematics}, \overbrace{\zeta^\mathrm{EM}_\mathrm{clus}}^\text{signal relevance}, \underbrace{\mathrm{Var}_\mathrm{clus}(t_\mathrm{cell}), \lambda_\mathrm{clus}, |\vec{c}_\mathrm{clus}|, \langle \rho_\mathrm{cell} \rangle, \langle \mathfrak{m}^2_\mathrm{long} \rangle, \langle \mathfrak{m}^2_\mathrm{lat} \rangle, p_\mathrm{T}D, f_\mathrm{emc}}_\text{shower nature (position, compactness, signal density and internal time structure)}, \overbrace{f_\mathrm{iso}}^\text{topology (isolation)}, \underbrace{t_\mathrm{clus}, N_\mathrm{PV}, \mu}_\text{pile-up} \right\}. \quad (3)
$$

It reflects the guidance discussed above.[7] Included are the kinematic variables $E^\mathrm{EM}_\mathrm{clus}$ and $y^\mathrm{EM}_\mathrm{clus}$, both basic inputs for any calorimeter calibration. Also included are all of the topo-cluster observables employed by the classification and the hadronic calibration in the LCW sequence, thus removing the need for an explicit cluster classification but preserving the information entering this step. In addition, the event-level pile-up measures $N_\mathrm{PV}$ and $\mu$ and the topo-cluster isolation measure $f_\mathrm{iso}$ are included to represent topological

---

[7] The DNN trained in the context of the studies presented in Ref. [4] employs the same feature set.

(varying cluster densities in jets) and environmental (pile-up) effects on the cluster signal. All members of $\mathcal{X}_{\text{clus}}$ are available for both experimental data and MC simulations.

## 3.2 Topo-cluster calibration training

The hadronic calibration in the LCW sequence aims at the (true) deposited energy $E_{\text{clus}}^{\text{dep}}$ as a calibration target. The determination of the local topo-cluster calibration using machine learning techniques introduces a set of novel aspects. Rather than learning $E_{\text{clus}}^{\text{dep}}$ with a typical value ranging from $\mathcal{O}(100\,\text{MeV})$ to $\mathcal{O}(1\,\text{TeV})$ directly, the response $\mathcal{R}_{\text{clus}}^{\text{EM}}$ as a function of the feature set $\mathcal{X}_{\text{clus}}$ from Eq. 3 is learned for each topo-cluster in a multi-dimensional regression fit. The target $\mathcal{R}_{\text{clus}}^{\text{EM}}$ is defined as the ratio of the cluster signal $E_{\text{clus}}^{\text{EM}}$ at EM scale and the deposited energy in the cluster $E_{\text{clus}}^{\text{dep}}$,

$$\mathcal{R}_{\text{clus}}^{\text{EM}} = E_{\text{clus}}^{\text{EM}}/E_{\text{clus}}^{\text{dep}} \ . \tag{4}$$

It is available in MC simulations for each topo-cluster extracted from the jet and selected according to the criteria introduced in Section 3.1.1. The significantly reduced value range for $\mathcal{R}_{\text{clus}}^{\text{EM}}$ lowers the computational effort in the training of the network by reducing the number of epochs, with the promise of a more accurate and precise calibration.

Both the BNN and the RE learn uncertainties related to the trained calibration for each topo-cluster individually. These uncertainties allow for a better understanding of (1) detector signal features, (2) possible signal quality issues in the data, and (3) possible fundamental limitations and other issues associated with the network training. They provide measures of the ultimate accuracy achieved with the trained calibration network.

### 3.2.1 Training target

In addition to using the topo-cluster response $\mathcal{R}_{\text{clus}}^{\text{EM}}$ as a calibration target, instead of the deposited energy $E_{\text{clus}}^{\text{dep}}$ that is the direct target of the LCW hadronic calibration, the dedicated classification step characteristic for the LCW procedure is omitted when training the BNN and the RE network. Instead, the features $E_{\text{clus}}^{\text{EM}}$, $y_{\text{clus}}^{\text{EM}}$, $\lambda_{\text{clus}}$ and $\langle\rho_{\text{cell}}\rangle \equiv \rho_{\text{clus}}$ used in the LCW classification are included, as presented in Table 1 and Eq. 3. The result of the BNN(RE) training is the individual response prediction $\mathcal{R}_{\text{clus}}^{\text{BNN(RE)}}(\mathcal{X}_{\text{clus}})$ for a given topo-cluster with a feature set $\mathcal{X}_{\text{clus}}$, with

$$\mathcal{R}_{\text{clus}}^{\text{BNN(RE)}}(\mathcal{X}_{\text{clus}}) \overset{\text{trained}}{=} \mathcal{R}_{\text{clus}}^{\text{EM}}(\mathcal{X}_{\text{clus}}) \ , \tag{5}$$

with $\mathcal{R}_{\text{clus}}^{\text{EM}}$ from Eq. 4. Both networks are independently trained to encode the respective representation $\mathcal{R}_{\text{clus}}^{\text{BNN(RE)}}$ of the target $\mathcal{R}_{\text{clus}}^{\text{EM}}$ as a function of $\mathcal{X}_{\text{clus}}$ given in Eq. 3. The derivation of the corresponding calibrated topo-cluster energy $E_{\text{clus}}^{\text{BNN(RE)}}$ then employs $\mathcal{R}_{\text{clus}}^{\text{BNN(RE)}}$ such that

$$E_{\text{clus}}^{\text{BNN(RE)}} = \frac{E_{\text{clus}}^{\text{EM}}}{\mathcal{R}_{\text{clus}}^{\text{BNN(RE)}}(\mathcal{X}_{\text{clus}})} \ . \tag{6}$$

Figure 1 shows the complex distributions for $E_{\text{clus}}^{\text{dep}}$, $E_{\text{clus}}^{\text{EM}}$ and $\mathcal{R}_{\text{clus}}^{\text{EM}}$ for the three topo-cluster categories defined in Section 2.3.2. The particular distribution shapes observed for both $E_{\text{clus}}^{\text{dep}}$ and $E_{\text{clus}}^{\text{EM}}$ are driven

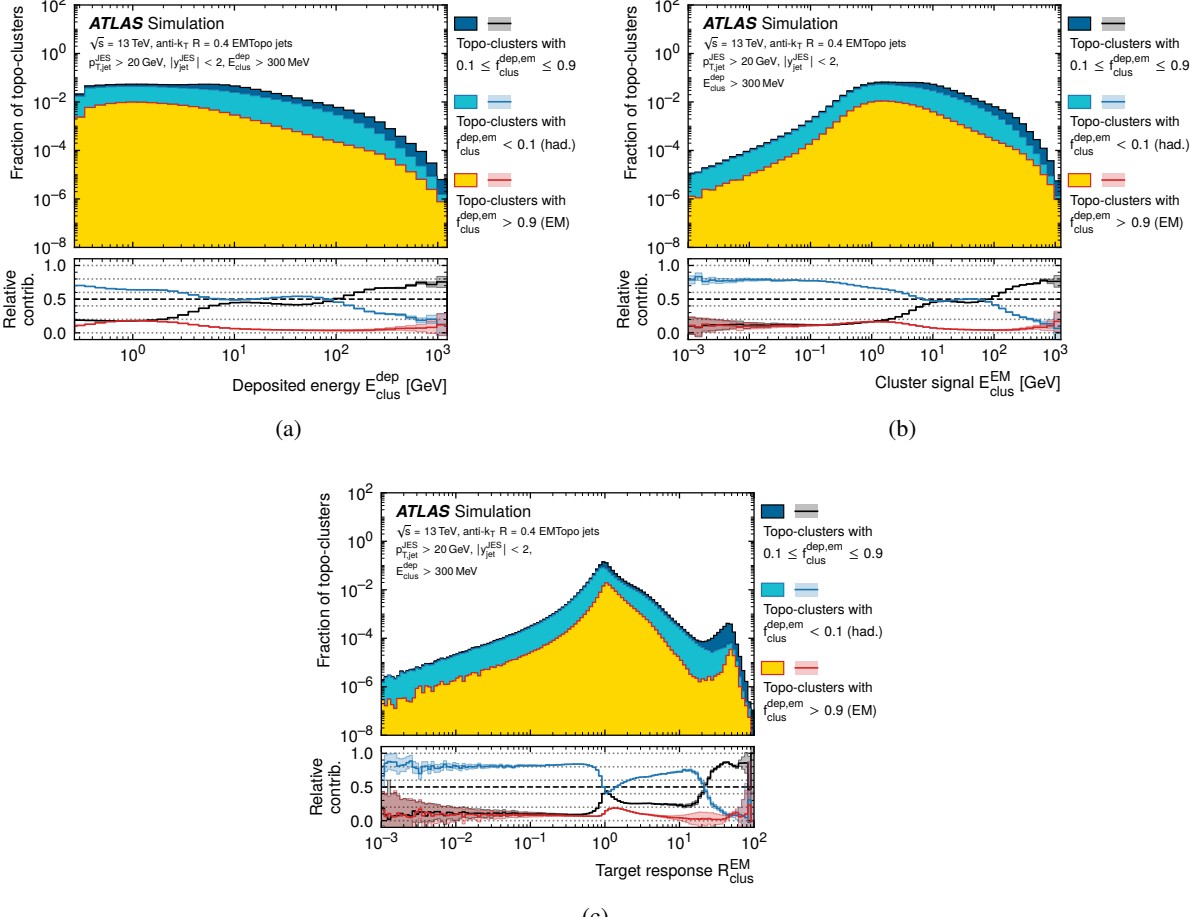

(a)

(b)

(c)

Figure 1: The distributions of the energy $E_{\mathrm{clus}}^{\mathrm{dep}}$ deposited in topo-clusters are shown in (a) in a stack, for the clusters categorised as electromagnetic, hadronic or composite by applying the classifications introduced in Section 2.3.2. In (b), the corresponding stacked distributions for the basic cluster signal $E_{\mathrm{clus}}^{\mathrm{EM}}$ at EM scale are shown. The distributions of the machine-learning training target, the topo-cluster response $\mathcal{R}_{\mathrm{clus}}^{\mathrm{EM}}$ constructed from $E_{\mathrm{clus}}^{\mathrm{EM}}$ and $E_{\mathrm{clus}}^{\mathrm{dep}}$ according to Eq. 4, are shown stacked in (c) for the same three categories. The distributions are filled from the sample of topo-clusters collected from fully simulated calorimeter jets, as described in Section 3.1.1. The respective lower panels show the relative contributions from the three categories to the inclusive spectrum. The shaded areas indicated the statistical errors.

by the multijet final state simulation configuration that determines the incoming flow of particles from the selected jets in the MC simulations, with its calorimeter representation subjected to the topo-cluster selection discussed in Section 3.1.1. The distributions provide an integral view on regional hadronic signal characteristics, like the varying levels of non-compensation across the illuminated topo-cluster rapidity range.

The contribution of electromagnetic topo-clusters to the $E_{\mathrm{clus}}^{\mathrm{EM}}$ distribution in Figure 1(b) is near constant for $E_{\mathrm{clus}}^{\mathrm{EM}} \lesssim 2\,\mathrm{GeV}$. The lower $E_{\mathrm{clus}}^{\mathrm{EM}}$ regime is largely populated by hadronic topo-clusters, with their relative contribution dropping to match the one from composite topo-clusters for $10\,\mathrm{GeV} \lesssim E_{\mathrm{clus}}^{\mathrm{EM}} \lesssim 100\,\mathrm{GeV}$. Composite topo-clusters then dominate in the high signal regime $E_{\mathrm{clus}}^{\mathrm{EM}} > 100\,\mathrm{GeV}$, indicating a dominant contribution from clusters generated by merged showers and shower fragments.

Table 2: Summary of the transformations applied to the various observables collected into the feature set $\mathcal{X}_{\text{clus}}$ from Eq. 3, and the target response $\mathcal{R}^{\text{EM}}_{\text{clus}}$ from Eq. 4.

| Transformation (preprocessing) | Features |
|---|---|
| Logarithmic ($\log_{10}$) standardisation (Eq. 7) | $E^{\text{EM}}_{\text{clus}}, \zeta^{\text{EM}}_{\text{clus}}, \text{Var}_{\text{clus}}(t_{\text{cell}}), \lambda_{\text{clus}}, \langle \rho_{\text{cell}} \rangle$ |
| Linear standardisation (Eq. 8) | $y^{\text{EM}}_{\text{clus}}, |\vec{c}_{\text{clus}}|, f_{\text{emc}}, \langle \text{m}^2_{\text{long}} \rangle, \langle \text{m}^2_{\text{lat}} \rangle, p_{\text{T}}D, f_{\text{iso}}, N_{\text{PV}}, \mu$ |
| Maximum-absolute normalisation (Eq. 9) | $t_{\text{clus}}$ |
| Logarithmic ($\log_{10}$) transformation (Eq. 10) | $\mathcal{R}^{\text{EM}}_{\text{clus}}$ |

The resulting distributions of $\mathcal{R}^{\text{EM}}_{\text{clus}}$ shown on Figure 1(c) for the topo-cluster categories depict the complexity of the calibration task. Small $\mathcal{R}^{\text{EM}}_{\text{clus}} \ll 1$ indicate that the contribution of the deposited energy $E^{\text{dep}}_{\text{clus}}$ to the topo-cluster signal $E^{\text{EM}}_{\text{clus}}$ is insignificant. This is the case for $E^{\text{dep}}_{\text{clus}}$ predominantly arising from stopped particles with invisible decay products like neutrinos together with a small fraction of $E^{\text{dep}}_{\text{clus}}$ generating the signal, or in deep inelastic hadronic interactions where a large amount of invisible energy is produced. In addition, the shower-generated signal reflecting $E^{\text{dep}}_{\text{clus}}$ may be suppressed because the total signal $E^{\text{EM}}_{\text{clus}}$ of the topo-cluster is dominated by negative cell signals from out-of-time pile-up noise [2].

Most topo-clusters with large responses $\mathcal{R}^{\text{EM}}_{\text{clus}} \gtrsim 10$ are located in the Tile gap region [31], where the response reflects detector effects that are not only related to pile-up. These are further discussed in the context of the predicted uncertainties in Section 5.5. In addition, large $\mathcal{R}^{\text{EM}}_{\text{clus}}$ also indicate significant signal contributions from (predominantly in-time) pile-up that are not related to $E^{\text{dep}}_{\text{clus}}$. The affected topo-clusters are typically located in detector regions closer to the $pp$ collision region. These comprise the electromagnetic calorimeters and the Tile gap scintillators, where the exposure to the flow of particles from pile-up is largest.

Generally, and in extension to the interpretations of the $E^{\text{dep}}_{\text{clus}}$ and $E^{\text{EM}}_{\text{clus}}$ spectra given above, the complexity of the $\mathcal{R}^{\text{EM}}_{\text{clus}}$ distributions reflect the incoming particle flow, the collision environment, and the varying detector geometries and, consequently, the varying calorimeter signal characteristics.

### 3.2.2 Feature and target transformations and projections

To limit the numerical value range of the network inputs and to create more appropriate (peaked) feature distributions for the training whenever possible, the features are transformed following three different procedures. If the spectrum of a feature $x \in \mathcal{X}_{\text{clus}}$ is smoothly falling and $x > 0$ always, $x$ is replaced by a *standardised logarithmic representation*,

$$x \rightarrow \frac{\log_{10} x - \langle \log_{10} x \rangle}{\text{std}(\log_{10} x)} \ . \tag{7}$$

Here $\langle \log_{10} x \rangle$ is the statistical average of the $\log_{10} x$ distribution from the training sample, and $\text{std}(\log_{10} x)$ denotes the corresponding standard deviation of the transformed features $x$.

For other distribution shapes, $x$ is standardised without additional transformation (*linear standardisation*),

$$x \rightarrow \frac{x - \langle x \rangle}{\text{std}\, x} \ , \tag{8}$$

Table 3: The architecture and hyper-parameters for both the BNN and the RE network with the three-mode Gaussian mixture likelihood. Both networks feature a decaying learning rate (LR) implemented by the learning-rate scheduler STEPLR from the PYTORCH [32] software tool kit, with LR reduced by a factor $\gamma$ at each of the epochs within the range specified in braces below.

| Hyper-parameter | Network architecture and setup |
| --- | --- |
| Likelihood model | Gaussian mixture |
| Number of mixture modes $N_{\text{mix}}$ | 3 |
| Number of hidden layers | 4 |
| Number of nodes per layer (input, hidden, and output) | $\{15, 64, 64, 64, 64, 9\}$ |
| Activation function (inner layers) | rectified linear unit (ReLU) |
| Activation function (last layer) | none |
| Central-value prediction (mean or maximum) | maximum of the likelihood |
| Optimizer and learning rate (LR) | ADAM [33] with LR $= 10^{-4}$ |
| Learning-rate scheduler | STEPLR, epochs $\{25, 100\}$, $\gamma = 0.1$ |
| Number of training epochs | 150 |
| Training (and inference) batch size | 4096 (512) |
| Dataset size (all selected topo-clusters) | $14.5 \times 10^6$ |
| Data sample size for training ($N_{\text{train}}$) | $8.7 \times 10^6$ (60.0%) |
| Data sample size for validation | $5.0 \times 10^5$ (3.4%) |
| Data sample size for testing | $5.3 \times 10^6$ (36.6%) |
| Sampling at inference on the test dataset (BNN only, $N$) | 50 times |
| Number of ensemble members (RE only, $N$) | 50 |

where $\langle x \rangle$ and std $x$ are the mean and the standard deviation of the distribution of $x$ from the training sample. An exception is the cluster time $t_{\text{clus}}$, which is normalised to its maximum absolute value $\max(|t_{\text{clus}}|)$ found in the training sample, to keep its sparsity (*maximum-absolute normalisation*),

$$t_{\text{clus}} \rightarrow \frac{t_{\text{clus}}}{\max(|t_{\text{clus}}|)} \ . \tag{9}$$

For the regression target, a simple logarithmic transform is applied to reduce its numerical range further,

$$\mathcal{R}_{\text{clus}}^{\text{EM}} \rightarrow \log_{10} \mathcal{R}_{\text{clus}}^{\text{EM}} \ . \tag{10}$$

The transformations of the individual features listed in Table 1 and the regression target from Eq. 4 are summarised in Table 2.

## 3.3 Models and network setup

To learn the response given in Eq. 4 as a function of $\mathcal{X}_{\text{clus}}$, both a Bayesian neural network and a repulsive ensemble are employed. Both are configured with the same network model, with the hyper-parameters presented in Table 3. They differ significantly in the implementation of the variational approach to

determine the uncertainties, and thus in the set of learned network parameters and the respective loss functions, as further discussed in Sections 3.3.2 and 3.3.4.

### 3.3.1 Principle design of a Bayesian neural network

As previously mentioned, Bayesian neural networks do not only predict the target of regressions like the one exercised in the context of the topo-cluster calibration, but also provide uncertainty predictions associated with the central value prediction. Figure 2 illustrates the principle setup for the training of the BNN and its inference on the (test) data.

The most significant deviation of the BNN model from the usual deep neural network design is that the weights connecting the nodes in adjacent layers are not trained as fixed values, rather they are represented by learned average values and widths of probability distributions $q(\theta)$.[8] Here $\theta$ are the trained network parameters. When the BNN is inferred on test data, these distributions are sampled a fixed number of times, denoted by $N$ and set to $N = 50$ for this result. This is the same as collecting the predictions from $N$ different networks trained using the same data, without the need to do this training $N$ times. The central prediction $\mathcal{R}_{\text{clus}}^{\text{BNN}}$ is then constructed from the $N$ sampled predictions such that it reflects the prediction with the highest probability to represent the target $\mathcal{R}_{\text{clus}}^{\text{EM}}$, as indicated in Figure 2. The uncertainties are constructed as indicated in the same figure. They are further discussed in Section 5.

### 3.3.2 BNN loss function

The loss function employed in the BNN training is designed to maximise the likelihood $p(\theta|D_{\text{train}})$ of the network parameters $\theta$ to describe the training dataset $D_{\text{train}}$. This dataset is represented by a vector of target and feature set pairs $\left\{\mathcal{R}_{\text{clus}}^{\text{EM}}, \mathcal{X}_{\text{clus}}\right\}_j$ for $j = 1, \ldots, N_{\text{train}}$ topo-clusters,

$$D_{\text{train}} = \left(\left\{\mathcal{R}_{\text{clus}}^{\text{EM}}, \mathcal{X}_{\text{clus}}\right\}_1, \left\{\mathcal{R}_{\text{clus}}^{\text{EM}}, \mathcal{X}_{\text{clus}}\right\}_2, \ldots, \left\{\mathcal{R}_{\text{clus}}^{\text{EM}}, \mathcal{X}_{\text{clus}}\right\}_{N_{\text{train}}}\right) . \tag{11}$$

The form of $p(\theta|D_{\text{train}})$ is not known a priori. Therefore, and following Bayes' theorem, the posterior $p(D_{\text{train}}|\theta)$ that provides the likelihood that $D_{\text{train}}$ describes $\theta$, is maximised instead of $p(\theta|D_{\text{train}})$, as discussed in Ref. [3]. Assuming a Gaussian likelihood for $p(D_{\text{train}}|\theta)$ is well motivated, but in case of the topo-cluster-response predictions, a mixture of $N_{\text{mix}} = 3$ Gaussian functions is chosen to accommodate possible deviations from this shape,

$$p(D_{\text{train}}|\theta) = p(\{\mathcal{R}_{\text{clus}}^{\text{EM}}\}|\theta, \{\mathcal{X}_{\text{clus}}\}) = \prod_{j=1}^{N_{\text{train}}} \sum_{i=1}^{N_{\text{mix}}} \alpha_{\theta,i} \, \mathcal{N}(\mathcal{R}_{\text{clus},j}^{\text{EM}}|\langle\mathcal{R}\rangle_{\theta,i}, \sigma_{\theta,i}) . \tag{12}$$

Here $\alpha_{\theta,i}$ are the weights with which the Gaussian functions $\mathcal{N}(\mathcal{R}_{\text{clus}}^{\text{EM}}|\langle\mathcal{R}\rangle_{\theta,i}, \sigma_{\theta,i})$ with mean $\langle\mathcal{R}\rangle_{\theta,i}$ and standard deviation $\sigma_{\theta,i}$ contribute to $p(D_{\text{train}}|\theta)$, with

$$\alpha_{\theta,i} \geq 0 \qquad \text{and} \qquad \sum_{i=1}^{N_{\text{mix}}} \alpha_{\theta,i} = 1 .$$

---

[8] The distributions $q(\theta)$ are chosen to be Gaussian functions. This is not required but enhances computational performance without observable effects on the accuracy of the predictions.

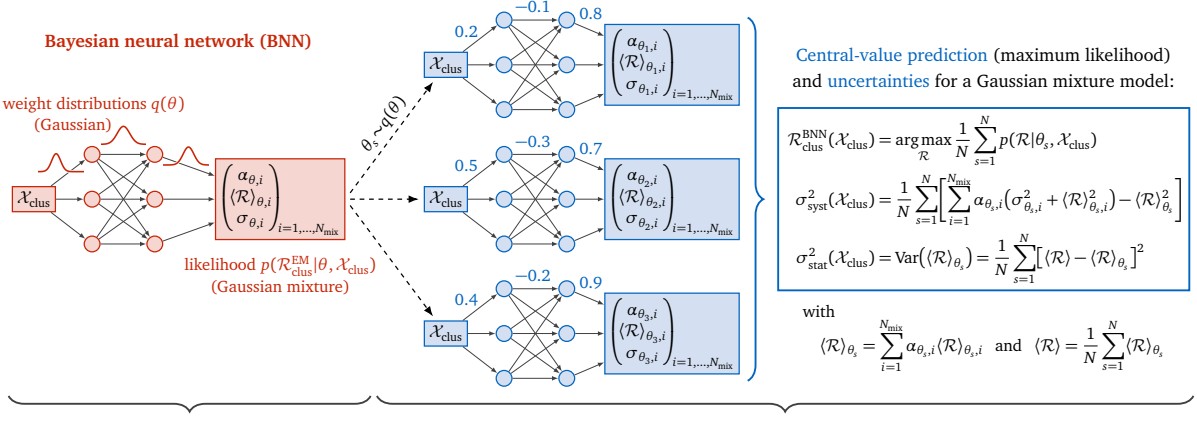

Figure 2: The principal design of the Bayesian neural network (BNN) employed for the regression fits and the corresponding uncertainty predictions for the topo-cluster calibration. The network on the left is designed such that the resulting weights linking the nodes of adjacent layers are described by weight functions $q(\theta)$ with learned averages and widths. The inference of the trained model, represented by a set of learned network parameters $\theta$, is shown on the right, where an ensemble of $N$ networks is sampled from $q(\theta)$ to generate $N$ predictions from the model. The central prediction $\mathcal{R}_{\mathrm{clus}}^{\mathrm{BNN}}$ is the mode (most probable value) of the average of $N$ probability density functions $p(\mathcal{R}|\theta_s, \mathcal{X}_{\mathrm{clus}})$, where each individual $p(\mathcal{R}|\theta_s, \mathcal{X}_{\mathrm{clus}})$ represents the probability that the network describes the target $\mathcal{R} = \mathcal{R}_{\mathrm{clus}}^{\mathrm{EM}}$ for a given topo-cluster with feature set $\mathcal{X}_{\mathrm{clus}}$. Each of the $p(\mathcal{R}|\theta_s, \mathcal{X}_{\mathrm{clus}})$ is defined as the mixture of $i = 1 \ldots N_{\mathrm{mix}}$ Gaussian distributions with learned means $\langle \mathcal{R} \rangle_{\theta_s, i}$, widths $\sigma_{\theta_s, i}$ and coefficients $\alpha_{\theta_s, i}$, as introduced in Eq. 12, with $N_{\mathrm{mix}} = 3$. The averaged function is the normalised sum of all probability density functions obtained by sampling weights from $q(\theta)$ $N$ times, with $N = 50$. It is composed of a total of $N \times N_{\mathrm{mix}}$ Gaussian distributions. The weighted average response $\langle \mathcal{R} \rangle_{\theta_s}$ is the sum of $\alpha_{\theta_s, i} \langle \mathcal{R} \rangle_{\theta_s, i}$ from the $N_{\mathrm{mix}}$ distributions for each sample $s$. It is needed to predict the systematic ($\sigma_{\mathrm{syst}}^{\mathrm{BNN}}(\mathcal{X}_{\mathrm{clus}})$) and statistical ($\sigma_{\mathrm{stat}}^{\mathrm{BNN}}(\mathcal{X}_{\mathrm{clus}})$) uncertainties. It is calculated for each of the $N$ sampled sets of network parameters $\theta_s$ to obtain $\langle \mathcal{R} \rangle$, the mean response averaged over the $N$ corresponding $\langle \mathcal{R} \rangle_{\theta_s}$. This mean is needed exclusively for the calculation of $\sigma_{\mathrm{stat}}^{\mathrm{BNN}}$. All calculations are individually performed for any given topo-cluster at inference of the trained BNN model. The numbers written above some of the edges (links) between nodes are weights sampled from $q(\theta)$ at network inference. They are for illustration only and show the sampling character of the predictions.

All $\alpha_{\theta, i}$, $\langle \mathcal{R} \rangle_{\theta, i}$ and $\sigma_{\theta, i}$ are predicted by the BNN and are thus functions of $\mathcal{X}_{\mathrm{clus}}$. The mixture weights $\alpha_{\theta, i}$ are forced to be positive and to add up to unity by applying a *softmax* [34, 35] function.

To find the maximum of $p(D_{\mathrm{train}}|\theta)$, the negative logarithmic likelihood $-\log p(D_{\mathrm{train}}|\theta)$ is minimised. Without further modifications, this minimisation tends to prefer larger values of $\sigma_{\theta, i}$ in the exponents of the Gaussian functions. To avoid trivial solutions $\sigma_{\theta, i} \to \infty$ and to generally restrict the values of $\sigma_{\theta, i}$ to meaningful ranges, a Kullback-Leibler divergence $D_{\mathrm{KL}}[q(\theta), p_{\mathrm{prior}}(\theta)]$ [36] is introduced to act as a regulator in the BNN loss function $\mathcal{L}_{\mathrm{BNN}}$,

$$\mathcal{L}_{\mathrm{BNN}} = D_{\mathrm{KL}}[q(\theta), p_{\mathrm{prior}}(\theta)] - \underbrace{\big\langle \log p(D_{\mathrm{train}}|\theta) \big\rangle_{\theta \sim q(\theta)}}_{\substack{\theta \text{ sampled} \\ \text{from } q(\theta)}}, \tag{13}$$

The $D_{\mathrm{KL}}[q(\theta), p_{\mathrm{prior}}(\theta)]$ term vanishes when the distribution of learned network parameters $q(\theta)$ matches the prior $p_{\mathrm{prior}}(\theta)$. Both $q(\theta)$ and $p_{\mathrm{prior}}(\theta)$ are chosen to be Gaussians so that $D_{\mathrm{KL}}[q(\theta), p_{\mathrm{prior}}(\theta)]$ can be calculated analytically. The prior mean and standard deviation are set to zero and one, respectively. The

**Repulsive ensemble (RE)**

repulsive term
(Gaussian kernel)

**Central-value prediction** (maximum likelihood)
and uncertainties for a Gaussian mixture model:

$$\mathcal{R}^{\text{RE}}_{\text{clus}}(\mathcal{X}_{\text{clus}}) = \arg\max_{\mathcal{R}} \frac{1}{N} \sum_{s=1}^{N} p(\mathcal{R}|\theta_s, \mathcal{X}_{\text{clus}})$$

$$\sigma^2_{\text{syst}}(\mathcal{X}_{\text{clus}}) = \frac{1}{N} \sum_{s=1}^{N} \left[ \sum_{i=1}^{N_{\text{mix}}} \alpha_{\theta_s,i} \left( \sigma^2_{\theta_s,i} + \langle \mathcal{R} \rangle^2_{\theta_s,i} \right) - \langle \mathcal{R} \rangle^2_{\theta_s} \right]$$

$$\sigma^2_{\text{stat}}(\mathcal{X}_{\text{clus}}) = \text{Var}\left( \langle \mathcal{R} \rangle_{\theta_s} \right) = \frac{1}{N} \sum_{s=1}^{N} \left[ \langle \mathcal{R} \rangle - \langle \mathcal{R} \rangle_{\theta_s} \right]^2$$

with

$$\langle \mathcal{R} \rangle_{\theta_s} = \sum_{i=1}^{N_{\text{mix}}} \alpha_{\theta_s,i} \langle \mathcal{R} \rangle_{\theta_s,i} \quad \text{and} \quad \langle \mathcal{R} \rangle = \frac{1}{N} \sum_{s=1}^{N} \langle \mathcal{R} \rangle_{\theta_s}$$

**Training:** Repulsive term connecting the function space of all $N$ simultaneously trained networks forces the ensemble to spread out and cover the loss around the actual minimum

**Inference:** Same formulas as for the BNN, using the $N$ simultaneously trained ensemble members

Figure 3: Schematics describing the design of the repulsive ensemble (RE) model employed for the regression fits and the corresponding uncertainty predictions of the topo-cluster calibration. An ensemble of $N$ networks with identical configurations is configured. These $N$ networks are trained simultaneously with interconnections between their loss functions acting as repulsive forces between them (shown as spring connectors in the schematic). The predictions from the $N$ networks are collected for each topo-cluster in each processed batch during training. A central prediction and the uncertainties are calculated in the same way as illustrated in Figure 2 for the BNN, with the same nomenclature introduced there. The numbers written above some of the edges (links) between nodes illustrate the learned weights for one of the $N$ repulsive ensembles. They are shown to illustrate that each network fits its own weights within the repulsive action of the loss function.

likelihood $p(D_{\text{train}}|\theta)$ is taken from Eq. 12 and the network parameters $\theta$ are sampled once from $q(\theta)$ to calculate the average negative logarithmic likelihood term in Eq. 13. More details of the motivation and the design of $\mathcal{L}_{\text{BNN}}$ are given in Appendix C.1.

Compared to a regression training without uncertainty predictions, the network output in case of a loss defined by a single Gaussian function, configured by using $N_{\text{mix}} = 1$ in Eq. 12, is doubled. It consists of a local central value function as a representation of the regression target and a local uncertainty map. Generally, the number of network outputs increases by a factor of $3N_{\text{mix}} - 1$ for a mixture of $N_{\text{mix}}$ Gaussian distributions.

The predictions from the BNN are a novel approach to learn local uncertainty contributions to calorimeter signal calibrations. They potentially represent important contributions to the nuisances characterising the overall local systematic uncertainties.

### 3.3.3 Repulsive ensembles

To gain confidence in the BNN uncertainty predictions for the local topo-cluster calibration, an alternative and independent model for learning such uncertainties employing a network of repulsive ensembles, is considered. The basic idea of RE is to determine uncertainties for the learned parameters $\theta$, and consequently the functions they describe, by forcing ensembles of identically configured training networks to not predict the same best-fit network parameters, but rather explore the landscape around the corresponding minimum of the loss. While some variation of the predictions between members of an ensemble of networks can be observed due to random network initialisation and random training batch selection, there is no explicit

mechanism that guarantees that the full weight space encoding $p(D_{\text{train}}|\theta)$ is mapped out. It can be shown that adding a repulsive force (network interaction) to the usual update rule that minimises $-\log p(D_{\text{train}}|\theta)$ based on gradient descent, maps the weight space completely when applied to all members of the network ensemble simultaneously at each update $t \to t+1$ evolving $\theta^t \to \theta^{t+1}$. This procedure allows predicting uncertainties, see e.g., the comprehensive discussion in Ref. [3]. The basic design of the RE training model is sketched in Figure 3. While the formal determination of the central prediction and the uncertainties shown in this figure is identical to the one for the BNN shown in Figure 2, the underlying sampling strategies are fundamentally different. The BNN directly samples in weight space, while the RE samples in the function space establishing the repulsive force between networks. Some relevant details of the complex theoretical foundation and the implementation of RE are given in Appendix C.2.

### 3.3.4 Repulsive ensemble loss function

The loss function of the RE network is derived from the modified update rule. Its principal form is a sum of $i = 1, \ldots, N$ terms $\mathcal{L}_{\text{RE},i}$ for an ensemble of $N$ networks,

$$\mathcal{L}_{\text{RE}} = \sum_{i=1}^{N} \mathcal{L}_{\text{RE},i} \ . \tag{14}$$

Each $\mathcal{L}_{\text{RE},i}$ has three components comprising

1. the negative logarithm of a posterior likelihood $p(D_{\text{train}}|\theta)$ of Bayes' theorem representing $p(\theta|D_{\text{train}})$, the likelihood of the network parameters $\theta$ to describe the training dataset $D_{\text{train}}$ in Eq. 11, similar to the approach in the design of the BNN loss function in Section 3.3.2;

2. a repulsive term connecting the loss function space of all $N$ networks with a given network $i \in [1, N]$ using Gaussian kernels;

3. and a Gaussian prior introduced by the application of Bayes' theorem when representing $p(\theta|D_{\text{train}})$ by $p(D_{\text{train}}|\theta)$, again similar to what is introduced for the BNN in Section 3.3.2.

The gradient of $\mathcal{L}_{\text{RE}}$ is computed relative to all network parameters $\theta$ in the ensemble. The repulsive term in item 2 includes a *stop-gradient operation* on all $\theta$ in the ensemble, to assure valid gradients at each update. This term and the prior are also normalised by $1/N_{\text{train}}$ to avoid any scaling with the size of the training sample $N_{\text{train}}$. More insight on the formalism of the RE loss function is given in Appendix C.3.

## 4 Performance evaluations

The performance of the learned topo-cluster responses is fully evaluated for the BNN predictions ($\mathcal{R}_{\text{clus}}^{\text{BNN}}$) by applying the trained model to a test dataset that has no content overlap with the training dataset. Principal measures to determine the quality of this calibration are prediction power, the signal linearity and the local energy resolution. The median is used as a measure of the central tendency of the analysed distributions, because, unlike the mode, it is analytically accessible without explicit determining (fitting) the shape of the distribution around the most probable value. It is also relatively stable with respect to sample sizes, and thus better defined than the statistical mean.

In addition, the performance of the BNN-derived topo-cluster calibration is compared to the one learned by the RE. Both calibrations are expected to produce the same central predictions and thus yield the same median signal linearity and energy resolution. This is a necessary requirement for a meaningful comparison of the uncertainties predicted by these two models.

## 4.1 Evaluation metrics

Determining the performance of the ML calibration cluster-by-cluster includes the evaluation of its accuracy or *prediction power*, similar to a closure test of a given functional model to predict the target in any calibration attempt, including non-machine-learning-based ones.[9] In addition, the *signal linearity* achieved for the ML-calibrated energies are compared with the corresponding ones at other (calibrated) energy scales by determining their respective proportionality to the energy $E_{\text{clus}}^{\text{dep}}$ deposited in the topo-cluster. Both prediction power and signal linearity are trivially connected for each topo-cluster, but show different distribution shapes for the test variables defined in Sections 4.1.1 and 4.1.2 below, leading to different measures of both centrality and width.

The energy resolution for individual topo-clusters is determined before any local calibration and after application of the ML- and the LCW-based calibrations. These corresponding comparisons allow identifying the best approach to mitigate calorimeter- and collision-environment-induced signal fluctuations affecting the cluster signal quality. The immediate expectation here is that calibration functions that are highly sensitive to the signal source yield the best resolution power.

The learned uncertainties are only available from the BNN and the RE network, and are further discussed in Section 5.

### 4.1.1 Accuracy of the learned response

To evaluate the accuracy of the ML calibrations, the test dataset is used to compare the outputs $\mathcal{R}_{\text{clus}}^{\kappa}$ of the networks $\kappa$ for individual topo-clusters with the respective training target $\mathcal{R}_{\text{clus}}^{\text{EM}}$. The deviation from an accurate prediction $\Delta_{\mathcal{R}}^{\kappa}$ is introduced as a measure of the prediction power of the trained networks, with

$$\Delta_{\mathcal{R}}^{\kappa} = \frac{\mathcal{R}_{\text{clus}}^{\kappa}}{\mathcal{R}_{\text{clus}}^{\text{EM}}} - 1 \quad \text{with} \quad \kappa \in \{\text{DNN}, \text{BNN}\} \ . \tag{15}$$

It is calculated for each cluster in the test dataset with the expectation to approach zero for accurate predictions (highest prediction power). The prediction power of the DNN is taken from Ref. [4], which uses the same datasets.

### 4.1.2 Signal linearity

A fully calibrated calorimeter signal is expected to be directly proportional to a reference energy, which can be the incoming particle energy or, as in the case of the topo-cluster calibration, the locally deposited

---

[9] Closure tests are usually performed using the same dataset with which the calibration was derived. In the context of the machine-learning-derived calibration, an independent dataset is used for this test.

energy $E_{\text{clus}}^{\text{dep}}$ introduced in Section 2.3.2. The signal linearity can be measured cluster-by-cluster by $\Delta_E^\kappa$, which quantifies its deviation from perfect linearity by[10]

$$\Delta_E^\kappa = \frac{E_{\text{clus}}^\kappa}{E_{\text{clus}}^{\text{dep}}} - 1 \quad \text{with} \quad \kappa \in \{\text{EM}, \text{had}, \text{DNN}, \text{BNN}\} \ . \tag{16}$$

The considered topo-cluster energy scales $\kappa$ are the basic EM scale given in Eq. 1, the hadronic (had) energy scale provided in the LCW sequence presented in Section 2.3.1, and the ML calibrations from the DNN [4] and the BNN, with the respective response predictions applied according to Eq. 6 in Section 3.2. The cluster signal at EM scale is not expected to be linear, due to the non-compensating character of the ATLAS calorimeters and the absence of corrections for that at this scale. It is therefore not considered to be calibrated, unlike all other considered energy scales, in the context of the results presented here. Nevertheless, it provides a baseline for the signal-related topo-cluster performance. It is expected that

$$|\Delta_E^{\text{EM}}| \geq \Delta_E^\kappa \quad \forall \quad \kappa \in \{\text{had}, \text{DNN}, \text{BNN}\}$$

over the full evaluation space.[11] Considerable deviations of $\Delta_E^{\text{EM}}$ from zero are expected especially for small $E_{\text{clus}}^{\text{dep}}$ and in pile-up-dominated regimes, where its effect on the signal $E_{\text{clus}}^{\text{EM}}$ is large.

Calibrated energy scales are supposed to provide signals directly proportional to $E_{\text{clus}}^{\text{dep}}$. A residual level of fluctuations can be expected after topo-cluster calibration, due to the highly stochastic nature of its response. Nevertheless, the chosen measure of centrality, the median deviation from linearity $\langle \Delta_E \rangle_{\text{med}}$, is expected to be zero after calibrations are applied. Any deviation from this expectation can affect the measurement of final state objects like particles, jets and event shapes composed of topo-clusters in terms of the introduction of kinematic biases or shifts, an increase of systematic uncertainties and a potential degradation of the resolution.

### 4.1.3 Relative local energy resolution

The measure of the relative local energy resolution $\sigma_{\text{rel}}^E(E_{\text{clus}}^\kappa/E_{\text{clus}}^{\text{dep}})$ employs the 68% inter-quantile range $Q_{f=68\%}^w$ and the median of the respective $\Delta_E^\kappa + 1 = E_{\text{clus}}^\kappa/E_{\text{clus}}^{\text{dep}}$ distribution,

$$\sigma_{\text{rel}}^E(E_{\text{clus}}^\kappa/E_{\text{clus}}^{\text{dep}}) = \frac{Q_{f=68\%}^w(E_{\text{clus}}^\kappa/E_{\text{clus}}^{\text{dep}})}{2\langle E_{\text{clus}}^\kappa/E_{\text{clus}}^{\text{dep}}\rangle_{\text{med}}} \quad \text{with} \quad \kappa \in \{\text{EM}, \text{had}, \text{DNN}, \text{BNN}\} \ , \tag{17}$$

on the various energy scales $\kappa$. Its construction is based on Eqs. 37 and 39 given in Appendix D.1.

---

[10] The prediction power represented by $\Delta_{\mathcal{R}}$ and the signal linearity represented by $\Delta_E$ are related by the response definition given in Eq. 4 and the reconstruction of the calibrated energy given in Eq. 6, such that $\Delta_E = -\Delta_{\mathcal{R}}/(\Delta_{\mathcal{R}} + 1)$. This generally leads to two different distributions with possibly different central values, $\langle \Delta_{\mathcal{R}} \rangle_{\text{med}} \neq \langle \Delta_E \rangle_{\text{med}}$, and widths for any selected sample of topo-clusters, and thus to differences in the performance. Only when the response is perfectly learned, indicated by $\Delta_{\mathcal{R}} = 0$, a perfect signal linearity $\Delta_E = 0$ is achieved for a given topo-cluster.

[11] The EM-scale signal $E_{\text{clus}}^{\text{EM}}$ represents $E_{\text{clus}}^{\text{dep}}$ better for higher topo-cluster energies $E_{\text{clus}}^{\text{EM}} \approx E_{\text{clus}}^{\text{dep}}$ for $E_{\text{clus}}^{\text{dep}} \gtrsim 200\,\text{GeV}$.

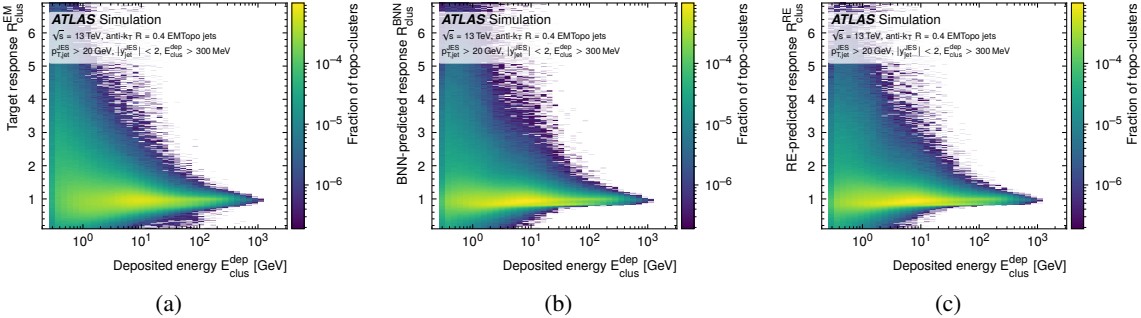

(a)                      (b)                      (c)

Figure 4: The topo-cluster response $\mathcal{R}_{\text{clus}}$ is evaluated cluster-by-cluster as a function of the energy $E_{\text{clus}}^{\text{dep}}$ deposited in the cluster. In (a), the distribution of the response at EM scale $\mathcal{R}_{\text{clus}}^{\text{EM}}$ (the training target) is shown, while (b) shows the distribution of the response $\mathcal{R}_{\text{clus}}^{\text{BNN}}$ predicted by the trained BNN. The corresponding distribution of the response $\mathcal{R}_{\text{clus}}^{\text{RE}}$ predicted by the RE is shown in (c). The topo-clusters are extracted from MC simulations, as described in Section 3.1.1.

## 4.2 Prediction power

The prediction power of the ML-learned topo-cluster calibration can be evaluated cluster-by-cluster as functions of various variables available from MC simulations and as functions of any of the features from the feature set $\mathcal{X}_{\text{clus}}$ used for the training, for both the BNN-derived and the RE-derived calibrations. A qualitative analysis allows comparing distribution shapes for the predictions from the networks with the corresponding distributions for the target, without the need to introduce statistical measures that may not describe the overall distribution well. The quantitative analysis introduces measures for the central values of these distributions that allow comparing them directly using e.g., $\Delta_{\mathcal{R}}$ from Eq. 15.

### 4.2.1 Distribution shapes

A qualitative comparison of the distributions of the training target $\mathcal{R}_{\text{clus}}^{\text{EM}}$ and the predictions from both the BNN and the RE, all as a function of the energy $E_{\text{clus}}^{\text{dep}}$ deposited in the topo-cluster, is shown in Figure 4. The target distribution in Figure 4(a) is reproduced well by both the BNN (Figure 4(b)) and RE (Figure 4(c)) predictions. Some small shifts in the populations between target and predictions can be observed in that the networks provide a sharper image of regions of higher population density. These indicate that the learned $\mathcal{R}_{\text{clus}}^{\text{EM}}$ is dominated by the statistically better illuminated regions, thus giving a qualitative impression of the limitations of the networks to predict outliers. With this respect, both the BNN and the RE network show very similar behaviour.

Further qualitative analyses of the BNN and RE predictions as functions of selected features used for the network training of the topo-cluster response are collected in Figure 5. As already observed in Figures 4(b) and 4(c), the learned responses show some differences at the edges of the densely populated ridges when compared with the corresponding target distributions and among themselves, and evaluated as a function of $E_{\text{clus}}^{\text{EM}}$. This again is a display of *statistical dominance* of the distribution regions with higher population densities in the derivation of the predictions.

The complex distributions of $\mathcal{R}_{\text{clus}}^{\text{EM}}$ as a function of the cluster signal fraction in the electromagnetic calorimeter $f_{\text{emc}}$ shown in Figure 5(d) arise from the response characteristics and the calorimeter geometry.

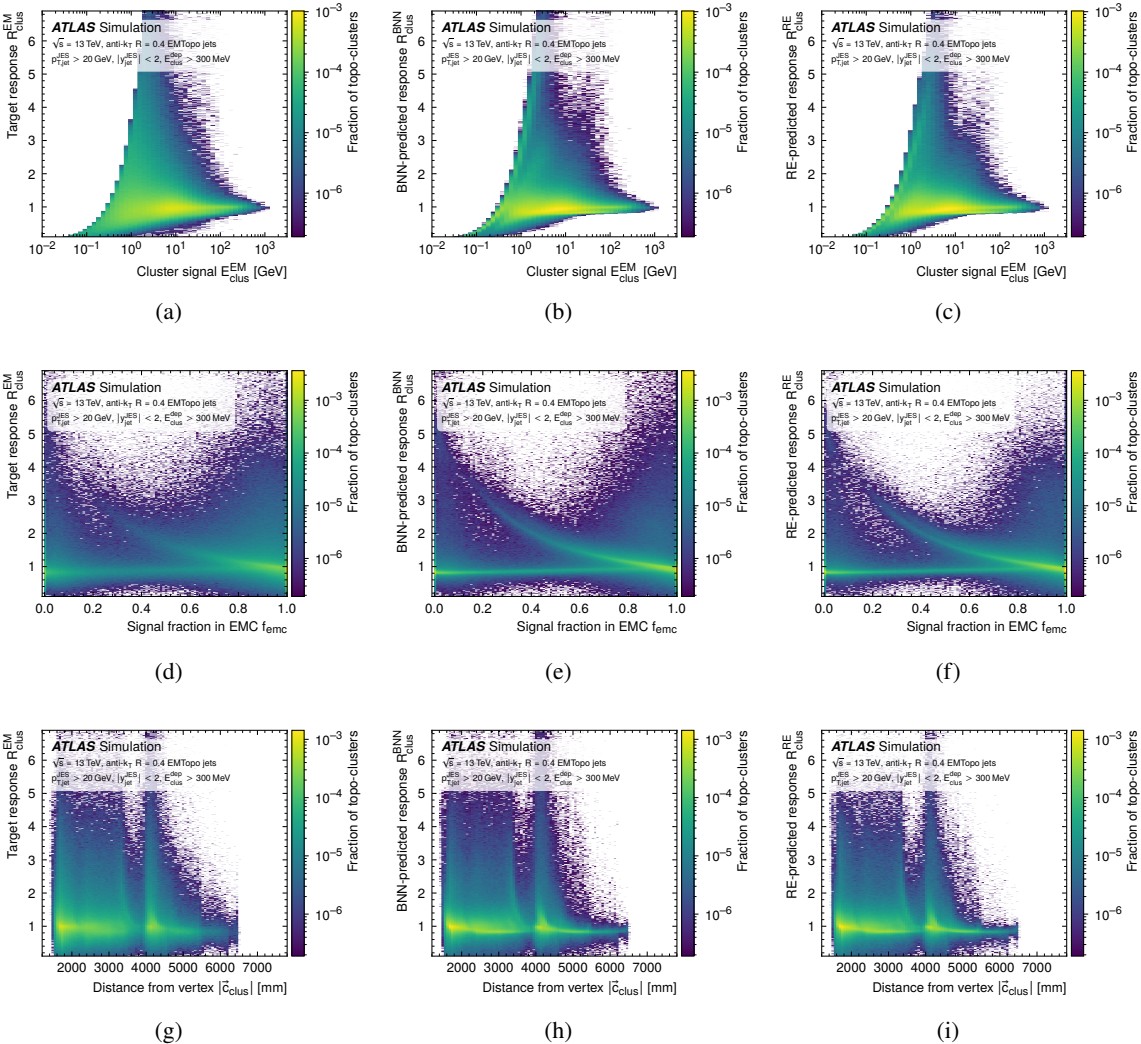

Figure 5: The distributions of the topo-cluster response $\mathcal{R}^{EM}_{clus}$ and the corresponding predictions from the BNN ($\mathcal{R}^{BNN}_{clus}$) and the RE ($\mathcal{R}^{RE}_{clus}$) are evaluated cluster-by-cluster as a function of selected features. The distributions of the training target $\mathcal{R}^{EM}_{clus}$ are shown in (a) as a function of the cluster signal $E^{EM}_{clus}$, in (d) as a function of the cluster signal fraction $f_{emc}$ in the electromagnetic calorimeter, and in (g) as a function of the distance of the topo-cluster centre-of-gravity $|\vec{c}_{clus}|$. The distributions of the predicted responses $\mathcal{R}^{BNN}_{clus}$ and $\mathcal{R}^{RE}_{clus}$ as functions of the same features are respectively shown in (b), (c) for $E^{EM}_{clus}$, in (e), (f) for $f_{emc}$, and in (h), (i) for $|\vec{c}_{clus}|$. The topo-clusters are extracted from MC simulations, as described in Section 3.1.1.

The near horizontal ridge extending from $\mathcal{R}^{EM}_{clus} \approx 0.8$ at $f_{emc} = 0$ to $\mathcal{R}^{EM}_{clus} \approx 1.0$ for $f_{emc} = 1$ shows the expected behaviour for a non-compensating calorimeter where a lower $f_{emc}$ is indicative to an increased contribution from hadronic showers in topo-clusters exclusively located in the hadronic calorimeter or extending from the electromagnetic to the hadronic calorimeter. The upper ridge populated by topo-clusters with rising $\mathcal{R}^{EM}_{clus}$ for decreasing $f_{emc}$ is likely introduced by $E^{EM}_{clus}$ at EM scale that overestimates energy deposits more and more dominated by ionisations in hadronic showers extending into the Tile calorimeter.

Figures 5(g)–(i) show the distributions of $\mathcal{R}^{EM}_{clus}$, $\mathcal{R}^{BNN}_{clus}$ and $\mathcal{R}^{RE}_{clus}$ as a function of the distance of the topo-cluster centre-of-gravity $|\vec{c}_{clus}|$ from the nominal vertex in ATLAS. The most prominent observed

structure around $|\vec{c}_{\text{clus}}| \approx 3900\,\text{mm}$ is introduced by the detector geometry. It shows the effect of the transition from the central barrel to the endcap calorimeters.

The general observation from the comparison of the $\mathcal{R}_{\text{clus}}^{\text{EM}}(f_{\text{emc}})$ distribution in Figures 5(d)–(f) and the $\mathcal{R}_{\text{clus}}^{\text{EM}}(|\vec{c}_{\text{clus}}|)$ distribution in Figures 5(g)–(i) is that the complex structures of the target distributions are well reproduced by both the BNN and the RE predictions, with some slightly sharpened edges around the ridges of higher topo-cluster populations for the predictions due to the statistical dominance of these regions in the learning.

### 4.2.2 Central prediction power

Results from a quantitative analysis of the prediction power of the BNN topo-cluster calibration model are collected in Figure 6. The metric used here is the relative difference $\Delta_{\mathcal{R}}$ between learned $\mathcal{R}_{\text{clus}}^{\text{BNN}}$ and the target $\mathcal{R}_{\text{clus}}^{\text{EM}}$, as defined in Eq. 15 in Section 4.1.1. It is measured cluster-by-cluster and analysed as functions of variables from MC simulations and the features in $\mathcal{X}_{\text{clus}}$, using two different measures of centrality for the resulting $\Delta_{\mathcal{R}}$ distributions. For example, Figure 6 shows the mean and median prediction power of the BNN as function of $E_{\text{clus}}^{\text{dep}}$, $E_{\text{clus}}^{\text{EM}}$, $f_{\text{emc}}$ and $|\vec{c}_{\text{clus}}|$ and compares those to the median and mean prediction power of the DNN [4]. The expectation from the particular (different) choices of loss functions for the BNN and the DNN training is that the mode $\langle\Delta_{\mathcal{R}}\rangle_{\text{mode}} \approx 0$ for all $\Delta_{\mathcal{R}}$ distributions in bins of features. It is found that $\langle\Delta_{\mathcal{R}}\rangle_{\text{med}} \approx \langle\Delta_{\mathcal{R}}\rangle_{\text{mode}}$ for both networks and all evaluation scales, with the advantage that $\langle\Delta_{\mathcal{R}}\rangle_{\text{med}}$ is well defined and does not require a particular algorithm or peak shape fit to be determined. The findings presented in Figure 6 show that the BNN has a similar if not stronger median prediction power than the DNN for all shown evaluation scales with the exception of $E_{\text{clus}}^{\text{EM}}$. The mean prediction power is, as expected from the choice of the loss functions for the BNN and the DNN, not as strong as the median for all evaluation scales.

### 4.2.3 Network training issues

About 3% of all topo-clusters collected from the simulated calorimeter jets were found to be resistant to the BNN, RE, and DNN training. These clusters are characterised by a target response $\mathcal{R}_{\text{clus}}^{\text{EM}} \gg 1$. This resistance potentially arises from a large pile-up contribution that is not accounted for in $E_{\text{clus}}^{\text{dep}}$ but increases $E_{\text{clus}}^{\text{EM}}$ significantly. As the affected topo-clusters are mostly observed in the hadronic calorimeters with considerably coarser readout geometry than the electromagnetic ones, this signal contribution from pile-up at the cluster location may not be reflected in the features due to a limited (spatial) resolution. The consequential lack of precision of the features can lead to a loss of expressiveness concerning the particular topo-cluster response and thus a severely reduced prediction power. In addition to the location, this interpretation is supported by the frequency of occurrence of these training-resistant topo-clusters as a function of pile-up, rising from about 0.2% of all clusters collected at $\mu \simeq 0$ (no pile-up) to close to 10% of all clusters collected at $\mu \simeq 90$ (highest observed pile-up level in the dataset).

A dedicated neural network was trained to tag the affected topo-clusters and remove them from the training and test datasets for the results presented in this paper. The ability to efficiently tag them, and to locate them in certain calorimeter regions under certain pile-up conditions, supports the potential application of dedicated calibrations and corrections beyond the scope of the study presented here. Further explorations are under way to address these issues.

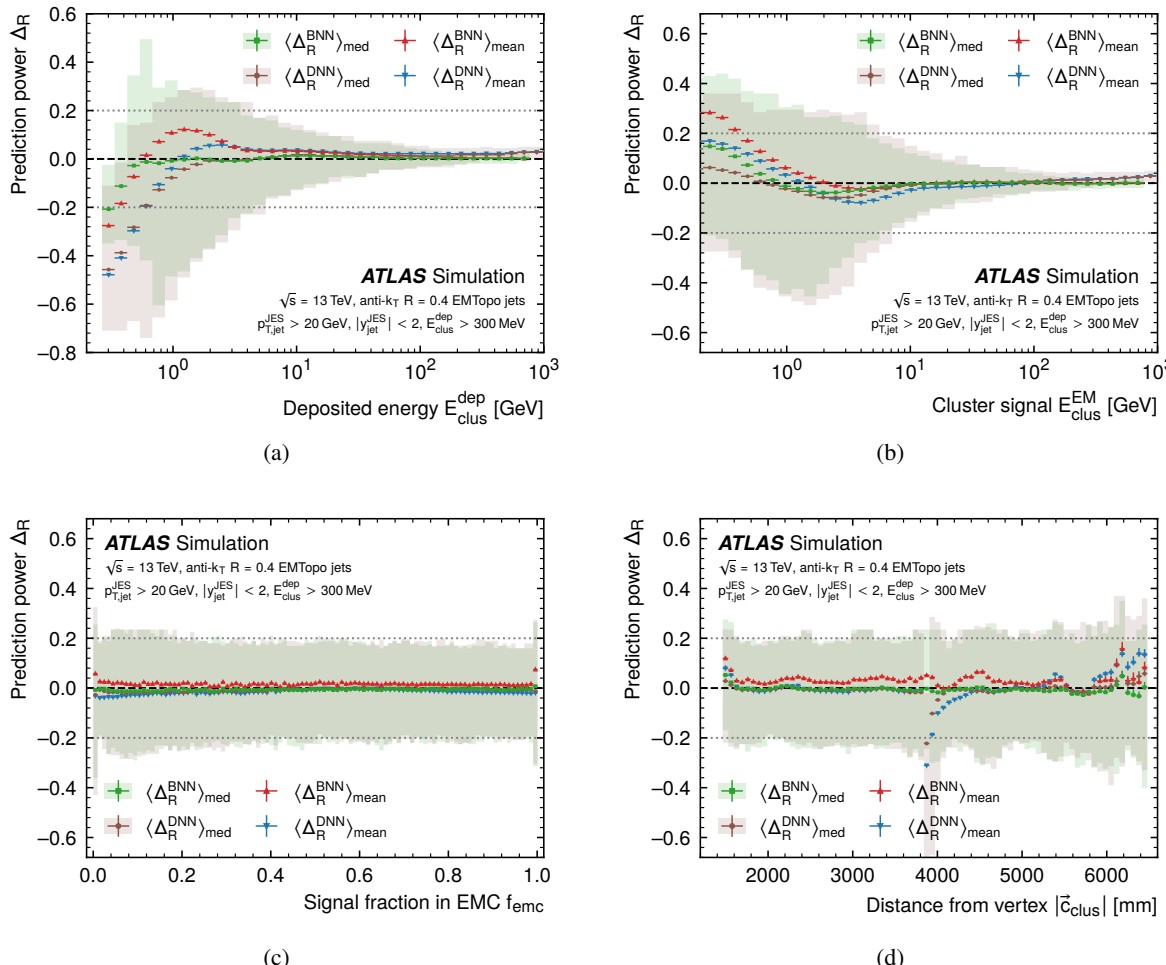

Figure 6: The mean and median prediction power of the trained BNN ($\Delta_{\mathcal{R}}^{\mathrm{BNN}}$) and the trained DNN ($\Delta_{\mathcal{R}}^{\mathrm{DNN}}$) are compared as a function of (a) the energy $E_{\mathrm{clus}}^{\mathrm{dep}}$ deposited in the topo-cluster, (b) the cluster signal $E_{\mathrm{clus}}^{\mathrm{EM}}$, (c) the cluster signal fraction in the electromagnetic calorimeter $f_{\mathrm{emc}}$, and (d) the cluster centre-of-gravity $|\vec{c}_{\mathrm{clus}}|$. The measures of centrality for the BNN are determined from the corresponding distributions shown in Figure 4(b) and in Figure 5(b), (e) and (h), while the ones for the DNN are extracted from Ref. [4]. The topo-clusters are collected from MC simulations, as described in Section 3.1.1. The shaded areas indicate the narrowest range of the respective $\Delta_{\mathcal{R}}$ distributions containing 68% of all entries. Statistical uncertainties are indicated by vertical bars if larger than the symbol size.

## 4.3 Central calibration performance

The prediction power discussed above is not a sufficient indicator for the accuracy and precision of the corresponding topo-cluster calibration. The deviation from signal linearity measure $\Delta_E$ introduced in Section 4.1.2 provides a direct scale to determine both, and thus the overall quality of the learned calibration. This measure can be constructed at various energy scales. For comparisons, the median deviation from linearity is evaluated for topo-clusters from the test dataset as a function of $E_{\mathrm{clus}}^{\mathrm{dep}}$ and selected features for the basic EM scale ($\Delta_E^{\mathrm{EM}}$), the LCW hadronic scale ($\Delta_E^{\mathrm{had}}$), the energy scale derived from the DNN calibration ($\Delta_E^{\mathrm{DNN}}$), and the scale after applying the BNN-learned calibration ($\Delta_E^{\mathrm{BNN}}$).

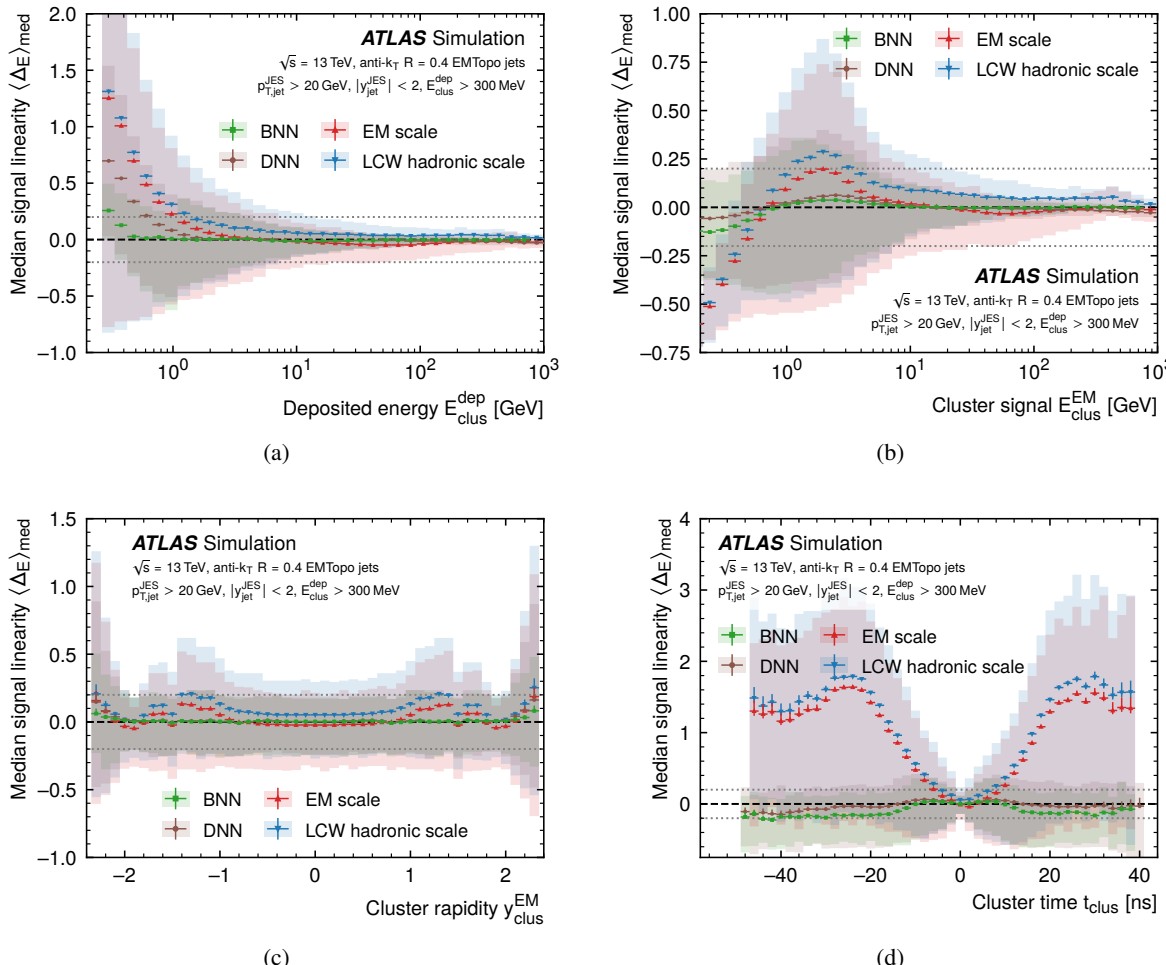

(a)

(b)

(c)

(d)

Figure 7: The median deviation from linearity $\langle\Delta_E\rangle_{\mathrm{med}}$ for topo-clusters at the basic EM, the LCW hadronic, the BNN-derived and the DNN-derived energy scales is shown as a function of (a) the energy $E_{\mathrm{clus}}^{\mathrm{dep}}$ deposited in the cluster, (b) the cluster signal $E_{\mathrm{clus}}^{\mathrm{EM}}$, (c) the cluster rapidity reconstructed at the basic EM scale $y_{\mathrm{clus}}^{\mathrm{EM}}$, and (d) the cluster signal time $t_{\mathrm{clus}}$. The topo-clusters are extracted from MC simulations according to the description in Section 3.1.1. The shaded areas indicate the narrowest range of the respective $\Delta_E$ distributions containing 68% of all entries. Statistical uncertainties are indicated by vertical bars if larger than the symbol size.

Figure 7 compares $\Delta_E$ on the various energy scales, as a function of selected topo-cluster features and $E_{\mathrm{clus}}^{\mathrm{dep}}$. As seen in Figure 7(a), the BNN-derived calibration extends the region of accurate energy reconstruction with $|\langle\Delta_E^{\mathrm{BNN}}\rangle_{\mathrm{med}}| \lesssim 2\%$ for $E_{\mathrm{clus}}^{\mathrm{dep}} \gtrsim 500\,\mathrm{MeV}$ down from the next-best calibration based on the DNN predictions, where a similar median deviation from linearity is reached for $E_{\mathrm{clus}}^{\mathrm{dep}} \gtrsim 1\,\mathrm{GeV}$. This is a significant gain of accuracy in the low energy regime.

When evaluated as a function of the topo-cluster signal $E_{\mathrm{clus}}^{\mathrm{EM}}$, the accuracy of both ML calibrations is significantly improved compared to the standard LCW-based hadronic calibration and the EM scale energy reconstruction, as is shown in Figure 7(b). The accuracy is not as high as observed in Figure 7(a), which is likely due to a residual, unmitigated and highly stochastic contribution to smaller signals that is mostly introduced by pile-up dominating $E_{\mathrm{clus}}^{\mathrm{EM}}$ in this region. Nevertheless, both machine-learning based

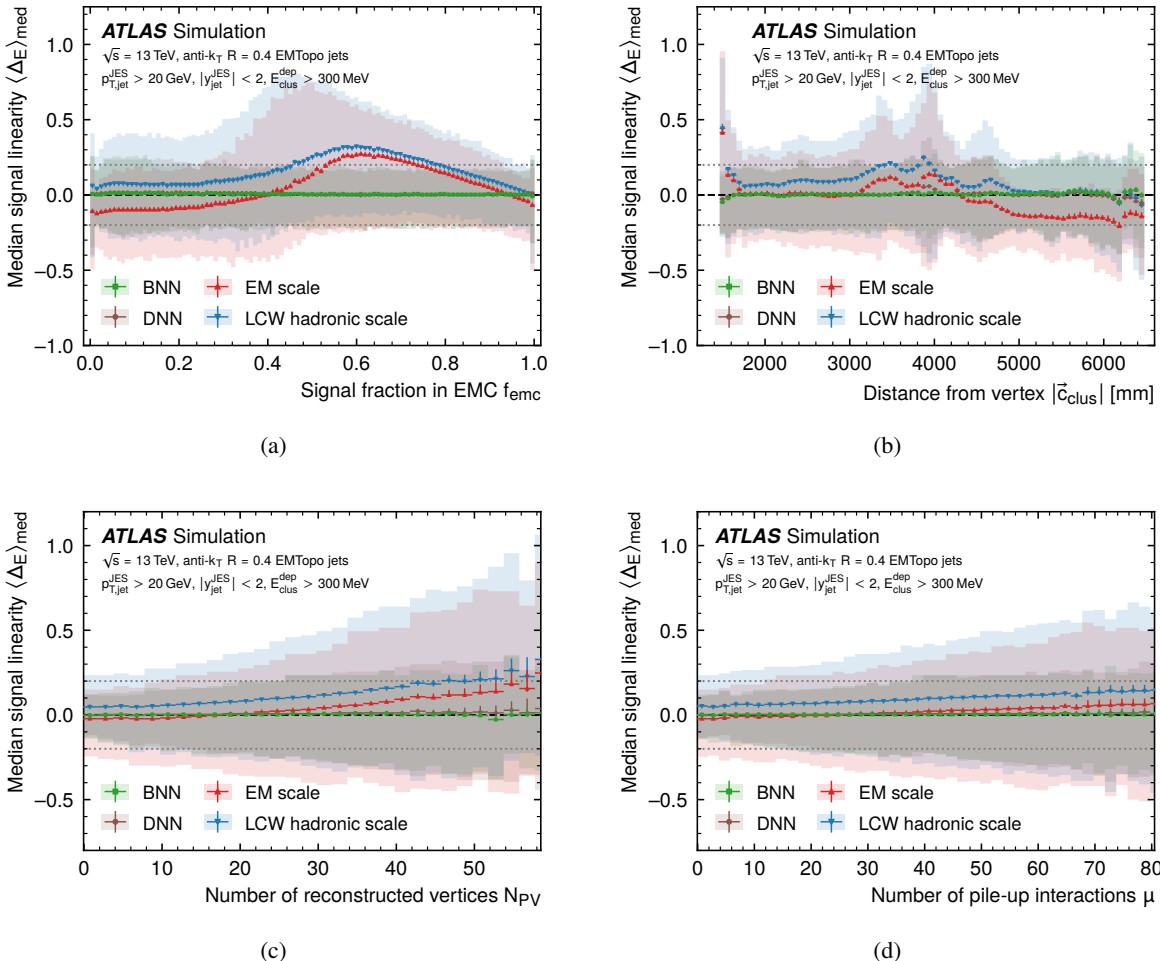

(a)

(b)

(c)

(d)

Figure 8: The median deviation from signal linearity $\langle \Delta_E \rangle_{\mathrm{med}}$ for topo-cluster signals at the basic EM, the LCW hadronic, the BNN-derived and the DNN-derived energy scales is shown as a function of (a) the cluster signal fraction in the electromagnetic calorimeter $f_{\mathrm{emc}}$, (b) the distance of the cluster centre-of-gravity from the nominal collision vertex $|\vec{c}_{\mathrm{clus}}|$, (c) the number of reconstructed primary vertices $N_{\mathrm{PV}}$ (in-time pile-up measure), and (d) the number of pile-up collisions in the bunch crossing $\mu$ (out-of-time pile-up measure). The results are achieved with topo-clusters from the test dataset, extracted as described in Section 3.1.1. The shaded areas indicate the narrowest range of the respective $\Delta_E$ distributions containing 68% of all entries. Statistical uncertainties are indicated by vertical bars if larger than the symbol size.

calibrations show significant less deviation from linearity than the standard LCW hadronic energy scale, which is derived from a calibration that uses $E_{\mathrm{clus}}^{\mathrm{EM}}$ as input as well.

Both $E_{\mathrm{clus}}^{\mathrm{BNN}}$ and $E_{\mathrm{clus}}^{\mathrm{DNN}}$ provide an accurate median measure of $E_{\mathrm{clus}}^{\mathrm{dep}}$ across the illuminated cluster rapidity $y_{\mathrm{clus}}^{\mathrm{EM}}$ space, as shown in Figure 7(c). Some increase of both $\langle \Delta_E^{\mathrm{BNN}} \rangle_{\mathrm{med}}$ and $\langle \Delta_E^{\mathrm{DNN}} \rangle_{\mathrm{med}}$ can be observed at the boundaries of this space at $|y_{\mathrm{clus}}^{\mathrm{EM}}| \approx 2.5$. The topo-clusters reconstructed at these edges can extend beyond the $|\eta_{\mathrm{det}}| = 2.5$ boundary, where the cell readout granularity of the ATLAS calorimeters is reduced by a factor of four. The increase in cell size potentially reduces the expressiveness of some features for these clusters. This leads to a loss of central accuracy, mainly due to a loss of sensitivity of the affected features that mitigate stochastic effects on $\mathcal{R}_{\mathrm{clus}}^{\mathrm{EM}}$. A larger deviation from linearity measured by $\langle \Delta_E^{\mathrm{had}} \rangle_{\mathrm{med}}$ at

the hadronic LCW scale, already observed in Figure 7(b) for $\langle\Delta_E^{\text{had}}\rangle_{\text{med}}(E_{\text{clus}}^{\text{EM}})$, is seen for $\langle\Delta_E^{\text{had}}\rangle_{\text{med}}(y_{\text{clus}}^{\text{EM}})$ in Figure 7(c) as well for $|y_{\text{clus}}^{\text{EM}}| \lesssim 2$, with maxima for topo-clusters located in the complex calorimeter transition region at $|\eta_{\text{det}}| \approx 1.5$. Both BNN and DNN learned calibrations measure $E_{\text{clus}}^{\text{dep}}$ in this transition region well, indicating that the feature expressiveness is sufficient for an accurate calibration.

Figure 7(d) shows the significant deviation from linearity for the basic topo-cluster signal $E_{\text{clus}}^{\text{EM}}$, and similarly for $E_{\text{clus}}^{\text{had}}$, for topo-cluster signal times $t_{\text{clus}}$ reconstructed outside of the bunch crossing time window of $-12.5\,\text{ns} < t_{\text{clus}} < 12.5\,\text{ns}$. In particular $\langle\Delta_E^{\text{EM}}\rangle_{\text{med}}(t_{\text{clus}}) > 1$ hints on the cluster response $\mathcal{R}_{\text{clus}}^{\text{EM}}$ being dominated by out-of-time pile-up signal contributions. This leads to a slightly increased $\langle\Delta_E^{\text{had}}\rangle_{\text{med}}(t_{\text{clus}}) > \langle\Delta_E^{\text{EM}}\rangle_{\text{med}}(t_{\text{clus}})$ for $|t_{\text{clus}}| \gtrsim 12.5\,\text{ns}$, as the hadronic calibration from LCW is agnostic to $t_{\text{clus}}$.

Both BNN and DNN trained calibrations learn $\mathcal{R}_{\text{clus}}^{\text{EM}}(t_{\text{clus}})$ well, which leads to $\langle\Delta_E^{\text{BNN}}\rangle_{\text{med}}(|t_{\text{clus}}| < 12.5\,\text{ns}) \approx 0$ and $\langle\Delta_E^{\text{DNN}}\rangle_{\text{med}}(|t_{\text{clus}}| < 12.5\,\text{ns}) \approx 0$. Outside of this time window, the BNN-derived calibration shows a near-constant median deviation from linearity by underestimating $E_{\text{clus}}^{\text{dep}}$ by about 20%. The DNN-derived calibration shows a better performance for $t_{\text{clus}} > 12.5\,\text{ns}$, with a deviation from linearity within a few percent, indicating that the signal contributions from the following bunch crossings are mitigated well.

Most of the features in $\mathcal{X}_{\text{clus}}$ are not involved in the reconstruction of both $E_{\text{clus}}^{\text{EM}}$ and $E_{\text{clus}}^{\text{had}}$, at least not directly as inputs. Extending the dimensionality of the local topo-cluster calibration, which is well supported by the machine learning applications, allows inclusion of information reflecting the cluster source and the $pp$ collision environment. In addition to the results presented in Figure 7, the devation from linearity is evaluated cluster-by-cluster for each feature in the full feature space represented in the test dataset. Figure 8 shows a selection of results for some of the features that both the EM scale and the hadronic scale from LCW are agnostic to. The results for the remaining features are collected in Appendix E.1.

A median calibration accuracy of better than 2% is achieved with both the BNN and the DNN nearly everywhere, when evaluated over the whole $f_{\text{emc}}$, $|\vec{c}_{\text{clus}}|$, $N_{\text{PV}}$ and $\mu$ value ranges. This indicates that both networks learn the response well from these features, even if those features are distributed in a complex way, as seen for $f_{\text{emc}}$ and $|\vec{c}_{\text{clus}}|$ in Figure 6(c) and 6(d), for example. In addition, the presence of a significant pile-up activity that increases the stochasticity of the topo-cluster signal and its features is well mitigated when the corresponding pile-up measures $N_{\text{PV}}$ and $\mu$ are included in the (ML-learned) calibration.

## 4.4 Local topo-cluster energy resolution

The relative local energy resolution of topo-clusters is evaluated at the basic EM, the hadronic scale from LCW, and for the BNN- and DNN-derived calibrations. It is measured using $\sigma_{\text{rel}}^E(E_{\text{clus}}^{\kappa}/E_{\text{clus}}^{\text{dep}})$ introduced in Eq. 17 in Section 4.1.3, where $\kappa$ names the energy scale.

Figure 9 shows $\sigma_{\text{rel}}^E(E_{\text{clus}}^{\kappa}/E_{\text{clus}}^{\text{dep}})$ as a function of $E_{\text{clus}}^{\text{dep}}$, $E_{\text{clus}}^{\text{EM}}$, and the pile-up measures $N_{\text{PV}}$ and $\mu$. Both ML calibrations yield a significantly better relative resolution than the standard local hadronic calibration, together with a significant improvement relative to the EM scale resolution, on all evaluation scales. This is expected from the larger number of features entering the BNN and the DNN and the resulting increased exposure of these calibrations to more information concerning the topo-cluster signal source, in combination with the exploitation of feature correlations lost for other energy scale calibrations, and the learned smooth calibration functions.

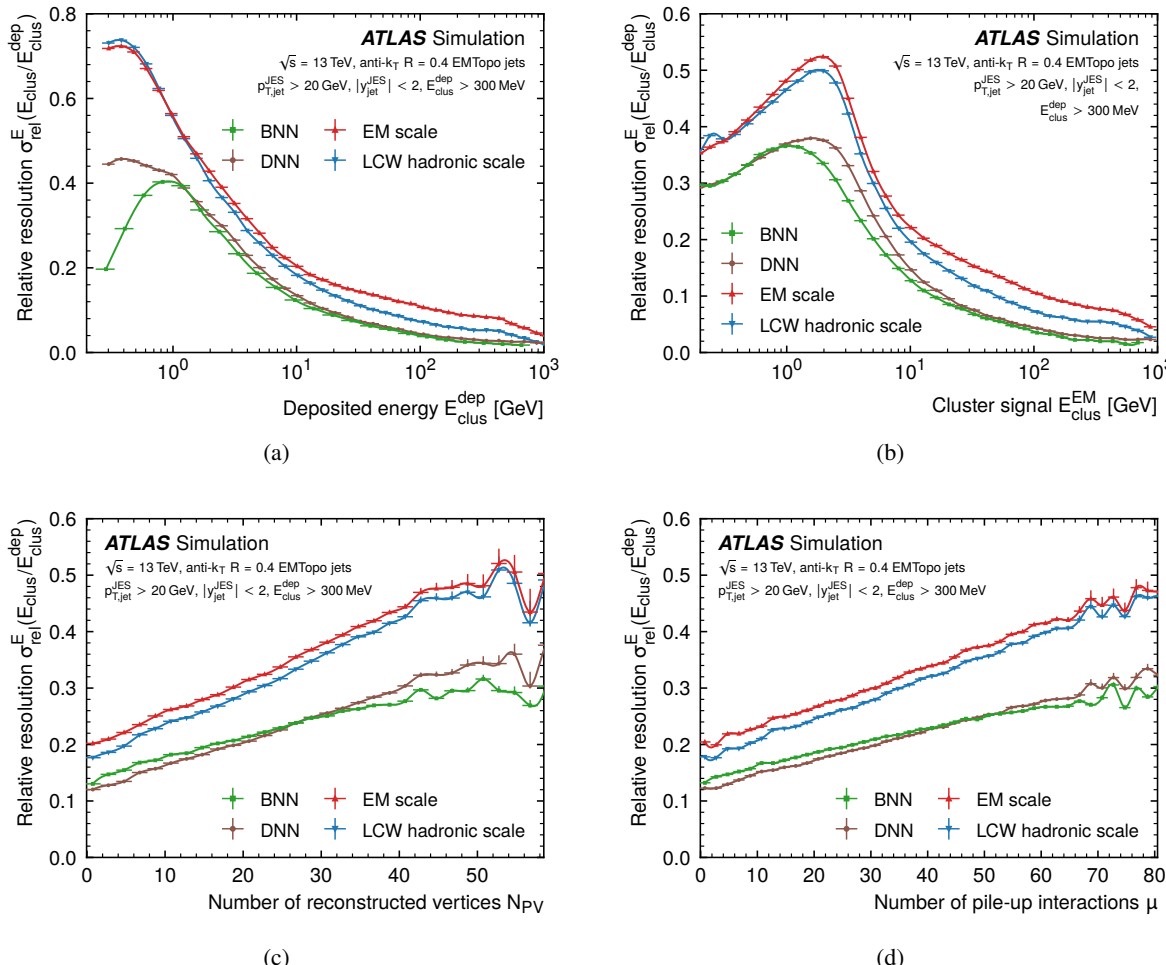

(a)

(b)

(c)

(d)

Figure 9: The relative local energy resolution of topo-clusters at the basic EM, the LCW hadronic, the BNN-derived and the DNN-derived energy scales, measured using Eq. 17, as a function of (a) the energy deposited in the cluster $E_{clus}^{dep}$, (b) the basic cluster signal $E_{clus}^{EM}$, (c) the number of reconstructed vertices $N_{PV}$, and (d) the number of pile-up collisions $\mu$ in the bunch crossing. The topo-clusters are extracted from the test dataset as discussed in Section 3.1.1. Statistical uncertainties are indicated by vertical bars if larger than the symbol size.

The steeper downturn of the calibrated signal fluctuations for the BNN-derived calibration observed in Figure 9(a) for $E_{clus}^{dep} \lesssim 1$ GeV indicates that this model best learns the transition of the signal source from inelastic hadronic interactions to more ionisation-dominated signals.[12] This is likely associated with the use a Gaussian mixture model for $p(D_{train}|\theta)$ in the BNN loss function, as described in Eq. 12 of Section 3.3.2, among other possible differences between the BNN and the DNN. The comparisons of the individual shapes of the $\Delta_E^{BNN}$ and the $\Delta_E^{DNN}$ distributions in bins of $E_{clus}^{dep}$ show significantly suppressed tails in case of the BNN-derived calibrations. This is expected for signals from ionisation losses, which are subject to considerably less intrinsic fluctuations than signals from energy loss mechanisms occurring in inelastic

---

[12] Hadrons can traverse relatively large distances in matter, if their energy is below the typical threshold of about 1 GeV for inelastic interactions, according to Ref. [37]. This suggests that energy deposits in topo-clusters within $0 < E_{clus}^{dep} \lesssim 1$ GeV reflect a significant contribution to $E_{clus}^{dep}$ from energy loss mechanisms not involving hadronic inelastic interactions, like ionisations by charged particles.

processes. A representative selection of these comparisons is shown in Figure 19 in Appendix E.2.

The relative topo-cluster energy resolution after application of the ML-based calibrations is increased, compared to the corresponding resolutions obtained for the standard energy scales, when analysed as a function of the in-time pile-up measure $N_{\mathrm{PV}}$ in Figure 9(c), and as a function of the out-of-time pile-up measure $\mu$ in Figure 9(d). The increased but still partial cluster-by-cluster pile-up mitigation is manifested by the reduced relative signal fluctuations observed at any given $N_{\mathrm{PV}}$ and $\mu$ when compared with the other energy scales. The fluctuations after applying the BNN-derived topo-cluster calibration show a more flat dependence on $N_{\mathrm{PV}}$ than the residual fluctuations after application of the DNN-based calibration, with a slightly worse performance at low $N_{\mathrm{PV}}$ (low pile-up) but a better performance at higher $N_{\mathrm{PV}}$ (higher pile-up). A similar, but less pronounced behaviour is observed in Figure 9(d), where the relative energy resolutions achieved with the BNN- and DNN-derived calibrations are shown as a function of $\mu$.

# 5 Analyzing the uncertainty predictions

The uncertainty predictions from both the BNN and the RE are based on predictions sampled from a model representing an ensemble of trained networks. In case of the BNN, this ensemble is formed by sampling the network parameters at inference from variational distributions with learned parameters, as schematically laid out in Figure 2 in Section 3.3.1. For the RE, the ensemble is a collection of networks interconnected by a repulsive term during training, as discussed in Section 3.3.3.

## 5.1 Predictive uncertainties

Learned uncertainties are usually referred to as *predictive uncertainties* [10, 38, 39]. They are divided into two major categories,

1. *epistemic* or *model uncertainties* that are related to how machines can learn and what cannot be learned, and

2. *aleatoric* or *data uncertainties* that are related to the residual level of stochasticity in the data that cannot be mitigated any further by the model.

The expectation is that the epistemic uncertainties, after the final network optimisation for the topo-cluster calibration, are dominated by contributions inherent to the data that limit what the BNN can learn. In particular, contributions to these uncertainties are possible from a statistically insufficient coverage of the whole feature space, or certain regions of it, by topo-clusters in the training dataset. These contributions are thus expected to largely vanish in the limit of infinite training data and a sufficient number of epochs. As said above, the aleatoric uncertainties reflect the lack of expressiveness of the data introduced by a relatively high level of noise predominantly introduced by pile-up that cannot be overcome by increasing the training dataset or improving the network model.

The predictive uncertainty of a trained network model is a measure for its lack of expressive power arising from either of the two categories of uncertainties introduced above. This ambiguity can be resolved by relating the learned uncertainties to the standard categories of statistical and systematic uncertainties that are usually associated with the application of a calibration model, or with a measurement in general. The chosen convention here follows the general expectation that the statistical uncertainty vanishes in the limit of infinite training data, and the systematic uncertainty remains finite at this limit. When evaluating

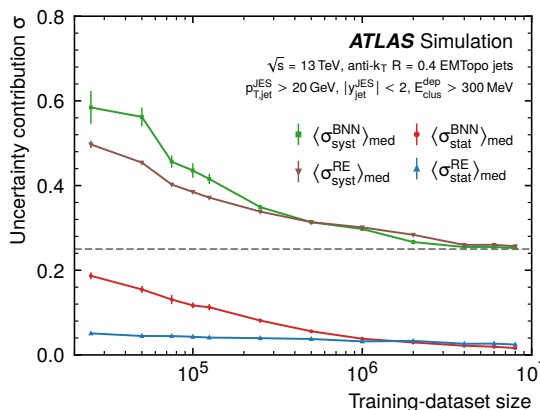

Figure 10: The dependence of the predicted uncertainty components on the training dataset size. Shown are the median uncertainty components discussed in Section 5 for training datasets of varying sizes and their specific asymptotic behaviours. The topo-clusters used here are collected into training datasets employing the selections shown in Section 3.1.1. The statistical uncertainties on the medians are indicated as vertical bars when exceeding the symbol size. The dashed horizontal line indicates the asymptotic limit of the systematic uncertainties.

the quality of the network, both the statistical and the systematic uncertainty contribute to the learned uncertainty as a systematic component in the sense that evaluating the network many times, without changing the training dataset size, does not reduce them. They are thus typically considered as external nuisance parameters associated with the specific network configuration and training conditions.

## 5.2 Uncertainty components

Separately analysing the statistical ($\sigma_{\text{stat}}$) and systematic ($\sigma_{\text{syst}}$) component of the total learned uncertainty $\sigma_{\text{tot}}$ requires a sufficiently large training dataset. It is possible for both the BNN and the RE network employed here, as they use the same large training dataset introduced in Table 3 of Section 3.3. This means that

$$\sigma_{\text{tot}} = \sqrt{\sigma_{\text{syst}}^2 + \sigma_{\text{stat}}^2} \; . \tag{18}$$

Following the previous discussion, $\sigma_{\text{syst}}$ and $\sigma_{\text{stat}}$ are characterised by their behaviour in the limit of an infinite training dataset size $N_{\text{train}}$,

$$N_{\text{train}} \rightarrow \infty \;\; \Rightarrow \;\; \begin{cases} \sigma_{\text{syst}} \rightarrow \text{const.} > 0 \\ \sigma_{\text{stat}} \rightarrow 0 \end{cases} \; .$$

With this understanding, $\sigma_{\text{syst}}$ represents the limit of the accuracy to which the model learns to describe the target with the training data available. In case of a non-deterministic or stochastic target (noisy data) [12], like $\mathcal{R}_{\text{clus}}^{\text{EM}}$, limited expressiveness of the network (structure uncertainty) [13], and non-optimal choices of hyper-parameters (uncertainty in model parameters) can all be interpreted in the spirit of contributions to the systematic uncertainty inherent to the calibration model and the training data. It has contributions to both epistemic and aleatoric uncertainty components.

The statistical uncertainty component $\sigma_{\text{stat}}$ is constructed to become insignificantly small with increasing $N_{\text{train}}$, and is thus expected to become irrelevant for a sufficiently large training dataset. Therefore, $\sigma_{\text{stat}}$ can

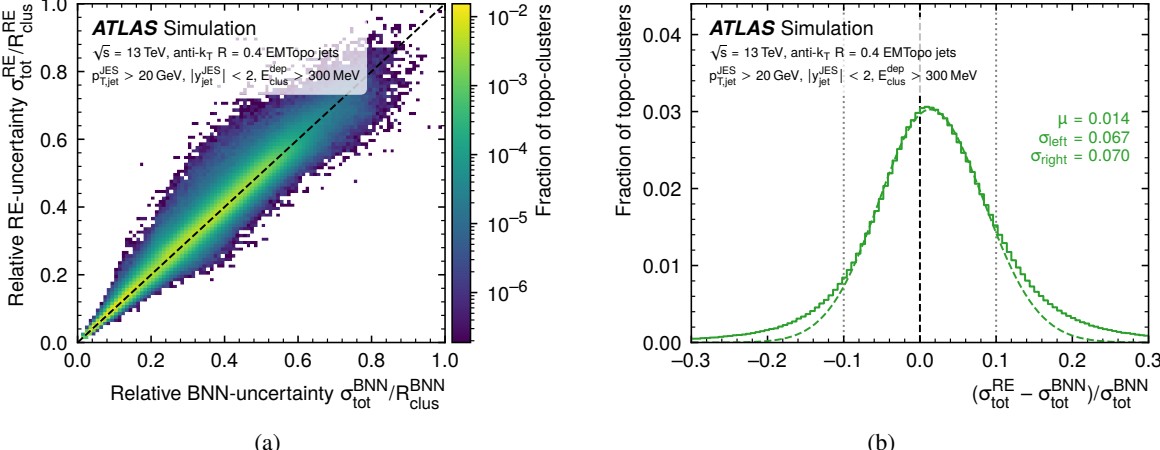

(a)                                                                    (b)

Figure 11: In (a), the relative total uncertainty prediction $\sigma_{\text{tot}}^{\text{RE}}/\mathcal{R}_{\text{clus}}^{\text{RE}}$ from the RE is shown as a function of the relative uncertainty $\sigma_{\text{tot}}^{\text{BNN}}/\mathcal{R}_{\text{clus}}^{\text{BNN}}$ from the BNN. The distribution of the relative difference between the two uncertainties $(\sigma_{\text{tot}}^{\text{RE}} - \sigma_{\text{tot}}^{\text{BNN}})/\sigma_{\text{tot}}^{\text{BNN}}$ is shown in (b). For illustration, the dashed (centre) line indicates the completely unbiased mean of the distribution, and the dotted lines indicate the 10% range around it. The dashed curve shows the result of a two-sided fit of Gaussian functions to the distribution, with a common mean $\mu$ and the independently fitted widths $\sigma_{\text{left}}$ and $\sigma_{\text{right}}$. Both uncertainties are predicted for each topo-cluster in the test dataset. These clusters are collected from the simulated jets, as described in Section 3.1.1.

be considered as a significant part of the epistemic uncertainties. A brief and more formal introduction to the calculation of $\sigma_{\text{syst}}$ and $\sigma_{\text{stat}}$ can be found in Appendix D.2.

The asymptotic behaviours of the uncertainty components as a function of the training dataset size are summarised in Figure 10. The figure shows the two BNN components ($\sigma_{\text{syst}}^{\text{BNN}}$, $\sigma_{\text{stat}}^{\text{BNN}}$) and the two RE ($\sigma_{\text{syst}}^{\text{RE}}$, $\sigma_{\text{stat}}^{\text{RE}}$) components separately. The results discussed in the following were achieved with a training dataset where $N_{\text{train}}$ was sufficiently large to assume $\sigma_{\text{syst}} \gg \sigma_{\text{stat}} \Rightarrow \sigma_{\text{tot}} = \sigma_{\text{syst}}$.

## 5.3 Comparison of the learned uncertainties

The uncertainty predictions provided along with the learned topo-cluster calibration need to be analysed and interpreted to rule out that they are artefacts introduced by the machine learning model itself. Rather, they are supposed to reflect limitations in the data that lower the ability of the applied network model to accurately regress on the target, in addition to limitations introduced by the model itself.

To address these concerns about specific network-introduced artefacts, the individual uncertainty predictions from the two different network models BNN and RE are directly compared cluster-by-cluster. Figure 11(a) shows the correlation between the relative learned uncertainties of the BNN ($\sigma_{\text{tot}}^{\text{BNN}}/\mathcal{R}_{\text{clus}}^{\text{BNN}}$) and the RE ($\sigma_{\text{tot}}^{\text{RE}}/\mathcal{R}_{\text{clus}}^{\text{RE}}$). As by construction the systematic uncertainties are not deterministic, a certain spread can be expected and is observed. Nevertheless, there is no sign of a bias in this correlation, the two predictions track each other well.

Figure 11(b) shows the distribution of the relative differences $(\sigma_{\text{tot}}^{\text{RE}} - \sigma_{\text{tot}}^{\text{BNN}})/\sigma_{\text{tot}}^{\text{BNN}}$ between the (absolute) BNN and the RE uncertainty predictions. A two-sided fit of Gaussian functions yields a near symmetric shape, indicating an agreement between the two estimates at the level of 10%. Differences to the fitted

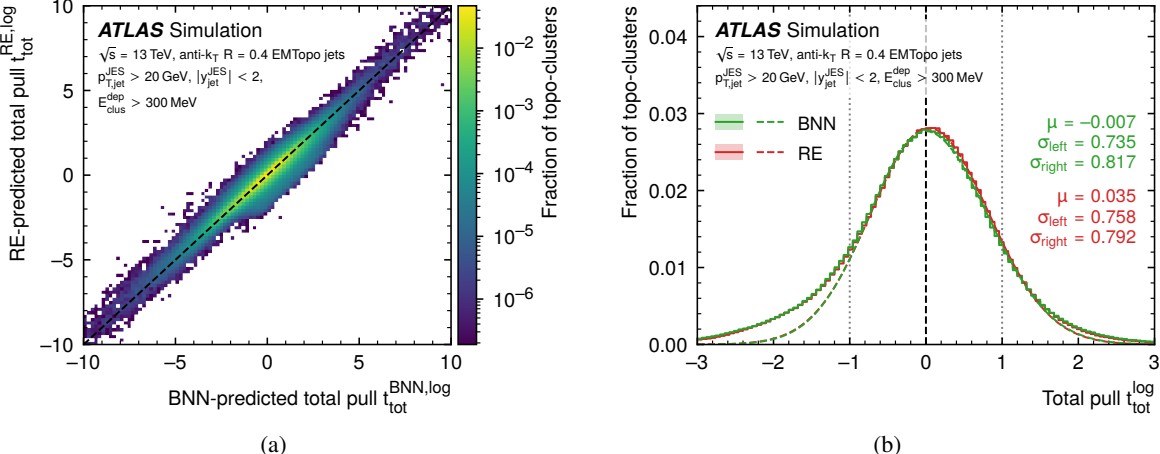

(a)  (b)

Figure 12: The observed pull from the RE training ($t_{\text{tot}}^{\text{RE,log}}$) is shown as a function of the one from the BNN training ($t_{\text{tot}}^{\text{BNN,log}}$) in (a). The inclusive distributions of $t_{\text{tot}}^{\text{BNN,log}}$ and $t_{\text{tot}}^{\text{RE,log}}$ are compare in (b). Both $t_{\text{tot}}^{\text{BNN,log}}$ and $t_{\text{tot}}^{\text{RE,log}}$ are calculated using Eq. 20, for a test dataset containing topo-clusters collected from full MC simulations, as described in Section 3.1.1. The dashed vertical line indicates the unbiased centre of the $t_{\text{tot}}^{\text{log}}$ distributions, while the dotted lines show the ±1 range around it. The dashed curves indicate the two-sided fits of Gaussian functions for each network, with the common average $\mu$ and the independently fitted widths $\sigma_{\text{left}}$ and $\sigma_{\text{right}}$. The corresponding fitted quantities are quoted.

Gaussian functions in the tails between 10% and 30%, and −30% and −10%, are fairly identical and thus unbiased. They are expected for highly stochastic data like the response and the reconstructed features from topo-clusters with significant near-random contributions from the high levels of pile-up experienced in ATLAS at LHC Run 2 operations. This high level of comparability between the learned uncertainties shows that those predominately reflect features of the data, and not artefacts introduced by either network.

## 5.4 Uncertainties and data spread

The interpretation from the previous section are augmented with the analysis of the predicted value of the uncertainty for any given topo-cluster. Such an analysis can be done using the *pull t*,

$$t(\mathcal{X}_{\text{clus}}) = \frac{\mathcal{R}_{\text{clus}}^{\text{BNN(RE)}}(\mathcal{X}_{\text{clus}}) - \mathcal{R}_{\text{clus}}^{\text{EM}}}{\sigma_{\text{tot}}^{\text{BNN(RE)}}(\mathcal{X}_{\text{clus}})} , \tag{19}$$

that measures the difference between the prediction $\mathcal{R}_{\text{clus}}^{\text{BNN(RE)}}$ from the BNN or RE and the target $\mathcal{R}_{\text{clus}}^{\text{EM}}$ on a scale defined by the respective predicted uncertainty $\sigma_{\text{tot}}^{\text{BNN(RE)}}$. Considering that the training produces a prediction for $\log_{10} \mathcal{R}_{\text{clus}}^{\text{BNN(RE)}}$ with a target of $\log_{10} \mathcal{R}_{\text{clus}}^{\text{EM}}$, the pull $t_{\text{tot}}^{\text{log}}$ is compared cluster-by-cluster,

$$t_{\text{tot}}^{\text{log}}(\mathcal{X}_{\text{clus}}) = \frac{\mathcal{R}_{\text{clus}}^{\text{BNN(RE)}}(\mathcal{X}_{\text{clus}})}{\sigma_{\text{tot}}^{\text{BNN(RE)}}(\mathcal{X}_{\text{clus}})} \cdot \ln 10 \cdot \left( \log_{10} \mathcal{R}_{\text{clus}}^{\text{BNN(RE)}}(\mathcal{X}_{\text{clus}}) - \log_{10} \mathcal{R}_{\text{clus}}^{\text{EM}} \right) . \tag{20}$$

Figure 12 compares $t_{\text{tot}}^{\text{log}}$ for the topo-cluster response predictions from the BNN ($t_{\text{tot}}^{\text{BNN,log}}$) with the ones from the RE ($t_{\text{tot}}^{\text{RE,log}}$). Both $t_{\text{tot}}^{\text{BNN,log}}$ and $t_{\text{tot}}^{\text{RE,log}}$ are highly correlated, as shown in Figure 12(a). This

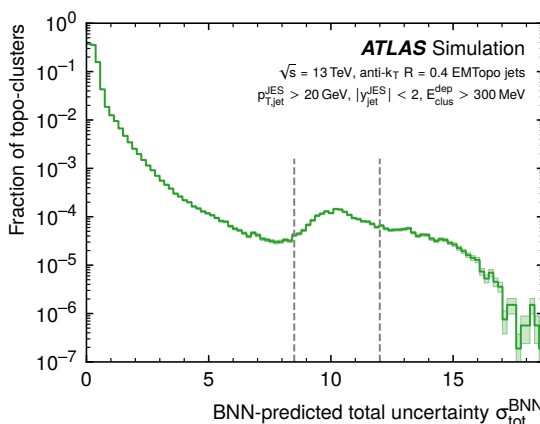

Figure 13: The inclusive spectrum of uncertainties $\sigma_{\text{tot}}^{\text{BNN}}$ learned by the BNN. The vertical lines indicate the region $8.5 < \sigma_{\text{tot}}^{\text{BNN}} < 12.0$ that is selected to analyse the source of the deviation from the expected smoothly falling spectrum. The uncertainties are predicted for topo-clusters from test dataset collected from the full MC simulations as described in Section 3.1.1.

indicates that both networks provide near identical pulls cluster-by-cluster, and confirms that $\mathcal{R}_{\text{clus}}^{\text{BNN}} \approx \mathcal{R}_{\text{clus}}^{\text{RE}}$ for $\sigma_{\text{tot}}^{\text{RE}}/\mathcal{R}_{\text{clus}}^{\text{RE}} \approx \sigma_{\text{tot}}^{\text{BNN}}/\mathcal{R}_{\text{clus}}^{\text{BNN}}$ found in the previous Section 5.3.

The pull distribution is expected to be close to a Gaussian distribution, with a central value $\langle t_{\text{tot}}^{\text{log}} \rangle = 0$ and a spread measured by its standard deviation of the order of unity, as motivated by the discussion in Appendix D.3. Figure 12(b) shows that both the $t_{\text{tot}}^{\text{BNN,log}}$ and the $t_{\text{tot}}^{\text{RE,log}}$ distributions follow Gaussian distributions in their bulk parts, with close to symmetric spreads determined by independent two-sided fits of $\sigma_{\text{left}}$ and $\sigma_{\text{right}}$ to each of these distributions. The asymmetry observed between the tails $t_{\text{tot}}^{\text{log}} < \sigma_{\text{left}}$, where some deviations from a Gaussian distribution are observed, and $t_{\text{tot}}^{\text{log}} > \sigma_{\text{right}}$, where the distribution more closely follows a Gaussian distribution, is an artefact introduced by the logarithmic transformation of the target and the respective predictions. The fitted spreads yield $\sigma_{\text{left}} \approx \sigma_{\text{right}} < 1$, indicating that both $\sigma_{\text{tot}}^{\text{BNN}}$ and $\sigma_{\text{tot}}^{\text{RE}}$ represent a conservative uncertainty estimate safely covering the residual difference between target and prediction for each topo-cluster. Considering these spreads together with central values fitted to be close to $t_{\text{tot}}^{\text{log}} = 0$, the conclusion from the pull distributions is that both $\sigma_{\text{tot}}^{\text{BNN}}$ and $\sigma_{\text{tot}}^{\text{RE}}$ are fairly unbiased and conservative uncertainties within expected value ranges.

## 5.5 Sources of uncertainties

The inclusive spectrum of $\sigma_{\text{tot}}^{\text{BNN}}$ learned by the BNN displayed in Figure 13 shows a rise of the population of topo-clusters with large uncertainties, deviating from the expected smoothly falling spectrum. The topo-clusters in the region $8.5 < \sigma_{\text{tot}}^{\text{BNN}} < 12.0$ around this emerging peak are investigated to search for possible reasons for its appearance. They show a large response $\mathcal{R}_{\text{clus}}^{\text{EM}} \gg 1$ and a small signal fraction in the electromagnetic calorimeter. Their centre-of-gravity is mostly located just in front of the endcap calorimeters, and within $1 \lesssim |y_{\text{clus}}^{\text{EM}}| \lesssim 1.4$. This points to the instrumentation of a specific transition range between ATLAS calorimeters generating the topo-cluster signals.

Figure 14(a) shows the topo-cluster centre-of-gravity projected into the $(|z_{\text{clus}}|, r_{\text{clus}})$ plane, where $z_{\text{clus}}$ is its Cartesian coordinate along the central axis of the ATLAS detector and $r_{\text{clus}}$ is its radial distance from this central axis, for all clusters in the test dataset. Selecting topo-clusters with $8.5 < \sigma_{\text{tot}}^{\text{BNN}} < 12.0$

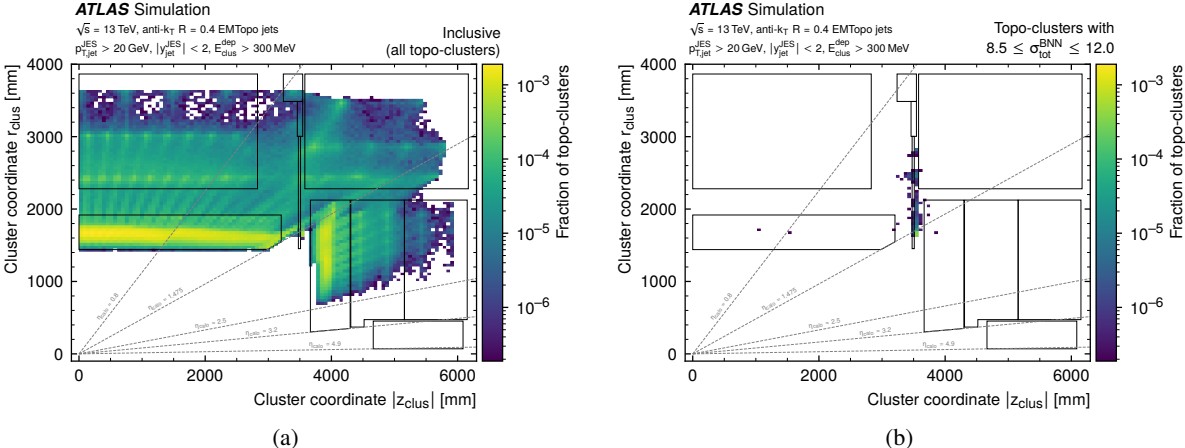

(a)                                                         (b)

Figure 14: In (a), the centre-of-gravity location projected into the $(|z_{clus}|, r_{clus})$ plane is shown for all topo-clusters in the test dataset. The outlined structures indicate the instrumented area covered by the various ATLAS calorimeters in this plane. The lines depict selected pseudorapidities $\eta_{calo}$ in the calorimeter pointing to transition regions. The lack of topo-clusters located beyond $|\eta_{calo}| \gtrsim 2.4$ is introduced by the jet selection described in Section 3.1.1. The location of the centre-of-gravity of topo-clusters with $8.5 < \sigma_{tot}^{BNN} < 12.0$ found in the same plane is shown in (b). These are predominantly inside, or in close proximity to, the thin Tile gap scintillators installed in the transition region between the central and the endcap calorimeters.

from this sample yields the $(|z_{clus}|, r_{clus})$-plane population depicted in Figure 14(b). The response and the features of these clusters are predominantly derived from signals in the Tile gap scintillator instrumenting the transition region between the central barrel and the endcap calorimeters. This detector is not a regular calorimeter in that it does not generate a signal corresponding to the energy loss generated in a (regular) sampling structure of active and inactive media in a calorimeter cell. Rather, it generates a signal reflecting the energy losses in the complex distribution of inactive material situated around it. This environment comprises cryostat walls, mechanical support structures and inner detector services. The energy deposits in these inactive materials are not considered in $\mathcal{R}_{clus}^{EM}$ defined in Eq. 4 of Section 3.2 for the topo-clusters located here, because they occur outside of calorimeter cells and are thus not accounted for in $E_{clus}^{dep}$.

The actual signal $E_{clus}^{EM}$ of topo-clusters located in the Tile gap scintillator is predominantly generated by ionisations. Among the sources of these ionisations are secondary particles produced in the development of showers in the material around it. These showers are generated by particles emerging from the hard scatter and the in-time pile-up interactions. The latter can also provide a second and more direct contribution to the signal when directly crossing this transition region, without previous interaction. The BNN calibration network learns $\mathcal{R}_{clus}^{EM}$ from the feature set here,[13] but the location of the topo-cluster centre-of-gravity suggests very little signal contributions from other calorimeters to it. In particular, features related to shower shapes are expected to be reconstructed with low precision, leading to a significantly reduced expressiveness that produces predictions with large uncertainties for topo-clusters in this region.

The fact that the BNN learns about these regional detector issues from the training data and predicts a large but appropriate and traceable total uncertainty $\sigma_{tot}^{BNN}$, suggests that a meaningful interpretation of $\sigma_{tot}^{BNN}$

---

[13] To a large extent, $E_{clus}^{dep}$ in these topo-clusters consists of the relatively small amounts of energy deposited by ionisations by charged particles traversing the thin scintillators, in addition to the energy deposited by very rare inelastic hadronic interactions in the scintillator volume.

cluster-by-cluster is possible in e.g., the context of topo-cluster signal accuracy and precision requirements in a given final-state analysis. Considering the previous discussions, $\sigma_{\text{tot}}^{\text{RE}}$ is expected to reflect the same detector-induced issues.

# 6 Conclusions

The principal signal of the ATLAS calorimeters are clusters of topologically connected cell signals (topo-clusters). They are individually calibrated to measure the energy deposited in them using an uncertainty-aware Bayesian neural network (BNN). This network is designed to predict the response and the corresponding uncertainty for a topo-cluster as a function of a set of 15 features associated with it. These features are selected under consideration of expressiveness manifested in their sensitivity to the signal source at the location of the cluster, their dependence on the detector-specific signal formation characteristics, and the proton–proton collision environment characterised by pile-up.

The datasets collected for the BNN training, validation and testing comprise topo-clusters collected from calorimeter jets in full and detailed Monte Carlo simulations of multijet production in proton–proton collisions at a centre-of-mass energy of $\sqrt{s} = 13$ TeV, with the ATLAS detector configurations in place at the time of the LHC Run 2 and with beam conditions generating levels of pile-up observed in this operational period (2015–2018).

The calibrated signal performance with respect to the deviation from signal linearity and the local relative energy resolution indicates significant improvements on most relevant evaluation scales, if compared to the two standard calibrations in ATLAS, the basic electromagnetic topo-cluster calibration and the table-look-up-based hadronic calibration implemented in the LCW sequence, and to the calibration learned by a deep neural network (DNN) employing the same features as inputs as the BNN. In particular, the application of the BNN calibration extends near-perfect signal linearity to a previously underperforming region populated with topo-clusters with low-energy deposits from particles emerging from the hard scatter interaction. This calibration also improves the relative energy resolution at cluster level when compared with the resolution achieved with the DNN-based calibration, which already shows a significant improvement with respect to the standard calibrations.

The uncertainties predicted by the BNN consist of two components that in the case of the sufficiently large training dataset used here can be distinguished and respectively interpreted as statistical and systematic uncertainties. The BNN-learned uncertainties are confirmed using an alternative and independent calibration network featuring repulsive ensembles (RE), which predicts uncertainties that are cluster-by-cluster identical within the level of stochasticity in the dataset. An analysis of the pull distributions finds that the uncertainties from both networks finds them to follow a Gaussian distribution in the bulk, with some small deviations observed in particular in the low pull tails. The central values are found to be close to zero, indicating an unbiased pull. The spread of each pull distribution is less than the expected unity, which suggests that the learned uncertainties are conservative.

Large but appropriate uncertainties of the calibrated energy derived from the BNN response predictions are observed for topo-clusters located in the insufficiently instrumented transition region between the central and the endcap calorimeter modules. This shows promise for a potential applications in which the uncertainty of the calibrated topo-cluster signals can be used as evaluation tool for the signal quality and thus allow for selections based on accuracy (scale) and precision (resolution) requirements before the

clusters enter the reconstruction of hadronic final-state objects like particles and jets, and the reconstruction of hadronic event shapes in general.

This study of the feasibility of the use of uncertainty-aware neural networks for the ATLAS calorimeter signal calibration in a realistically simulated proton–proton collision environment shows significant potential for a successful application in the context of physics analyses. The simulation-only results presented in this paper require investigations beyond these proof-of-principle studies, including the validation and performance evaluations for the application to experimental data at present day and at significantly increased future pile-up levels at the LHC. The BNN model and its inputs are deliberately designed to address these challenges by adapting the derived calibrations accordingly. This is mostly achieved by not using low-level data like calorimeter cell signals, but rather using features constructed from those as inputs. A potentially necessary retraining of the BNN in the case of a changing feature phase space due to changing operational conditions at the LHC is thus possible without the need for major signal reconstruction campaigns.

# Acknowledgements

We thank CERN for the very successful operation of the LHC and its injectors, as well as the support staff at CERN and at our institutions worldwide without whom ATLAS could not be operated efficiently.

The crucial computing support from all WLCG partners is acknowledged gratefully, in particular from CERN, the ATLAS Tier-1 facilities at TRIUMF/SFU (Canada), NDGF (Denmark, Norway, Sweden), CC-IN2P3 (France), KIT/GridKA (Germany), INFN-CNAF (Italy), NL-T1 (Netherlands), PIC (Spain), RAL (UK) and BNL (USA), the Tier-2 facilities worldwide and large non-WLCG resource providers. Major contributors of computing resources are listed in Ref. [40].

We gratefully acknowledge the support of ANPCyT, Argentina; YerPhI, Armenia; ARC, Australia; BMWFW and FWF, Austria; ANAS, Azerbaijan; CNPq and FAPESP, Brazil; NSERC, NRC and CFI, Canada; CERN; ANID, Chile; CAS, MOST and NSFC, China; Minciencias, Colombia; MEYS CR, Czech Republic; DNRF and DNSRC, Denmark; IN2P3-CNRS and CEA-DRF/IRFU, France; SRNSFG, Georgia; BMBF, HGF and MPG, Germany; GSRI, Greece; RGC and Hong Kong SAR, China; ICHEP and Academy of Sciences and Humanities, Israel; INFN, Italy; MEXT and JSPS, Japan; CNRST, Morocco; NWO, Netherlands; RCN, Norway; MNiSW, Poland; FCT, Portugal; MNE/IFA, Romania; MSTDI, Serbia; MSSR, Slovakia; ARIS and MVZI, Slovenia; DSI/NRF, South Africa; MICIU/AEI, Spain; SRC and Wallenberg Foundation, Sweden; SERI, SNSF and Cantons of Bern and Geneva, Switzerland; NSTC, Taipei; TENMAK, Türkiye; STFC/UKRI, United Kingdom; DOE and NSF, United States of America.

Individual groups and members have received support from BCKDF, CANARIE, CRC and DRAC, Canada; CERN-CZ, FORTE and PRIMUS, Czech Republic; COST, ERC, ERDF, Horizon 2020, ICSC-NextGenerationEU and Marie Skłodowska-Curie Actions, European Union; Investissements d'Avenir Labex, Investissements d'Avenir Idex and ANR, France; DFG and AvH Foundation, Germany; Herakleitos, Thales and Aristeia programmes co-financed by EU-ESF and the Greek NSRF, Greece; BSF-NSF and MINERVA, Israel; NCN and NAWA, Poland; La Caixa Banking Foundation, CERCA Programme Generalitat de Catalunya and PROMETEO and GenT Programmes Generalitat Valenciana, Spain; Göran Gustafssons Stiftelse, Sweden; The Royal Society and Leverhulme Trust, United Kingdom.

In addition, individual members wish to acknowledge support from Armenia: Yerevan Physics Institute (FAPERJ); CERN: European Organization for Nuclear Research (CERN DOCT); Chile: Agencia Nacional de Investigación y Desarrollo (FONDECYT 1230812, FONDECYT 1230987, FONDECYT

1240864); China: Chinese Ministry of Science and Technology (MOST-2023YFA1605700, MOST-2023YFA1609300), National Natural Science Foundation of China (NSFC - 12175119, NSFC 12275265, NSFC-12075060); Czech Republic: Czech Science Foundation (GACR - 24-11373S), Ministry of Education Youth and Sports (FORTE CZ.02.01.01/00/22_008/0004632), PRIMUS Research Programme (PRIMUS/21/SCI/017); EU: H2020 European Research Council (ERC - 101002463); European Union: European Research Council (ERC - 948254, ERC 101089007), European Union, Future Artificial Intelligence Research (FAIR-NextGenerationEU PE00000013), Italian Center for High Performance Computing, Big Data and Quantum Computing (ICSC, NextGenerationEU); France: Agence Nationale de la Recherche (ANR-20-CE31-0013, ANR-21-CE31-0013, ANR-21-CE31-0022, ANR-22-EDIR-0002); Germany: Baden-Württemberg Stiftung (BW Stiftung-Postdoc Eliteprogramme), Deutsche Forschungsgemeinschaft (DFG - 469666862, DFG - CR 312/5-2); Italy: Istituto Nazionale di Fisica Nucleare (ICSC, NextGenerationEU), Ministero dell'Università e della Ricerca (PRIN - 20223N7F8K - PNRR M4.C2.1.1); Japan: Japan Society for the Promotion of Science (JSPS KAKENHI JP22H01227, JSPS KAKENHI JP22H04944, JSPS KAKENHI JP22KK0227, JSPS KAKENHI JP23KK0245); Norway: Research Council of Norway (RCN-314472); Poland: Ministry of Science and Higher Education (IDUB AGH, POB8, D4 no 9722), Polish National Agency for Academic Exchange (PPN/PPO/2020/1/00002/U/00001), Polish National Science Centre (NCN 2021/42/E/ST2/00350, NCN OPUS 2023/51/B/ST2/02507, NCN OPUS nr 2022/47/B/ST2/03059, NCN UMO-2019/34/E/ST2/00393, NCN & H2020 MSCA 945339, UMO-2020/37/B/ST2/01043, UMO-2021/40/C/ST2/00187, UMO-2022/47/O/ST2/00148, UMO-2023/49/B/ST2/04085, UMO-2023/51/B/ST2/00920); Spain: Generalitat Valenciana (Artemisa, FEDER, IDIFEDER/2018/048), Ministry of Science and Innovation (MCIN & NextGenEU PCI2022-135018-2, MICIN & FEDER PID2021-125273NB, RYC2019-028510-I, RYC2020-030254-I, RYC2021-031273-I, RYC2022-038164-I); Sweden: Carl Trygger Foundation (Carl Trygger Foundation CTS 22:2312), Swedish Research Council (Swedish Research Council 2023-04654, VR 2018-00482, VR 2022-03845, VR 2022-04683, VR 2023-03403, VR grant 2021-03651), Knut and Alice Wallenberg Foundation (KAW 2018.0458, KAW 2019.0447, KAW 2022.0358); Switzerland: Swiss National Science Foundation (SNSF - PCEFP2_194658); United Kingdom: Leverhulme Trust (Leverhulme Trust RPG-2020-004), Royal Society (NIF-R1-231091); United States of America: U.S. Department of Energy (ECA DE-AC02-76SF00515), Neubauer Family Foundation.

# Appendix

## A  Local hadronic calibration sequence

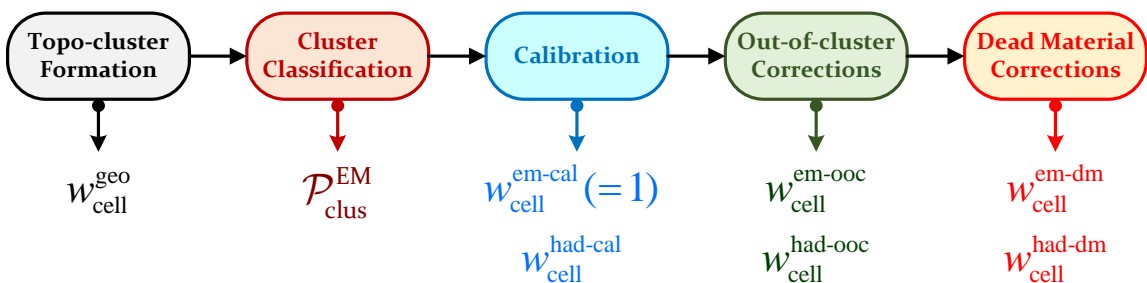

Figure 15: The LCW calibration sequence. Taken from Ref. [2].

The standard local hadronic calibration for topo-clusters employs cell signal weighting for hadronic calibration (local cell weighting, LCW), right after the cluster classification estimating the likelihood $\mathcal{P}_{\text{clus}}^{\text{EM}}$ of a cluster to be of electromagnetic origin, see sequence diagram shown in Figure 15. The electromagnetic and hadronic scale factors are applied at cell level such that the effective *cell signal weight* $w_{\text{cell}}^{\kappa}$ at each step $\kappa$ after the classification becomes

$$w_{\text{cell}}^{\kappa} = \mathcal{P}_{\text{clus}}^{\text{EM}} w_{\text{cell}}^{\text{em-}\kappa} + (1 - \mathcal{P}_{\text{clus}}^{\text{EM}}) w_{\text{cell}}^{\text{had-}\kappa} \quad \text{with} \quad \kappa \in \{\text{cal}, \text{ooc}, \text{dm}\} \ . \tag{21}$$

Here $\mathcal{P}_{\text{clus}}^{\text{EM}}$ is reconstructed in the classification step, and the 'em' and 'had' prefixes indicate the algorithm from which the signal weights are derived. The calibrated topo-cluster energies after each LCW calibration step $\kappa$ are reconstructed from sums of weighted cell energies, with weights

$$w_{\text{cell}}^{\kappa} w_{\text{cell}}^{\text{geo}} E_{\text{cell}} \ ,$$

applied to the cell signal $E_{\text{cell}}$ at the EM scale. The geometrical weight $w_{E_{\text{cell}}}^{\text{geo}}$ is $w_{E_{\text{cell}}}^{\text{geo}} < 1$ for cells shared between two topo-clusters, and $w_{E_{\text{cell}}}^{\text{geo}} = 1$ for cells collected into one topo-cluster only.

The topo-cluster energy $E_{\text{clus}}^{\text{had}}$, which reflects the hadronic calibration applied to the topo-cluster signal $E_{\text{clus}}^{\text{EM}}$ at step $\kappa = \text{cal}$ after classification, serves as a reference for comparisons with the machine-learned calibrations and is given by

$$E_{\text{clus}}^{\text{had}} = \sum_{i=1}^{N_{\text{clus}}^{\text{cell}}} w_{\text{cell},i}^{\text{geo}} E_{\text{cell},i}^{\text{EM}} \left[ \mathcal{P}_{\text{clus}}^{\text{EM}} w_{\text{cell},i}^{\text{had-em}} + \left( 1 - \mathcal{P}_{\text{clus}}^{\text{EM}} \right) w_{\text{cell},i}^{\text{had-had}} \right] = \sum_{i=1}^{N_{\text{clus}}^{\text{cell}}} w_{\text{cell},i}^{\text{cal}} E_{\text{cell},i}^{\text{EM}} \ . \tag{22}$$

Here $N_{\text{clus}}^{\text{cell}}$ is the total number of cells in the topo-cluster. Typical weights are $w_{\text{cell}}^{\text{em-cal}} = 1$, as $E_{\text{cell}}$ is already at EM scale, and $w_{E_{\text{cell}}}^{\text{had-cal}} \geq 1$ to calibrate non-compensation ($e/h > 1$), both used to construct the effective cell signal weight $w_{\text{cell}}^{\text{cal}}$ at the calibration step. The electromagnetic and hadronic weights applied in the following correction steps for out-of-cluster and inactive material energy losses are typically larger than unity. As they depend on the nature of the underlying particle shower, they are typically also not identical for the respective electromagnetic and hadronic correction.

# B Features

## B.1 Feature construction

All features collected into the topo-cluster feature set $\mathcal{X}_{\text{clus}}$ defined in Eq. 3 and further described in Table 1 in Section 3.1.2, are documented in Ref. [2], except for the cluster compactness measure $p_{\text{T}}D$ and the variance $\text{Var}_{\text{clus}}(t_{\text{cell}})$ of the cell time distribution within the cluster.

The construction of $p_{\text{T}}D$ is inspired by a variable introduced in the context of quark-gluon jet tagging in Ref. [30]. While preserving the naming convention introduced in the original reference, the definition of $p_{\text{T}}D$ is slightly modified for computational efficiency when considering that clustered calorimeter cells are directionally in close proximity, by replacing the transverse momenta of the jet constituents in the original definition by the cell energy $E_{\text{cell}}$ such that

$$p_{\text{T}}D = \frac{\sqrt{\sum_{i|E^{\text{EM}}_{\text{cell},i}>0}(w^{\text{geo}}_{\text{cell},i} E^{\text{EM}}_{\text{cell},i})^2}}{\sum_{i|E^{\text{EM}}_{\text{cell},i}>0} w^{\text{geo}}_{\text{cell},i} E^{\text{EM}}_{\text{cell},i}} \ . \tag{23}$$

Only cells with energies $E_{\text{cell}} > 0$ are considered in the sums. By definition, $p_{\text{T}}D$ is numerically restricted to $0 < p_{\text{T}}D \leq 1$.

A compactness of $p_{\text{T}}D \to 1$ indicates that the energy $E^{\text{EM}}_{\text{clus}}$ of the topo-cluster is increasingly contained in only one cell. A compactness measure of $p_{\text{T}}D = 1$ exact is only possible for a single cell cluster, which is not expected to happen, due to the topo-cluster formation rules stating that at least all neighbours of a seed cell are collected [2]. Very small number of cells are possible for topo-clusters at the edges of the calorimeter, like at the outer circumference of the Tile, the back of the HEC, and the front faces of the electromagnetic calorimeters EMB and EMEC. A smaller value of $p_{\text{T}}D$ indicates a disperse distribution of the energy across the cells in the cluster.

The variance $\text{Var}_{\text{clus}}(t_{\text{cell}})$ of the cell time ($t_{\text{cell}}$) distribution is given by

$$\text{Var}_{\text{clus}}(t_{\text{cell}}) = \langle t^2_{\text{cell}} \rangle - t^2_{\text{clus}} \ , \tag{24}$$

where $t_{\text{cell}}$ are the cell times and $t_{\text{clus}}$ is the cluster time. The second moment $\langle t^2_{\text{cell}} \rangle$ of the $t_{\text{cell}}$ distribution for cells collected in the topo-cluster is defined by

$$\langle t^2_{\text{cell}} \rangle = \frac{\sum_{i|\zeta_{\text{cell},i}>2}(w^{\text{geo}}_{\text{cell},i} E^{\text{EM}}_{\text{cell},i})^2 \cdot t^2_{\text{cell},i}}{\sum_{i|\zeta_{\text{cell},i}>2}(w^{\text{geo}}_{\text{cell},i} E^{\text{EM}}_{\text{cell},i})^2} \ , \tag{25}$$

following the definition of $t_{\text{clus}}$, which is constructed as a first moment of the $t_{\text{cell}}$ distribution [2]. Only cells with a signal significance (signal-over-noise) of $\zeta_{\text{cell}} > 2$ are considered when calculating $t_{\text{clus}}$ and $\text{Var}_{\text{clus}}(t_{\text{cell}})$.

## B.2 Relation of features to topo-cluster nature

The features collected into the feature set $\mathcal{X}_{\text{clus}}$ defined in Eq. 3 in Section 3.1.2 are evaluated with respect to the composition of the topo-cluster signal source $E^{\text{dep}}_{\text{clus}}$. For this, feature distributions for the three categories defined in Section 2.3.2, electromagnetic topo-clusters, hadronic topo-clusters and composite

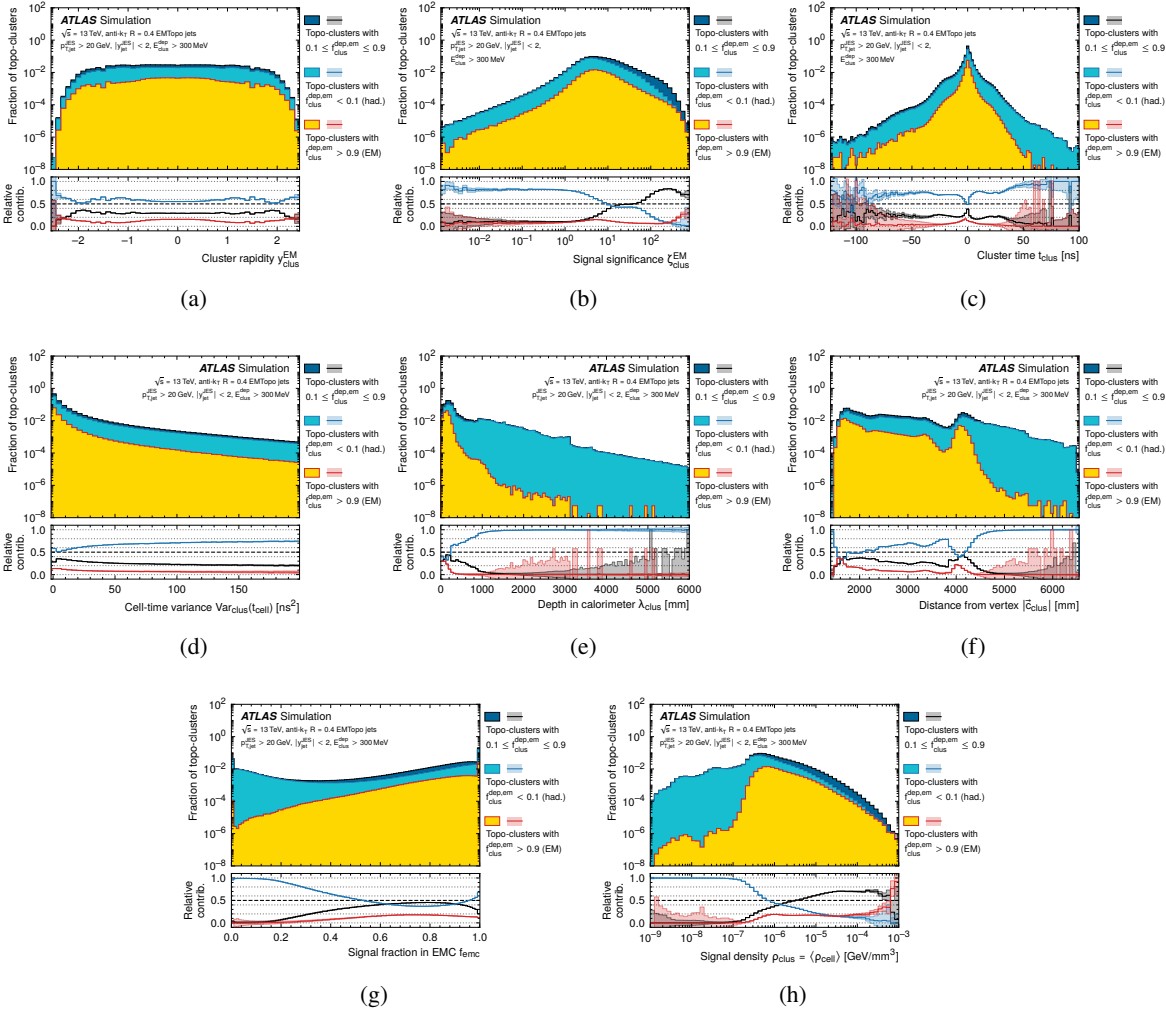

Figure 16: Distributions of some of the features given in Eq. 3 for topo-clusters classified as hadronic, electromagnetic or composite following the definitions given in Section 2.3.2. Distributions are shown for (a) the rapidity $y_{\text{clus}}^{\text{EM}}$, (b) the signal significance $\zeta_{\text{clus}}^{\text{EM}}$, (c) the time $t_{\text{clus}}$, (d) the variance of the cell time distribution $\text{Var}_{\text{clus}}(t_{\text{cell}})$, (e) the distance from the calorimeter front-face $\lambda_{\text{clus}}$, (f) the distance from the nominal vertex $|\vec{c}_{\text{clus}}|$, (g) the fraction of energy in the electromagnetic calorimeter $f_{\text{emc}}$, and (h) the signal density $\rho_{\text{clus}}$. The fractional contributions from the three topo-cluster categories are shown in the respective lower panels of each figure. The shaded regions show the statistical uncertainties. More information about the definition of $\text{Var}_{\text{clus}}(t_{\text{cell}})$ is given in Appendix B.1. The other features are defined in Ref. [2]. An overview on all features is given in Table 1.

topo-clusters, are compared in Figures 16 and 17. Their relative contribution to the corresponding inclusive distributions is shown in same figures.

As the topo-clusters considered here are extracted from jets, the typical fraction of electromagnetic topo-clusters in the sample can be estimated. Using the canonical fraction of the jet energy carried by photons, about 25%, together with the average topo-cluster multiplicity for hadrons shown in Ref. [2] and the assumption that each photon in a jet generates one cluster, this fraction is about 10%. This estimate is only given as guideline, as it ignores all full or partial merging of shower signals into one topo-cluster and all shower signal losses introduced by the topological cell clustering algorithm and its configuration.

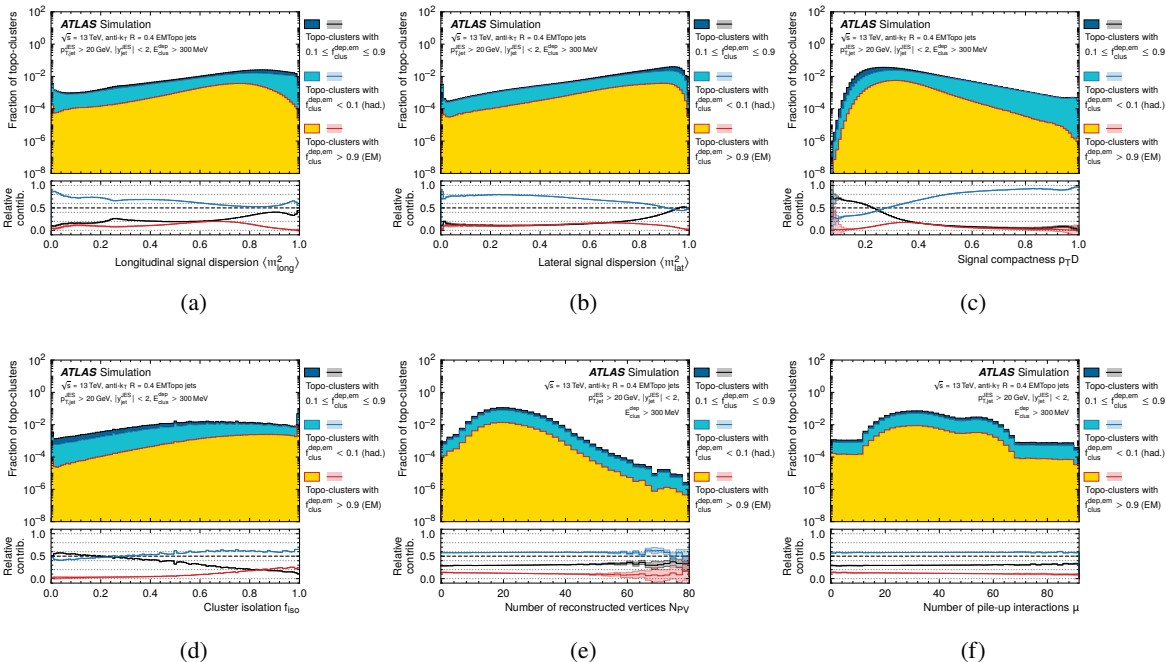

Figure 17: Distributions of some of the features given in Eq. 3 for topo-clusters classified as hadronic, electromagnetic or composite following the definitions given in Section 2.3.2. Distributions are shown for (a) the longitudinal energy dispersion $\langle m_{\text{long}}^2 \rangle$, (b) the lateral energy dispersion $\langle m_{\text{lat}}^2 \rangle$, (c) the compactness measure $p_{\text{T}}D$, (d) the isolation measure $f_{\text{iso}}$, (e) the number of reconstructed vertices $N_{\text{PV}}$, and (f) the number of pile-up interactions $\mu$. The fractional contributions from the three topo-cluster categories are shown in the respective lower panels of each figure. The shaded regions show the statistical uncertainties. More information about the definition of $p_{\text{T}}D$ is given in Appendix B.1. The other features are defined in Ref. [2]. An overview on all features is given in Table 1.

Other feature distributions with near-constant contributions from electromagnetic topo-clusters are $y_{\text{clus}}^{\text{EM}}$ shown in Figure 16(a), $N_{\text{PV}}$ in Figure 17(e) and $\mu$ in Figure 17(f). Those also show approximately constant contributions from the other topo-cluster categories. This insensitivity to the cluster nature is expected, as neither $y_{\text{clus}}^{\text{EM}}$ nor $N_{\text{PV}}$ or $\mu$ should show any effect on the composition of the energy deposits, which to first order only depend on the fragmentation of jets from the hard-scatter interactions. Slight variations may be introduced by the amount of inactive material in front of the active calorimeter volume that shows some variations with $y_{\text{clus}}^{\text{EM}}$. These features are kept in $\mathcal{X}_{\text{clus}}$ because they provide sensitivity to changes of the detector readout geometry, including the varying projection of the particle showers onto the calorimeter cell geometry and thus the reconstruction of most spatial, compactness and density features, and the pile-up environment. Among the strongest indicators of the topo-cluster nature are $\zeta_{\text{clus}}^{\text{EM}}$ (Figure 16(b)), the location measures $\lambda_{\text{clus}}$ (Figure 16(e)) and $|\vec{c}_{\text{clus}}|$ (Figure 16(f)), $f_{\text{emc}}$ (Figure 16(g)), $\rho_{\text{clus}}$ (Figure 16(h)), the dispersion and compactness measures $\langle m_{\text{long}}^2 \rangle$, $\langle m_{\text{lat}}^2 \rangle$ and $p_{\text{T}}D$ in Figures 17(a)–17(c). These figures all confirm expectations from electromagnetic and hadronic shower developments in the ATLAS calorimeters. For example, the contribution of electromagnetic topo-clusters to the $f_{\text{emc}}$ distribution is generated by the photons in jets, nearly all of which are represented as a single cluster. They are mostly stopped in the electromagnetic calorimeters. This is in full agreement with the observation that electromagnetic topo-clusters are more isolated, as they contribute more than 25% of all clusters with $f_{\text{iso}} \gtrsim 0.8$, at an inclusive contribution of about 10%, as discussed above.

The larger contribution of electromagnetic topo-clusters to the population of higher-density clusters seen in Figure 16(h) is also well within the expectation of higher-density deposits in electromagnetic showers in the smaller volume cells of the electromagnetic calorimeters, with a higher signal yield when compared with hadronic showers. Also as expected, hadronic topo-clusters are dominantly contributing to the population of clusters deep in the calorimeter, as seen for the $|\vec{c}_{\text{clus}}|$ and $\lambda_{\text{clus}}$ distributions in Figures 16(e) and 16(f). Compact topo-clusters with $p_{\text{T}}D \gtrsim 0.6$ are most likely of hadronic nature, as seen in Figure 17(c). These compact topo-clusters often have one dominant cell signal, which can occur in the course of the hadronic shower development in the hadronic calorimeters with their coarse spatial readout granularity and e.g., in the Tile calorimeter for signals from ionisations by charged hadrons. As already seen, electromagnetic topo-clusters are much more likely to appear in the highly granular electromagnetic calorimeters, where despite their high energy loss density signal sharing between cells is very likely, due to the relatively small cell size, compared to typical electromagnetic shower sizes.

## C  Network implementations and loss functions

### C.1  Bayesian loss function

As discussed in Section 3.3, a Bayesian neural network (BNN) is employed to encode the response function $\mathcal{R}(\mathcal{X}_{\text{clus}}) = \mathcal{R}_{\text{clus}}^{\text{BNN}}(\mathcal{X}_{\text{clus}})$ and its uncertainty over the space of all features sets $\mathcal{X}_{\text{clus}}$ found for the topo-clusters in the training dataset. It is designed to learn $\mathcal{R} = \mathcal{R}(\mathcal{X}_{\text{clus}})$ as a probability distribution $p(\mathcal{R})$, with the mean

$$\langle \mathcal{R} \rangle = \int \mathrm{d}\mathcal{R} \, \mathcal{R} p(\mathcal{R}) \, . \tag{26}$$

The network encodes the learned probability in weight configurations, conditional on the training data. The training can be described as constructing a variational approximation [41, 42], replacing $p(\theta|D_{\text{train}})$ introduced in Section 3.3 with a learned $q(\theta)$ such that

$$p(\mathcal{R}) = \int \mathrm{d}\theta \, p(\mathcal{R}|\theta)p(\theta|D_{\text{train}}) \approx \int \mathrm{d}\theta \, p(\mathcal{R}|\theta)q(\theta) \, . \tag{27}$$

The dataset $D_{\text{train}}$ is defined over the feature space containing all possible feature sets $\mathcal{X}_{\text{clus}}$, following the nomenclature given in Eq. 11 of Section 3.3.2. To learn $q(\theta)$, the Kullback-Leibler (KL) [36] divergence including the Bayesian transform $p(\theta|D_{\text{train}}) \rightarrow p(D_{\text{train}}|\theta)$

$$\begin{aligned} D_{\text{KL}}[q(\theta), p(\theta|D_{\text{train}})] &= \int \mathrm{d}\theta \, q(\theta) \log \frac{q(\theta)}{p(\theta|D_{\text{train}})} \\ &= \int \mathrm{d}\theta \, q(\theta) \log \frac{q(\theta)p(D_{\text{train}})}{p_{\text{prior}}(\theta)p(D_{\text{train}}|\theta)} \\ &\approx D_{\text{KL}}[q(\theta), p_{\text{prior}}(\theta)] - \underbrace{\int \mathrm{d}\theta \, q(\theta) \log p(D_{\text{train}}|\theta)}_{\langle \log p(D_{\text{train}}|\theta) \rangle_{\theta \sim q(\theta)}} \end{aligned} \tag{28}$$

is minimized. The explicit form of $p(D_{\text{train}}|\theta)$ is given in Eq. 12 in Section 3.3.2.

The first term in Eq. 28 arises from the prior $p_{\text{prior}}(\theta)$ and can be viewed as a weight regularisation. The second term samples from the negative logarithmic likelihood, where the sampling generalises the standard dropout in neural network training. The BNN loss is then defined as given in Eq. 13,

$$\mathcal{L}_{\text{BNN}} = D_{\text{KL}}[q(\theta), p_{\text{prior}}(\theta)] - \left\langle \log p(D_{\text{train}}|\theta) \right\rangle_{\theta \sim q(\theta)} . \tag{29}$$

A Gaussian distribution is used as a convenient prior $p_{\text{prior}}(\theta)$. Its width is a hyper-parameter that could be varied on a performance plateau to make the training more efficient [11]. If the weight distributions $q(\theta)$ are also chosen as Gaussian functions, the KL-divergence can be computed analytically. Provided the network is sufficiently deep, the assumption of independent Gaussian distributions for each network parameter should not affect its expressiveness.

## C.2 Implementation of repulsive ensemble networks

The principle implementation of the repulsive ensemble (RE) network training is described in Ref. [3]. As already mentioned in Section 3.3.3, RE networks modify the update rule that minimises $-\log p(\theta|D_{\text{train}})$ by gradient descent. This is done by evaluating this update rule for an ensemble of $N$ networks and introduce a repulsive term that forces the ensemble to spread out and cover the loss around the actual minimum. The repulsive term is designed to increase with the proximity of the network parameters $\theta$ of a given ensemble member to all other members. It is implemented by a Gaussian kernel $k(\theta, \theta_j)$, with $j = 1, \ldots, N$, such that the update from training step $t$ to training step $t + 1$ becomes

$$\theta^{t+1} = \theta^t + \alpha \nabla_{\theta^t} \log p(\theta|D_{\text{train}}) - \alpha \frac{\nabla_{\theta^t} \sum_{j=1}^{N} k(\theta^t, \theta_j^t)}{\sum_{i=1}^{N} k(\theta^t, \theta_i^t)} . \tag{30}$$

The first two terms represent the standard network training by gradient descent, with the learning rate $\alpha$ and a gradient $\nabla_{\theta^t}$ reflecting the evolution of the network parameters $\theta^t$ at the training step $t \to t + 1$. The additional kernel term runs over the $N$ networks of an ensemble and defines repulsive ensembles.

Using properties of ordinary differential equations describing the network training as a time evolution and the unique stationary solution of the Fokker-Planck equation (see e.g., Ref. [43]), it is found that this update rule leads to an ensemble of networks with parameters sampled from $\theta \sim p(\theta|D_{\text{train}})$.

To fit a calibration function for topo-clusters, and similar other regression tasks, the function encoded by the network is of interets, not its latent representation $\theta$ in weight space. For instance, two networks can encode the same function by permuting the weights, unaffected by a repulsive force in weight space. The repulsive term therefore needs to act in the space of the actual network outputs $\mathcal{R}_{\text{clus}}^{\text{RE}}(X_{\text{clus}}) = \mathcal{R}_{\theta}(X_{\text{clus}})$ represented by $\theta$. The update rule in this function space is given by

$$\frac{\mathcal{R}^{t+1} - \mathcal{R}^t}{\alpha} = \nabla_{\mathcal{R}^t} \log p(\mathcal{R}|D_{\text{train}}) - \frac{\sum_j \nabla_{\mathcal{R}^t} k(\mathcal{R}, \mathcal{R}_j)}{\sum_j k(\mathcal{R}, \mathcal{R}_j)} . \tag{31}$$

It is formally similar to the one in the weight space given in Eq. 30.

The network training is still performed in the weight space, so a translation of the function-space update rule into this space is needed,

$$\frac{\theta^{t+1} - \theta^t}{\alpha} = \nabla_{\theta^t} \log p(\theta^t|D_{\text{train}}) - \frac{\partial \mathcal{R}^t}{\partial \theta^t} \frac{\sum_j \nabla_{\mathcal{R}} k(\mathcal{R}_{\theta^t}, \mathcal{R}_{\theta_j^t})}{\sum_j k(\mathcal{R}_{\theta^t}, \mathcal{R}_{\theta_j^t})} . \tag{32}$$

The repulsive kernel in the function space cannot be evaluated, so the function must be evaluated for a finite batch of topo-clusters represented by points $x = \mathcal{X}_{\text{clus}}$ in the feature space,

$$\frac{\theta^{t+1} - \theta^t}{\alpha} \approx \nabla_{\theta^t} \log p(\theta^t | D_{\text{train}}) - \frac{\sum_{j=1}^{N} \nabla_{\theta^t} k(\mathcal{R}_{\theta^t}(x), \mathcal{R}_{\theta_j^t}(x))}{\sum_{j=1}^{N} k(\mathcal{R}_{\theta^t}(x), \mathcal{R}_{\theta_j^t}(x))} \ . \tag{33}$$

Here $\mathcal{R}_{\theta^t}(x)$ in Eqs. 32 and 33 is understood as evaluating the function $\mathcal{R}$ for members $x$ of $D_{\text{train}}$ at training step $t$.

## C.3  Repulsive ensemble loss function

The modified update rule of Eq. 33 can be turned into a loss function for the RE training. Using Bayes' theorem and neglecting the evidence $p(D_{\text{train}})$, the likelihood loss measure $p(\theta | D_{\text{train}})$ can be written as

$$\log p(\theta | D_{\text{train}}) = \log p(D_{\text{train}} | \theta) \underbrace{- \frac{\theta^2}{2\sigma^2} + \text{const.}}_{\text{exponential of } p_{\text{prior}}(\theta)} . \tag{34}$$

The prior $p_{\text{prior}}(\theta)$ is chosen to be Gaussian.

For a training dataset of size $N_{\text{train}}$, split into batches of size $N_{\text{batch}}$, $\mathcal{R}_{\theta^t}(x)$ is evaluated as the function of members $x_1, \ldots, x_{N_{\text{batch}}}$ for all batches of the training dataset $D_{\text{train}}$. The update rule from Eq. 33 can be rewritten using Eq. 34,

$$\frac{\theta^{t+1} - \theta^t}{\alpha} \approx \nabla_{\theta^t} \frac{N_{\text{train}}}{N_{\text{batch}}} \sum_{b=1}^{N_{\text{batch}}} \log p(x_b | \theta) - \frac{\sum_{j=1}^{N} \nabla_{\theta^t} k(\mathcal{R}_{\theta_i}(x), \mathcal{R}_{\theta_j}(x))}{\sum_{j=1}^{N} k(\mathcal{R}_{\theta_i}(x), \mathcal{R}_{\theta_j}(x))} - \nabla_{\theta^t} \frac{\theta^2}{2\sigma^2} \ . \tag{35}$$

Based on this rule, the loss function is constructed by inverting the sign in front of the gradient in Eq. 35 and divide by the training dataset size $N_{\text{train}}$ to remove the scaling with $N_{\text{train}}$. In addition, not all occurrences of $\theta$ are inside the gradient, so a *stop-gradient operation* is introduced, denoted by $\overline{\mathcal{R}}_{\theta_j}(x)$. With this, the RE loss function is given by

$$\mathcal{L}_{\text{RE}} = \sum_{i=1}^{N} \left[ \underbrace{- \frac{1}{N_{\text{batch}}} \sum_{b=1}^{N_{\text{batch}}} \log p(x_b | \theta_i)}_{\text{(1) negative logarithmic likelihood averaged over batch}} + \overbrace{\frac{1}{N_{\text{train}}} \frac{\sum_{j=1}^{N} k(\mathcal{R}_{\theta_i}(x), \overline{\mathcal{R}}_{\theta_j}(x))}{\sum_{j=1}^{N} k(\overline{\mathcal{R}}_{\theta_i}(x), \overline{\mathcal{R}}_{\theta_j}(x))}}^{\text{(2) repulsive term in function space}} + \underbrace{\frac{1}{N_{\text{train}}} \frac{\theta_i^2}{2\sigma^2}}_{\text{(3) normalised exponent of } p_{\text{prior}}(\theta_i)} \right] . \tag{36}$$

Here the numbers (1), (2) and (3) refer to the discussion of the corresponding terms in Eq. 14 of Section 3.3.4. The normalised exponent of the prior implements a ridge ($L_2$) regularisation (see, for example, Ref. [44]). It controls possible over-fitting and focusses the fit on regions of relevant data.

A typical choice for the repulsive term is a Gaussian kernel, with a width given by the *median heuristic*[†] [45] between the $N$ ensemble members. It is divided by the training dataset size $N_{\text{train}}$ to remove any scaling with $N_{\text{train}}$.

---

[†] This is the median of all pairwise distances between $\theta_i$ and $\theta_j$ for all $N$ ensemble members ($i = 1, \ldots, N$ and $j = 1, \ldots, N$).

# D  Evaluation metrics

## D.1  Performance

For the evaluation of the performance of the machine-learned topo-cluster calibration, distributions of test variables like the prediction power $\Delta_{\mathcal{R}}$ defined in Eq. 15 in Section 4.1.1 and the deviation from signal linearity $\Delta_E$ defined in Eq. 16 in Section 4.1.2 are characterised by a measure of the central tendency and a corresponding measure of their spread. The median value of the distribution is used to express the central tendency, because it is an unbiased measure that does not require any assumption on the shape of the distribution (like the mode does) and it shows the least dependence on the sample size, compared with the statistical average of an asymmetric distribution with developing tails.

The spread is defined by the *inter-quantile range* $Q^w_{f=68\%}$,

$$Q^w_{f=68\%} = \min(x_k - x_i) = x^{\text{right}}_q - x^{\text{left}}_q \quad \text{such that} \quad \frac{\int_{x^{\text{left}}_q}^{x^{\text{right}}_q} \mathrm{d}x \, p(x)}{\int_{-\infty}^{+\infty} \mathrm{d}x \, p(x)} = 0.68 \;, \tag{37}$$

which for any distribution $p(x)$ is the smallest of all $x$ value ranges that contains 68% of the collected statistics. This range corresponds to the $[-1\sigma, +1\sigma]$ range around the mean of a Gaussian distribution, and is symmetric around any measure of central tendency in case of symmetric distributions. Here $x^{\text{left}}_q$ and $x^{\text{right}}_q$ are the respective lower and upper limit of the $x$-value range defining the *inter-quantile* region $\text{IQR}_{f=68\%}$,

$$\text{IQR}_{f=68\%} = \left[ x^{\text{left}}_q, x^{\text{right}}_q \right] = \left[ x^{\text{left}}_q, x^{\text{left}}_q + Q^w_{f=68\%} \right] \;. \tag{38}$$

With the choices of distribution descriptors given above, a measure for the relative energy resolution $\sigma_{\text{rel}}$ can be constructed from the $\Delta_E$ distributions that employs both its median $\langle \Delta_E \rangle_{\text{med}}$ and its spread $Q^w_{f=68\%}(\Delta_E)$,

$$\sigma_{\text{rel}} = \frac{Q^w_{f=68\%}(\Delta_E)}{2(\langle \Delta_E \rangle_{\text{med}} + 1)} \;. \tag{39}$$

If $\Delta_E$ is distributed following a Gaussian, $Q^w_{f=68\%}(\Delta_E) = 2\sigma(\Delta_E)$ such that

$$\sigma_{\text{rel}} = \frac{\sigma(\Delta_E)}{\langle \Delta_E \rangle + 1} = \frac{\sigma(E^{\text{BNN(RE)}}_{\text{clus}})}{\langle E^{\text{BNN(RE)}}_{\text{clus}} \rangle} \approx \frac{\sigma(E^{\text{BNN(RE)}}_{\text{clus}})}{\langle E^{\text{dep}}_{\text{clus}} \rangle}$$

is the relative energy resolution for calorimeters. Equation 39 can therefore be considered as an appropriate representation of this resolution.

## D.2 Uncertainties

The learned uncertainties from the BNN are derived by sampling the network parameters $\theta$ and calculating the variance by

$$\langle \mathcal{R} \rangle = \int d\theta \, q(\theta) \langle \mathcal{R} \rangle_\theta \quad \text{with} \quad \langle \mathcal{R} \rangle_\theta = \int d\mathcal{R} \, \mathcal{R} p(\mathcal{R}|\theta) \, ,$$

$$\sigma_{\text{tot}}^2 = \int d\theta \, q(\theta) \left[ \langle \mathcal{R}^2 \rangle_\theta - \langle \mathcal{R} \rangle_\theta^2 + (\langle \mathcal{R} \rangle_\theta - \langle \mathcal{R} \rangle)^2 \right] \equiv \sigma_{\text{syst}}^2 + \sigma_{\text{stat}}^2 \, . \tag{40}$$

Here $\langle \mathcal{R} \rangle$ is the expectation value for the prediction $\mathcal{R}$ considering the probability $p(\mathcal{R}|\theta)$ that the actual $\theta$ represents $\mathcal{R}$, and $\sigma_{\text{tot}}^2$ is the variance associated with this expectation.[‡]

The total uncertainty $\sigma_{\text{tot}}^2$ factorises into two terms. The first term in Eq. 40 vanishes in the limit of arbitrarily well-known data, $p(\mathcal{R}|\theta) \rightarrow \delta(\mathcal{R}_{\text{clus}}^{\text{BNN}}(\theta) - \mathcal{R}_{\text{clus}}^{\text{EM}})$,

$$\sigma_{\text{syst}}^2 = \int d\theta \, q(\theta) \left[ \langle \mathcal{R}^2 \rangle_\theta - \langle \mathcal{R} \rangle_\theta^2 \right] \, . \tag{41}$$

This uncertainty component approaches a plateau for large training data sets, and is therefore considered as a systematic uncertainty that cannot be reduced by increasing dataset sizes or the number of training attempts (epochs). It is predicted for each individual topo-cluster that is calibrated using $\mathcal{R}_{\text{clus}}^{\text{BNN}}$.

The second contribution is the $\theta$-sampled variance

$$\sigma_{\text{stat}}^2 = \int d\theta \, q(\theta) \left[ \langle \mathcal{R} \rangle_\theta - \langle \mathcal{R} \rangle \right]^2 \, . \tag{42}$$

It vanishes in the limit of perfect training, $q(\theta) \rightarrow \delta(\theta - \theta_0)$. This limit can be approached for large training datasets. This uncertainty component is therefore referred to as statistical.

Contrary to the BNN, where the predictions are sampled in the variational weight space defined by $\theta$ to determine the uncertainties, the RE approach samples the predictions in function space, as briefly introduced in Appendix C.3. Also in this case, and for sufficient large training datasets like the one used for the topo-cluster calibration, statistical and systematic uncertainty components can be distinguished and compared to the ones from the BNN.

## D.3 Expectations for the pull distribution

The pull distributions shown in Figure 12(b) can be formally understood, and expectations for its width can be derived from the following considerations.

Generally, the pull of a learned distribution is defined as

$$t(x) = \frac{f_\theta(x) - f(x)}{\sigma(x)} \, , \tag{43}$$

where $f_\theta(x)$ is the learned function with parameters $\theta$, $f(x)$ the true function encoded in the training data, and $\sigma(x)$ is the learned uncertainty. For the calibration training discussed in this paper, $f_\theta(x) = \mathcal{R}_\theta(\mathcal{X}_{\text{clus}})$ for a learned set of network parameters $\theta$ and topo-cluster feature sets $\mathcal{X}_{\text{clus}}$, and $f(x) = \mathcal{R}_{\text{clus}}^{\text{EM}}(\mathcal{X}_{\text{clus}})$.

---

[‡] $\langle \mathcal{R} \rangle_\theta^2$ is defined in analogy to $\langle \mathcal{R} \rangle_\theta$.

Ideally, $\sigma(x)$ captures the absolute value of the deviation of the learned function from the target exactly and for any $x$-value,

$$f_\theta(x) = f(x) \pm \sigma(x) \quad \Rightarrow \quad t(x) = \pm 1 . \tag{44}$$

Alternatively, $\sigma(x)$ can capture the maximum absolute value of the deviation, but the actual deviation of $f_\theta(x)$ from $f(x)$ might be smaller. This actual deviation might also follow a flat distribution within the limits of $t(x) = \pm 1$,

$$p(f_\theta(x)) = \begin{cases} \text{const.} & f_\theta(x) \in [f(x) - \sigma(x), f(x) + \sigma(x)] \\ 0 & f_\theta(x) \notin [f(x) - \sigma(x), f(x) + \sigma(x)] \end{cases}$$

$$\Rightarrow \quad p(t(x)) = \begin{cases} \text{const.} & t \in [-1, 1] \\ 0 & t \notin [-1, 1] \end{cases} . \tag{45}$$

For a stochastic system like the one represented by the topo-clusters, the learned values $f_\theta(x)$ are expected have a mean $\langle f_\theta(x) \rangle = f(x)$, but with a Gaussian smearing $\mathcal{N}(f(x), \sigma(x))$ with $x$-dependent width $\sigma(x)$ introduced by the (mostly pile-up) noise,

$$p(f_\theta(x)) = \mathcal{N}(f(x), \sigma(x)) \propto \exp\left(-\frac{[f_\theta(x) - f(x)]^2}{2\sigma(x)^2}\right)$$

$$\Rightarrow \quad p(t(x)) = \mathcal{N}(0, 1) . \tag{46}$$

This shows that for a stochastic smearing the pull follows a standard Gaussian distribution.

# E  Calibration performance

## E.1  Performance in feature space

The deviation from signal linerarity is evaluated before and after local calibrations conprising the hadronic calibration from LCW, the BNN-derived and the DNN-derived calibrations. Before local calibration, the topo-cluster signal $E_{\text{clus}}^{\text{EM}}$ is provided on the basic EM scale. As discussed in Section 4.1.2, $E_{\text{clus}}^{\text{EM}}$ is considered as uncalibrated and cannot be expected to have a direct proportionality with the deposited energy $E_{\text{clus}}^{\text{dep}}$. The topo-cluster energy at the hadronic scale from LCW $E_{\text{clus}}^{\text{LCW}}$ is expected to represent a calibrated signal, thus it is expected to be calibrated. Both of these scales do not use any of the features presented here as inputs to their respective signal reconstruction. Both topo-cluster energies calibrated using the BNN ($E_{\text{clus}}^{\text{BNN}}$) or the DNN ($E_{\text{clus}}^{\text{DNN}}$) predictions are expected to show a more linear response, as all features are used as inputs to their respective networks.

Figure 18 summarises the deviation from linearity for all four signal scales as a function of features only used for the BNN and DNN predictions.

## E.2  Local relative energy resolution

Figure 19 shows the median deviation from linearity $\langle \Delta_E \rangle_{\text{med}}$ for the topo-cluster energy scales after applying the calibrations learned by the BNN and the DNN, as a function of the energy $E_{\text{clus}}^{\text{dep}}$ deposited in the cluster. The slices (a), (b) and (c) show individual distributions of $\Delta_E$ in selected bins of $E_{\text{clus}}^{\text{dep}}$, with the clear sign of significantly suppressed tails for the BNN-derived calibration at lower $E_{\text{clus}}^{\text{dep}}$. At higher $E_{\text{clus}}^{\text{dep}} \gtrsim 600\,\text{MeV}$ both predictions show increasingly similar distributions, with $\langle \Delta_E^{\text{BNN}} \rangle_{\text{med}} \approx 0$ and $\langle \Delta_E^{\text{DNN}} \rangle_{\text{med}} < 0$ for $E_{\text{clus}}^{\text{dep}} \gtrsim 80\,\text{GeV}$ indicating a more accurate median calibration by the trained BNN.

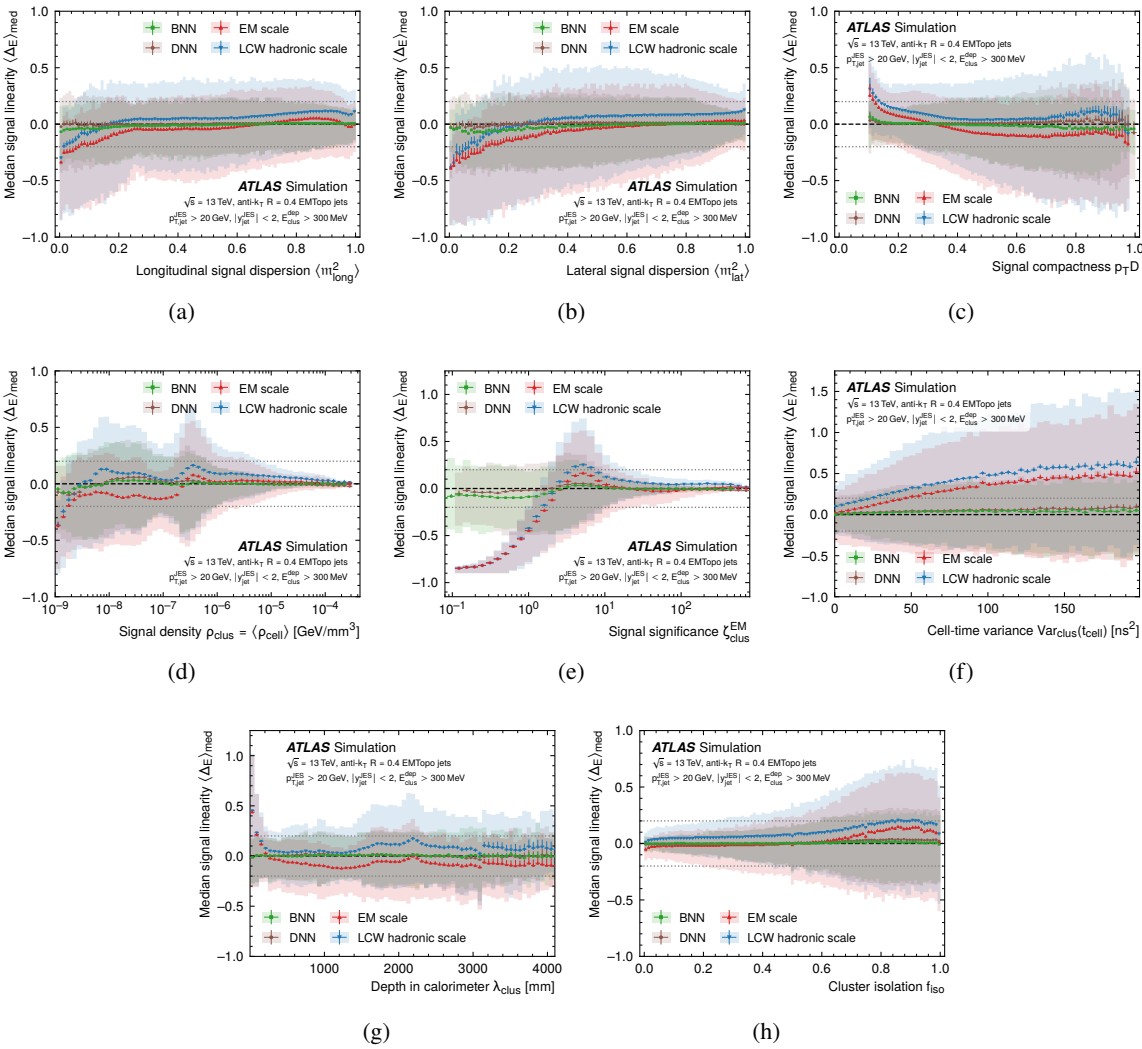

Figure 18: The median deviation from signal linearity for the topo-cluster at the basic EM scale ($\langle\Delta_E^{\mathrm{EM}}\rangle_{\mathrm{med}}$), the hadronic calibration from LCW ($\langle\Delta_E^{\mathrm{had}}\rangle_{\mathrm{med}}$), and at the BNN-learned ($\langle\Delta_E^{\mathrm{BNN}}\rangle_{\mathrm{med}}$) and the DNN-learned ($\langle\Delta_E^{\mathrm{DNN}}\rangle_{\mathrm{med}}$) calibrated energy scales, evaluated as a function of (a) the longitudinal signal dispersion measure $\langle\mathfrak{m}_{\mathrm{long}}^2\rangle$, (b) the lateral signal dispersion measure $\langle\mathfrak{m}_{\mathrm{lat}}^2\rangle$, (c) the signal compactness measure $p_{\mathrm{T}}D$ (see Appendix B.1), (d) the cluster signal density measure $\rho_{\mathrm{clus}}$, (e) the cluster signal significance $\zeta_{\mathrm{clus}}^{\mathrm{EM}}$, (f) the second moment of the energy-squared-weighted cell time distribution $\mathrm{Var}_{\mathrm{clus}}(t_{\mathrm{cell}})$ (see Appendix B.1), (g) the distance of the cluster centre-of-gravity from the calorimeter front face $\lambda_{\mathrm{clus}}$, and (h) the cluster isolation measure $f_{\mathrm{iso}}$. The topo-clusters of the test sample used to produce these results are extracted as described in Section 3.1.1.

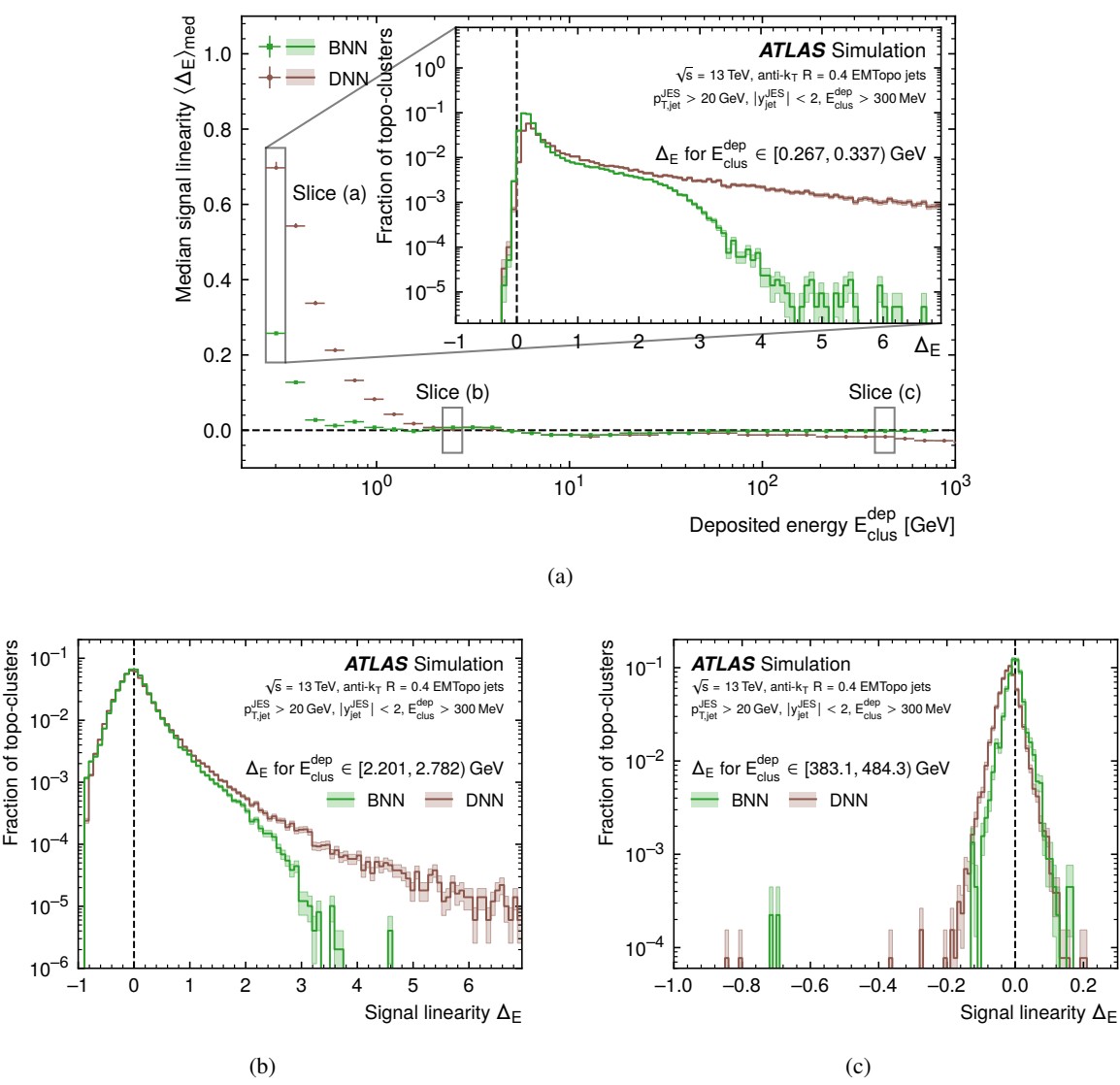

Figure 19: In (a), the median deviation from linearity $\langle \Delta_E \rangle_{\mathrm{med}}$ is shown as a function of the energy deposited in a topo-cluster $E_{\mathrm{clus}}^{\mathrm{dep}}$ for both the BNN- and the DNN-derived calibrations. The inset shows the shapes of the $\Delta_E^{\mathrm{BNN}}$ and the $\Delta_E^{\mathrm{DNN}}$ distributions in Slice (a), for $E_{\mathrm{clus}}^{\mathrm{dep}}$ around 300 MeV. The corresponding distributions for Slice (b) with $E_{\mathrm{clus}}^{\mathrm{dep}}$ around 2.5 GeV and Slice (c) with $E_{\mathrm{clus}}^{\mathrm{dep}}$ around 430 GeV are shown in (b) and (c), respectively. The shaded areas indicate the statistical uncertainties. The topo-clusters from the test dataset used to produce these results are extracted as described in Section 3.1.1.

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

# The ATLAS Collaboration

G. Aad [104], E. Aakvaag [17], B. Abbott [123], S. Abdelhameed [119a], K. Abeling [56], N.J. Abicht [50], S.H. Abidi [30], M. Aboelela [46], A. Aboulhorma [36e], H. Abramowicz [155], Y. Abulaiti [120], B.S. Acharya [70a,70b,n], A. Ackermann [64a], C. Adam Bourdarios [4], L. Adamczyk [87a], S.V. Addepalli [147], M.J. Addison [103], J. Adelman [118], A. Adiguzel [22c], T. Adye [137], A.A. Affolder [139], Y. Afik [41], M.N. Agaras [13], A. Aggarwal [102], C. Agheorghiesei [28c], F. Ahmadov [40,ad], S. Ahuja [97], X. Ai [63e], G. Aielli [77a,77b], A. Aikot [168], M. Ait Tamlihat [36e], B. Aitbenchikh [36a], M. Akbiyik [102], T.P.A. $\mathcal{R}$ Akesson [100], A.V. Akimov [149], D. Akiyama [173], N.N. Akolkar [25], S. Aktas [22a], G.L. Alberghi [24b], J. Albert [170], P. Albicocco [54], G.L. Albouy [61], S. Alderweireldt [53], Z.L. Alegria [124], M. Aleksa [37], I.N. Aleksandrov [40], C. Alexa [28b], T. Alexopoulos [10], F. Alfonsi [24b], M. Algren [57], M. Alhroob [172], B. Ali [135], H.M.J. Ali [93,w], S. Ali [32], S.W. Alibocus [94], M. Aliev [34c], G. Alimonti [72a], W. Alkakhi [56], C. Allaire [67], B.M.M. Allbrooke [150], J.S. Allen [103], J.F. Allen [53], P.P. Allport [21], A. Aloisio [73a,73b], F. Alonso [92], C. Alpigiani [142], Z.M.K. Alsolami [93], A. Alvarez Fernandez [102], M. Alves Cardoso [57], M.G. Alviggi [73a,73b], M. Aly [103], Y. Amaral Coutinho [84b], A. Ambler [106], C. Amelung [37], M. Amerl [103], C.G. Ames [111], D. Amidei [108], B. Amini [55], K.J. Amirie [159], A. Amirkhanov [40], S.P. Amor Dos Santos [133a], K.R. Amos [168], D. Amperiadou [156], S. An [85], V. Ananiev [128], C. Anastopoulos [143], T. Andeen [11], J.K. Anders [94], A.C. Anderson [60], A. Andreazza [72a,72b], S. Angelidakis [9], A. Angerami [43], A.V. Anisenkov [40], A. Annovi [75a], C. Antel [57], E. Antipov [149], M. Antonelli [54], F. Anulli [76a], M. Aoki [85], T. Aoki [157], M.A. Aparo [150], L. Aperio Bella [49], C. Appelt [155], A. Apyan [27], S.J. Arbiol Val [88], C. Arcangeletti [54], A.T.H. Arce [52], J-F. Arguin [110], S. Argyropoulos [156], J.-H. Arling [49], O. Arnaez [4], H. Arnold [149], G. Artoni [76a,76b], H. Asada [113], K. Asai [121], S. Asai [157], N.A. Asbah [37], R.A. Ashby Pickering [172], A.M. Aslam [97], K. Assamagan [30], R. Astalos [29a], K.S.V. Astrand [100], S. Atashi [163], R.J. Atkin [34a], H. Atmani[36f], P.A. Atmasiddha [131], K. Augsten [135], A.D. Auriol [42], V.A. Austrup [103], G. Avolio [37], K. Axiotis [57], G. Azuelos [110,ah], D. Babal [29b], H. Bachacou [138], K. Bachas [156,r], A. Bachiu [35], E. Bachmann [51], M.J. Backes [64a], A. Badea [41], T.M. Baer [108], P. Bagnaia [76a,76b], M. Bahmani [19], D. Bahner [55], K. Bai [126], J.T. Baines [137], L. Baines [96], O.K. Baker [177], E. Bakos [16], D. Bakshi Gupta [8], L.E. Balabram Filho [84b], V. Balakrishnan [123], R. Balasubramanian [4], E.M. Baldin [39], P. Balek [87a], E. Ballabene [24b,24a], F. Balli [138], L.M. Baltes [64a], W.K. Balunas [33], J. Balz [102], I. Bamwidhi [119b], E. Banas [88], M. Bandieramonte [132], A. Bandyopadhyay [25], S. Bansal [25], L. Barak [155], M. Barakat [49], E.L. Barberio [107], D. Barberis [58b,58a], M. Barbero [104], M.Z. Barel [117], T. Barillari [112], M-S. Barisits [37], T. Barklow [147], P. Baron [125], D.A. Baron Moreno [103], A. Baroncelli [63a], A.J. Barr [129], J.D. Barr [98], F. Barreiro [101], J. Barreiro Guimarães da Costa [14], M.G. Barros Teixeira [133a], S. Barsov [39], F. Bartels [64a], R. Bartoldus [147], A.E. Barton [93], P. Bartos [29a], A. Basan [102], M. Baselga [50], S. Bashiri[88], A. Bassalat [67,b], M.J. Basso [160a], S. Bataju [46], R. Bate [169], R.L. Bates [60], S. Batlamous[101], M. Battaglia [139], D. Battulga [19], M. Bauce [76a,76b], M. Bauer [80], P. Bauer [25], L.T. Bayer [49], L.T. Bazzano Hurrell [31], J.B. Beacham [112], T. Beau [130], J.Y. Beaucamp [92], P.H. Beauchemin [162], P. Bechtle [25], H.P. Beck [20,q], K. Becker [172], A.J. Beddall [83], V.A. Bednyakov [40], C.P. Bee [149], L.J. Beemster [16], M. Begalli [84d], M. Begel [30], J.K. Behr [49], J.F. Beirer [37], F. Beisiegel [25], M. Belfkir [119b], G. Bella [155], L. Bellagamba [24b], A. Bellerive [35], P. Bellos [21], K. Beloborodov [39], D. Benchekroun [36a], F. Bendebba [36a], Y. Benhammou [155], K.C. Benkendorfer [62], L. Beresford [49], M. Beretta [54], E. Bergeaas Kuutmann [166], N. Berger [4],

B. Bergmann [135], J. Beringer [18a], G. Bernardi [5], C. Bernius [147], F.U. Bernlochner [25],
F. Bernon [37], A. Berrocal Guardia [13], T. Berry [97], P. Berta [136], A. Berthold [51], S. Bethke [112],
A. Betti [76a,76b], A.J. Bevan [96], N.K. Bhalla [55], S. Bharthuar [112], S. Bhatta [149],
D.S. Bhattacharya [171], P. Bhattarai [147], Z.M. Bhatti [120], K.D. Bhide [55], V.S. Bhopatkar [124],
R.M. Bianchi [132], G. Bianco [24b,24a], O. Biebel [111], M. Biglietti [78a], C.S. Billingsley [46],
Y. Bimgdi [36f], M. Bindi [56], A. Bingham [176], A. Bingul [22b], C. Bini [76a,76b], G.A. Bird [33],
M. Birman [174], M. Biros [136], S. Biryukov [150], T. Bisanz [50], E. Bisceglie [45b,45a], J.P. Biswal [137],
D. Biswas [145], I. Bloch [49], A. Blue [60], U. Blumenschein [96], J. Blumenthal [102],
V.S. Bobrovnikov [40], M. Boehler [55], B. Boehm [171], D. Bogavac [37], A.G. Bogdanchikov [39],
L.S. Boggia [130], V. Boisvert [97], P. Bokan [37], T. Bold [87a], M. Bomben [5], M. Bona [96],
M. Boonekamp [138], A.G. Borbély [60], I.S. Bordulev [39], G. Borissov [93], D. Bortoletto [129],
D. Boscherini [24b], M. Bosman [13], K. Bouaouda [36a], N. Bouchhar [168], L. Boudet [4],
J. Boudreau [132], E.V. Bouhova-Thacker [93], D. Boumediene [42], R. Bouquet [58b,58a], A. Boveia [122],
J. Boyd [37], D. Boye [30], I.R. Boyko [40], L. Bozianu [57], J. Bracinik [21], N. Brahimi [4],
G. Brandt [176], O. Brandt [33], B. Brau [105], J.E. Brau [126], R. Brener [174], L. Brenner [117],
R. Brenner [166], S. Bressler [174], G. Brianti [79a,79b], D. Britton [60], D. Britzger [112], I. Brock [25],
R. Brock [109], G. Brooijmans [43], A.J. Brooks [69], E.M. Brooks [160b], E. Brost [30], L.M. Brown [170],
L.E. Bruce [62], T.L. Bruckler [129], P.A. Bruckman de Renstrom [88], B. Brüers [49], A. Bruni [24b],
G. Bruni [24b], D. Brunner [48a,48b], M. Bruschi [24b], N. Bruscino [76a,76b], T. Buanes [17], Q. Buat [142],
D. Buchin [112], A.G. Buckley [60], O. Bulekov [39], B.A. Bullard [147], S. Burdin [94], C.D. Burgard [50],
A.M. Burger [37], B. Burghgrave [8], O. Burlayenko [55], J. Burleson [167], J.T.P. Burr [33],
J.C. Burzynski [146], E.L. Busch [43], V. Büscher [102], P.J. Bussey [60], J.M. Butler [26], C.M. Buttar [60],
J.M. Butterworth [98], W. Buttinger [137], C.J. Buxo Vazquez [109], A.R. Buzykaev [40],
S. Cabrera Urbán [168], L. Cadamuro [67], D. Caforio [59], H. Cai [132], Y. Cai [24b,114c,24a], Y. Cai [114a],
V.M.M. Cairo [37], O. Cakir [3a], N. Calace [37], P. Calafiura [18a], G. Calderini [130], P. Calfayan [35],
G. Callea [60], L.P. Caloba [84b], D. Calvet [42], S. Calvet [42], R. Camacho Toro [130], S. Camarda [37],
D. Camarero Munoz [27], P. Camarri [77a,77b], M.T. Camerlingo [73a,73b], D. Cameron [37],
C. Camincher [170], M. Campanelli [98], A. Camplani [44], V. Canale [73a,73b], A.C. Canbay [3a],
E. Canonero [97], J. Cantero [168], Y. Cao [167], F. Capocasa [27], M. Capua [45b,45a], A. Carbone [72a,72b],
R. Cardarelli [77a], J.C.J. Cardenas [8], M.P. Cardiff [27], G. Carducci [45b,45a], T. Carli [37],
G. Carlino [73a], J.I. Carlotto [13], B.T. Carlson [132,s], E.M. Carlson [170], J. Carmignani [94],
L. Carminati [72a,72b], A. Carnelli [138], M. Carnesale [37], S. Caron [116], E. Carquin [140f],
I.B. Carr [107], S. Carrá [72a], G. Carratta [24b,24a], A.M. Carroll [126], M.P. Casado [13,i], M. Caspar [49],
F.L. Castillo [4], L. Castillo Garcia [13], V. Castillo Gimenez [168], N.F. Castro [133a,133e],
A. Catinaccio [37], J.R. Catmore [128], T. Cavaliere [4], V. Cavaliere [30], L.J. Caviedes Betancourt [23b],
Y.C. Cekmecelioglu [49], E. Celebi [83], S. Cella [37], V. Cepaitis [57], K. Cerny [125],
A.S. Cerqueira [84a], A. Cerri [75a,75b], L. Cerrito [77a,77b], F. Cerutti [18a], B. Cervato [145],
A. Cervelli [24b], G. Cesarini [54], S.A. Cetin [83], P.M. Chabrillat [130], J. Chan [18a], W.Y. Chan [157],
J.D. Chapman [33], E. Chapon [138], B. Chargeishvili [153b], D.G. Charlton [21], C. Chauhan [136],
Y. Che [114a], S. Chekanov [6], S.V. Chekulaev [160a], G.A. Chelkov [40,a], B. Chen [155], B. Chen [170],
H. Chen [114a], H. Chen [30], J. Chen [63c], J. Chen [146], M. Chen [129], S. Chen [89], S.J. Chen [114a],
X. Chen [63c], X. Chen [15,ag], C.L. Cheng [175], H.C. Cheng [65a], S. Cheong [147], A. Cheplakov [40],
E. Cheremushkina [49], E. Cherepanova [117], R. Cherkaoui El Moursli [36e], E. Cheu [7], K. Cheung [66],
L. Chevalier [138], V. Chiarella [54], G. Chiarelli [75a], N. Chiedde [104], G. Chiodini [71a],
A.S. Chisholm [21], A. Chitan [28b], M. Chitishvili [168], M.V. Chizhov [40,t], K. Choi [11], Y. Chou [142],
E.Y.S. Chow [116], K.L. Chu [174], M.C. Chu [65a], X. Chu [14,114c], Z. Chubinidze [54], J. Chudoba [134],
J.J. Chwastowski [88], D. Cieri [112], K.M. Ciesla [87a], V. Cindro [95], A. Ciocio [18a], F. Cirotto [73a,73b],

Z.H. Citron [174], M. Citterio [72a], D.A. Ciubotaru[28b], A. Clark [57], P.J. Clark [53], N. Clarke Hall [98],
C. Clarry [159], S.E. Clawson [49], C. Clement [48a,48b], Y. Coadou [104], M. Cobal [70a,70c],
A. Coccaro [58b], R.F. Coelho Barrue [133a], R. Coelho Lopes De Sa [105], S. Coelli [72a],
L.S. Colangeli [159], B. Cole [43], P. Collado Soto [101], J. Collot [61], P. Conde Muiño [133a,133g],
M.P. Connell [34c], S.H. Connell [34c], E.I. Conroy [129], F. Conventi [73a,ai], H.G. Cooke [21],
A.M. Cooper-Sarkar [129], F.A. Corchia [24b,24a], A. Cordeiro Oudot Choi [130], L.D. Corpe [42],
M. Corradi [76a,76b], F. Corriveau [106,ac], A. Cortes-Gonzalez [19], M.J. Costa [168], F. Costanza [4],
D. Costanzo [143], B.M. Cote [122], J. Couthures [4], G. Cowan [97], K. Cranmer [175], L. Cremer [50],
D. Cremonini [24b,24a], S. Crépé-Renaudin [61], F. Crescioli [130], M. Cristinziani [145],
M. Cristoforetti [79a,79b], V. Croft [117], J.E. Crosby [124], G. Crosetti [45b,45a], A. Cueto [101], H. Cui [98],
Z. Cui [7], W.R. Cunningham [60], F. Curcio [168], J.R. Curran [53], P. Czodrowski [37],
M.J. Da Cunha Sargedas De Sousa [58b,58a], J.V. Da Fonseca Pinto [84b], C. Da Via [103],
W. Dabrowski [87a], T. Dado [37], S. Dahbi [152], T. Dai [108], D. Dal Santo [20], C. Dallapiccola [105],
M. Dam [44], G. D'amen [30], V. D'Amico [111], J. Damp [102], J.R. Dandoy [35], D. Dannheim [37],
M. Danninger [146], V. Dao [149], G. Darbo [58b], S.J. Das [30], F. Dattola [49], S. D'Auria [72a,72b],
A. D'Avanzo [73a,73b], T. Davidek [136], I. Dawson [96], H.A. Day-hall [135], K. De [8],
C. De Almeida Rossi [159], R. De Asmundis [73a], N. De Biase [49], S. De Castro [24b,24a],
N. De Groot [116], P. de Jong [117], H. De la Torre [118], A. De Maria [114a], A. De Salvo [76a],
U. De Sanctis [77a,77b], F. De Santis [71a,71b], A. De Santo [150], J.B. De Vivie De Regie [61],
J. Debevc [95], D.V. Dedovich[40], J. Degens [94], A.M. Deiana [46], J. Del Peso [101], L. Delagrange [130],
F. Deliot [138], C.M. Delitzsch [50], M. Della Pietra [73a,73b], D. Della Volpe [57], A. Dell'Acqua [37],
L. Dell'Asta [72a,72b], M. Delmastro [4], C.C. Delogu [102], P.A. Delsart [61], S. Demers [177],
M. Demichev [40], S.P. Denisov [39], H. Denizli [22a,l], L. D'Eramo [42], D. Derendarz [88],
F. Derue [130], P. Dervan [94], K. Desch [25], C. Deutsch [25], F.A. Di Bello [58b,58a],
A. Di Ciaccio [77a,77b], L. Di Ciaccio [4], A. Di Domenico [76a,76b], C. Di Donato [73a,73b],
A. Di Girolamo [37], G. Di Gregorio [37], A. Di Luca [79a,79b], B. Di Micco [78a,78b], R. Di Nardo [78a,78b],
K.F. Di Petrillo [41], M. Diamantopoulou [35], F.A. Dias [117], T. Dias Do Vale [146],
M.A. Diaz [140a,140b], A.R. Didenko [40], M. Didenko [168], E.B. Diehl [108], S. Díez Cornell [49],
C. Diez Pardos [145], C. Dimitriadi [148], A. Dimitrievska [21], A. Dimri [149], J. Dingfelder [25],
T. Dingley [129], I-M. Dinu [28b], S.J. Dittmeier [64b], F. Dittus [37], M. Divisek [136], B. Dixit [94],
F. Djama [104], T. Djobava [153b], C. Doglioni [103,100], A. Dohnalova [29a], Z. Dolezal [136],
K. Domijan [87a], K.M. Dona [41], M. Donadelli [84d], B. Dong [109], J. Donini [42],
A. D'Onofrio [73a,73b], M. D'Onofrio [94], J. Dopke [137], A. Doria [73a], N. Dos Santos Fernandes [133a],
P. Dougan [103], M.T. Dova [92], A.T. Doyle [60], M.A. Draguet [129], M.P. Drescher [56], E. Dreyer [174],
I. Drivas-koulouris [10], M. Drnevich [120], M. Drozdova [57], D. Du [63a], T.A. du Pree [117],
F. Dubinin [39], M. Dubovsky [29a], E. Duchovni [174], G. Duckeck [111], P.K. Duckett[98], O.A. Ducu [28b],
D. Duda [53], A. Dudarev [37], E.R. Duden [27], M. D'uffizi [103], L. Duflot [67], M. Dührssen [37],
I. Duminica [28g], A.E. Dumitriu [28b], M. Dunford [64a], S. Dungs [50], K. Dunne [48a,48b],
A. Duperrin [104], H. Duran Yildiz [3a], M. Düren [59], A. Durglishvili [153b], D. Duvnjak [35],
B.L. Dwyer [118], G.I. Dyckes [18a], M. Dyndal [87a], B.S. Dziedzic [37], Z.O. Earnshaw [150],
G.H. Eberwein [129], B. Eckerova [29a], S. Eggebrecht [56], E. Egidio Purcino De Souza [84e],
G. Eigen [17], K. Einsweiler [18a], T. Ekelof [166], P.A. Ekman [100], S. El Farkh [36b], Y. El Ghazali [63a],
H. El Jarrari [37], A. El Moussaouy [36a], V. Ellajosyula [166], M. Ellert [166], F. Ellinghaus [176],
N. Ellis [37], J. Elmsheuser [30], M. Elsawy [119a], M. Elsing [37], D. Emeliyanov [137], Y. Enari [85],
I. Ene [18a], S. Epari [13], D. Ernani Martins Neto [88], M. Errenst [176], M. Escalier [67], C. Escobar [168],
E. Etzion [155], G. Evans [133a,133b], H. Evans [69], L.S. Evans [97], A. Ezhilov [39], S. Ezzarqtouni [36a],
F. Fabbri [24b,24a], L. Fabbri [24b,24a], G. Facini [98], V. Fadeyev [139], R.M. Fakhrutdinov [39],

D. Fakoudis [102], S. Falciano [76a], L.F. Falda Ulhoa Coelho [133a], F. Fallavollita [112],
G. Falsetti [45b,45a], J. Faltova [136], C. Fan [167], K.Y. Fan [65b], Y. Fan [14], Y. Fang [14,114c],
M. Fanti [72a,72b], M. Faraj [70a,70b], Z. Farazpay [99], A. Farbin [8], A. Farilla [78a], T. Farooque [109],
J.N. Farr [177], S.M. Farrington [137,53], F. Fassi [36e], D. Fassouliotis [9], L. Fayard [67], P. Federic [136],
P. Federicova [134], O.L. Fedin [39,a], M. Feickert [175], L. Feligioni [104], D.E. Fellers [18a], C. Feng [63b],
Z. Feng [117], M.J. Fenton [163], L. Ferencz [49], P. Fernandez Martinez [68], M.J.V. Fernoux [104],
J. Ferrando [93], A. Ferrari [166], P. Ferrari [117,116], R. Ferrari [74a], D. Ferrere [57], C. Ferretti [108],
M.P. Fewell [1], D. Fiacco [76a,76b], F. Fiedler [102], P. Fiedler [135], S. Filimonov [39], A. Filipčič [95],
E.K. Filmer [160a], F. Filthaut [116], M.C.N. Fiolhais [133a,133c,c], L. Fiorini [168], W.C. Fisher [109],
T. Fitschen [103], P.M. Fitzhugh [138], I. Fleck [145], P. Fleischmann [108], T. Flick [176], M. Flores [34d,ae],
L.R. Flores Castillo [65a], L. Flores Sanz De Acedo [37], F.M. Follega [79a,79b], N. Fomin [33],
J.H. Foo [159], A. Formica [138], A.C. Forti [103], E. Fortin [37], A.W. Fortman [18a], L. Fountas [9,j],
D. Fournier [67], H. Fox [93], P. Francavilla [75a,75b], S. Francescato [62], S. Franchellucci [57],
M. Franchini [24b,24a], S. Franchino [64a], D. Francis[37], L. Franco [116], V. Franco Lima [37],
L. Franconi [49], M. Franklin [62], G. Frattari [27], Y.Y. Frid [155], J. Friend [60], N. Fritzsche [37],
A. Froch [57], D. Froidevaux [37], J.A. Frost [129], Y. Fu [109], S. Fuenzalida Garrido [140f],
M. Fujimoto [104], K.Y. Fung [65a], E. Furtado De Simas Filho [84e], M. Furukawa [157], J. Fuster [168],
A. Gaa [56], A. Gabrielli [24b,24a], A. Gabrielli [159], P. Gadow [37], G. Gagliardi [58b,58a],
L.G. Gagnon [18a], S. Gaid [165], S. Galantzan [155], J. Gallagher [1], E.J. Gallas [129], A.L. Gallen [166],
B.J. Gallop [137], K.K. Gan [122], S. Ganguly [157], Y. Gao [53], A. Garabaglu [142],
F.M. Garay Walls [140a,140b], B. Garcia[30], C. García [168], A. Garcia Alonso [117],
A.G. Garcia Caffaro [177], J.E. García Navarro [168], M. Garcia-Sciveres [18a], G.L. Gardner [131],
R.W. Gardner [41], N. Garelli [162], R.B. Garg [147], J.M. Gargan [53], C.A. Garner[159], C.M. Garvey [34a],
V.K. Gassmann[162], G. Gaudio [74a], V. Gautam[13], P. Gauzzi [76a,76b], J. Gavranovic [95],
I.L. Gavrilenko [39], A. Gavrilyuk [39], C. Gay [169], G. Gaycken [126], E.N. Gazis [10], A. Gekow[122],
C. Gemme [58b], M.H. Genest [61], A.D. Gentry [115], S. George [97], W.F. George [21], T. Geralis [47],
A.A. Gerwin [123], P. Gessinger-Befurt [37], M.E. Geyik [176], M. Ghani [172], K. Ghorbanian [96],
A. Ghosal [145], A. Ghosh [163], A. Ghosh [7], B. Giacobbe [24b], S. Giagu [76a,76b], T. Giani [117],
A. Giannini [63a], S.M. Gibson [97], M. Gignac [139], D.T. Gil [87b], A.K. Gilbert [87a], B.J. Gilbert [43],
D. Gillberg [35], G. Gilles [117], L. Ginabat [130], D.M. Gingrich [2,ah], M.P. Giordani [70a,70c],
P.F. Giraud [138], G. Giugliarelli [70a,70c], D. Giugni [72a], F. Giuli [77a,77b], I. Gkialas [9,j],
L.K. Gladilin [39], C. Glasman [101], G. Glemža [49], M. Glisic[126], I. Gnesi [45b], Y. Go [30],
M. Goblirsch-Kolb [37], B. Gocke [50], D. Godin[110], B. Gokturk [22a], S. Goldfarb [107], T. Golling [57],
M.G.D. Gololo [34c], D. Golubkov [39], J.P. Gombas [109], A. Gomes [133a,133b], G. Gomes Da Silva [145],
A.J. Gomez Delegido [168], R. Gonçalo [133a], L. Gonella [21], A. Gongadze [153c], F. Gonnella [21],
J.L. Gonski [147], R.Y. González Andana [53], S. González de la Hoz [168], R. Gonzalez Lopez [94],
C. Gonzalez Renteria [18a], M.V. Gonzalez Rodrigues [49], R. Gonzalez Suarez [166],
S. Gonzalez-Sevilla [57], L. Goossens [37], B. Gorini [37], E. Gorini [71a,71b], A. Gorišek [95],
T.C. Gosart [131], A.T. Goshaw [52], M.I. Gostkin [40], S. Goswami [124], C.A. Gottardo [37],
S.A. Gotz [111], M. Gouighri [36b], A.G. Goussiou [142], N. Govender [34c], R.P. Grabarczyk [129],
I. Grabowska-Bold [87a], K. Graham [35], E. Gramstad [128], S. Grancagnolo [71a,71b], C.M. Grant [1,138],
P.M. Gravila [28f], F.G. Gravili [71a,71b], H.M. Gray [18a], M. Greco [112], M.J. Green [1], C. Grefe [25],
A.S. Grefsrud [17], I.M. Gregor [49], K.T. Greif [163], P. Grenier [147], S.G. Grewe[112], A.A. Grillo [139],
K. Grimm [32], S. Grinstein [13,x], J.-F. Grivaz [67], E. Gross [174], J. Grosse-Knetter [56], L. Guan [108],
G. Guerrieri [37], R. Gugel [102], J.A.M. Guhit [108], A. Guida [19], E. Guilloton [172], S. Guindon [37],
F. Guo [14,114c], J. Guo [63c], L. Guo [49], L. Guo [114b,v], Y. Guo [108], A. Gupta [50], R. Gupta [132],
S. Gurbuz [25], S.S. Gurdasani [49], G. Gustavino [76a,76b], P. Gutierrez [123],

L.F. Gutierrez Zagazeta [131], M. Gutsche [51], C. Gutschow [98], C. Gwenlan [129], C.B. Gwilliam [94],
E.S. Haaland [128], A. Haas [120], M. Habedank [60], C. Haber [18a], H.K. Hadavand [8], A. Haddad [42],
A. Hadef [51], A.I. Hagan [93], J.J. Hahn [145], E.H. Haines [98], M. Haleem [171], J. Haley [124],
G.D. Hallewell [104], L. Halser [20], K. Hamano [170], M. Hamer [25], E.J. Hampshire [97], J. Han [63b],
L. Han [114a], L. Han [63a], S. Han [18a], K. Hanagaki [85], M. Hance [139], D.A. Hangal [43],
H. Hanif [146], M.D. Hank [131], J.B. Hansen [44], P.H. Hansen [44], D. Harada [57], T. Harenberg [176],
S. Harkusha [178], M.L. Harris [105], Y.T. Harris [25], J. Harrison [13], N.M. Harrison [122],
P.F. Harrison[172], N.M. Hartman [112], N.M. Hartmann [111], R.Z. Hasan [97,137], Y. Hasegawa [144],
F. Haslbeck [129], S. Hassan [17], R. Hauser [109], C.M. Hawkes [21], R.J. Hawkings [37],
Y. Hayashi [157], D. Hayden [109], C. Hayes [108], R.L. Hayes [117], C.P. Hays [129], J.M. Hays [96],
H.S. Hayward [94], F. He [63a], M. He [14,114c], Y. He [49], Y. He [98], N.B. Heatley [96], V. Hedberg [100],
A.L. Heggelund [128], C. Heidegger [55], K.K. Heidegger [55], J. Heilman [35], S. Heim [49],
T. Heim [18a], T. Heimel[z], J.G. Heinlein [131], J.J. Heinrich [126], L. Heinrich [112,af], J. Hejbal [134],
A. Held [175], S. Hellesund [17], C.M. Helling [169], S. Hellman [48a,48b], L. Henkelmann [33],
A.M. Henriques Correia[37], H. Herde [100], Y. Hernández Jiménez [149], L.M. Herrmann [25],
T. Herrmann [51], G. Herten [55], R. Hertenberger [111], L. Hervas [37], M.E. Hesping [102],
N.P. Hessey [160a], J. Hessler [112], M. Hidaoui [36b], N. Hidic [136], E. Hill [159], S.J. Hillier [21],
J.R. Hinds [109], F. Hinterkeuser [25], M. Hirose [127], S. Hirose [161], D. Hirschbuehl [176],
T.G. Hitchings [103], B. Hiti [95], J. Hobbs [149], R. Hobincu [28e], N. Hod [174], M.C. Hodgkinson [143],
B.H. Hodkinson [129], A. Hoecker [37], D.D. Hofer [108], J. Hofer [168], M. Holzbock [37],
L.B.A.H. Hommels [33], B.P. Honan [103], J.J. Hong [69], J. Hong [63c], T.M. Hong [132],
B.H. Hooberman [167], W.H. Hopkins [6], M.C. Hoppesch [167], Y. Horii [113], M.E. Horstmann [112],
S. Hou [152], M.R. Housenga [167], A.S. Howard [95], J. Howarth [60], J. Hoya [6], M. Hrabovsky [125],
T. Hryn'ova [4], P.J. Hsu [66], S.-C. Hsu [142], T. Hsu [67], M. Hu [18a], Q. Hu [63a], S. Huang [33],
X. Huang [14,114c], Y. Huang [136], Y. Huang [114b], Y. Huang [102], Y. Huang [14], Z. Huang [103],
Z. Hubacek [135], M. Huebner [25], F. Huegging [25], T.B. Huffman [129],
M. Hufnagel Maranha De Faria[84a], C.A. Hugli [49], M. Huhtinen [37], S.K. Huiberts [17],
R. Hulsken [106], C.E. Hultquist [18a], N. Huseynov [12,g], J. Huston [109], J. Huth [62], R. Hyneman [7],
G. Iacobucci [57], G. Iakovidis [30], L. Iconomidou-Fayard [67], J.P. Iddon [37], P. Iengo [73a,73b],
R. Iguchi [157], Y. Iiyama [157], T. Iizawa [129], Y. Ikegami [85], D. Iliadis [156], N. Ilic [159],
H. Imam [84c], G. Inacio Goncalves [84d], S.A. Infante Cabanas [140c], T. Ingebretsen Carlson [48a,48b],
J.M. Inglis [96], G. Introzzi [74a,74b], M. Iodice [78a], V. Ippolito [76a,76b], R.K. Irwin [94], M. Ishino [157],
W. Islam [175], C. Issever [19], S. Istin [22a,am], H. Ito [173], R. Iuppa [79a,79b], A. Ivina [174], V. Izzo [73a],
P. Jacka [134], P. Jackson [1], P. Jain [49], K. Jakobs [55], T. Jakoubek [174], J. Jamieson [60],
W. Jang [157], M. Javurkova [105], P. Jawahar [103], L. Jeanty [126], J. Jejelava [153a], P. Jenni [55,f],
C.E. Jessiman [35], C. Jia [63b], H. Jia [169], J. Jia [149], X. Jia [14,114c], Z. Jia [114a], C. Jiang [53],
Q. Jiang [65b], S. Jiggins [49], J. Jimenez Pena [13], S. Jin [114a], A. Jinaru [28b], O. Jinnouchi [141],
P. Johansson [143], K.A. Johns [7], J.W. Johnson [139], F.A. Jolly [49], D.M. Jones [150], E. Jones [49],
K.S. Jones[8], P. Jones [33], R.W.L. Jones [93], T.J. Jones [94], H.L. Joos [56,37], R. Joshi [122],
J. Jovicevic [16], X. Ju [18a], J.J. Junggeburth [37], T. Junkermann [64a], A. Juste Rozas [13,x],
M.K. Juzek [88], S. Kabana [140e], A. Kaczmarska [88], M. Kado [112], H. Kagan [122], M. Kagan [147],
A. Kahn [131], C. Kahra [102], T. Kaji [157], E. Kajomovitz [154], N. Kakati [174], I. Kalaitzidou [55],
N.J. Kang [139], D. Kar [34g], K. Karava [129], E. Karentzos [25], O. Karkout [117], S.N. Karpov [40],
Z.M. Karpova [40], V. Kartvelishvili [93], A.N. Karyukhin [39], E. Kasimi [156], J. Katzy [49],
S. Kaur [35], K. Kawade [144], M.P. Kawale [123], C. Kawamoto [89], T. Kawamoto [63a], E.F. Kay [37],
F.I. Kaya [162], S. Kazakos [109], V.F. Kazanin [39], Y. Ke [149], J.M. Keaveney [34a], R. Keeler [170],
G.V. Kehris [62], J.S. Keller [35], J.J. Kempster [150], O. Kepka [134], J. Kerr [160b], B.P. Kerridge [137],

B.P. Keršnevan [95], L. Keszeghova [29a], R.A. Khan [132], A. Khanov [124], A.G. Kharlamov [39], T. Kharlamova [39], E.E. Khoda [142], M. Kholodenko [133a], T.J. Khoo [19], G. Khoriauli [171], J. Khubua [153b,*], Y.A.R. Khwaira [130], B. Kibirige [34g], D. Kim [6], D.W. Kim [48a,48b], Y.K. Kim [41], N. Kimura [98], M.K. Kingston [56], A. Kirchhoff [56], C. Kirfel [25], F. Kirfel [25], J. Kirk [137], A.E. Kiryunin [112], S. Kita [161], C. Kitsaki [10], O. Kivernyk [25], M. Klassen [162], C. Klein [35], L. Klein [171], M.H. Klein [46], S.B. Klein [57], U. Klein [94], A. Klimentov [30], T. Klioutchnikova [37], P. Kluit [117], S. Kluth [112], E. Kneringer [80], T.M. Knight [159], A. Knue [50], D. Kobylianskii [174], S.F. Koch [129], M. Kocian [147], P. Kodyš [136], D.M. Koeck [126], P.T. Koenig [25], T. Koffas [35], O. Kolay [51], I. Koletsou [4], T. Komarek [88], K. Köneke [56], A.X.Y. Kong [1], T. Kono [121], N. Konstantinidis [98], P. Kontaxakis [57], B. Konya [100], R. Kopeliansky [43], S. Koperny [87a], K. Korcyl [88], K. Kordas [156,e], A. Korn [98], S. Korn [56], I. Korolkov [13], N. Korotkova [39], B. Kortman [117], O. Kortner [112], S. Kortner [112], W.H. Kostecka [118], V.V. Kostyukhin [145], A. Kotsokechagia [37], A. Kotwal [52], A. Koulouris [37], A. Kourkoumeli-Charalampidi [74a,74b], C. Kourkoumelis [9], E. Kourlitis [112,af], O. Kovanda [126], R. Kowalewski [170], W. Kozanecki [126], A.S. Kozhin [39], V.A. Kramarenko [39], G. Kramberger [95], P. Kramer [25], M.W. Krasny [130], A. Krasznahorkay [105], A.C. Kraus [118], J.W. Kraus [176], J.A. Kremer [49], T. Kresse [51], L. Kretschmann [176], J. Kretzschmar [94], K. Kreul [19], P. Krieger [159], K. Krizka [21], K. Kroeninger [50], H. Kroha [112], J. Kroll [134], J. Kroll [131], K.S. Krowpman [109], U. Kruchonak [40], H. Krüger [25], N. Krumnack [82], M.C. Kruse [52], O. Kuchinskaia [39], S. Kuday [3a], S. Kuehn [37], R. Kuesters [55], T. Kuhl [49], V. Kukhtin [40], Y. Kulchitsky [40], S. Kuleshov [140d,140b], M. Kumar [34g], N. Kumari [49], P. Kumari [160b], A. Kupco [134], T. Kupfer [50], A. Kupich [39], O. Kuprash [55], H. Kurashige [86], L.L. Kurchaninov [160a], O. Kurdysh [4], Y.A. Kurochkin [38], A. Kurova [39], M. Kuze [141], A.K. Kvam [105], J. Kvita [125], N.G. Kyriacou [108], L.A.O. Laatu [104], C. Lacasta [168], F. Lacava [76a,76b], H. Lacker [19], D. Lacour [130], N.N. Lad [98], E. Ladygin [40], A. Lafarge [42], B. Laforge [130], T. Lagouri [177], F.Z. Lahbabi [36a], S. Lai [56], J.E. Lambert [170], S. Lammers [69], W. Lampl [7], C. Lampoudis [156,e], G. Lamprinoudis [102], A.N. Lancaster [118], E. Lançon [30], U. Landgraf [55], M.P.J. Landon [96], V.S. Lang [55], O.K.B. Langrekken [128], A.J. Lankford [163], F. Lanni [37], K. Lantzsch [25], A. Lanza [74a], M. Lanzac Berrocal [168], J.F. Laporte [138], T. Lari [72a], F. Lasagni Manghi [24b], M. Lassnig [37], V. Latonova [134], S.D. Lawlor [143], Z. Lawrence [103], R. Lazaridou [172], M. Lazzaroni [72a,72b], H.D.M. Le [109], E.M. Le Boulicaut [177], L.T. Le Pottier [18a], B. Leban [24b,24a], M. LeBlanc [103], F. Ledroit-Guillon [61], S.C. Lee [152], T.F. Lee [94], L.L. Leeuw [34c,ak], M. Lefebvre [170], C. Leggett [18a], G. Lehmann Miotto [37], M. Leigh [57], W.A. Leight [105], W. Leinonen [116], A. Leisos [156,u], M.A.L. Leite [84c], C.E. Leitgeb [19], R. Leitner [136], K.J.C. Leney [46], T. Lenz [25], S. Leone [75a], C. Leonidopoulos [53], A. Leopold [148], J.H. Lepage Bourbonnais [35], R. Les [109], C.G. Lester [33], M. Levchenko [39], J. Levêque [4], L.J. Levinson [174], G. Levrini [24b,24a], M.P. Lewicki [88], C. Lewis [142], D.J. Lewis [4], L. Lewitt [143], A. Li [30], B. Li [63b], C. Li [108], C-Q. Li [112], H. Li [63a], H. Li [63b], H. Li [103], H. Li [15], H. Li [63b], J. Li [63c], K. Li [14], L. Li [63c], R. Li [177], S. Li [14,114c], S. Li [63d,63c,d], T. Li [5], X. Li [106], Z. Li [157], Z. Li [14,114c], Z. Li [63a], S. Liang [14,114c], Z. Liang [14], M. Liberatore [138], B. Liberti [77a], K. Lie [65c], J. Lieber Marin [84e], H. Lien [69], H. Lin [108], L. Linden [111], R.E. Lindley [7], J.H. Lindon [2], J. Ling [62], E. Lipeles [131], A. Lipniacka [17], A. Lister [169], J.D. Little [69], B. Liu [14], B.X. Liu [114b], D. Liu [63d,63c], E.H.L. Liu [21], J.K.K. Liu [33], K. Liu [63d], K. Liu [63d,63c], M. Liu [63a], M.Y. Liu [63a], P. Liu [14], Q. Liu [63d,142,63c], X. Liu [63a], X. Liu [63b], Y. Liu [114b,114c], Y.L. Liu [63b], Y.W. Liu [63a], S.L. Lloyd [96], E.M. Lobodzinska [49], P. Loch [7], E. Lodhi [159], T. Lohse [19], K. Lohwasser [143], E. Loiacono [49], J.D. Lomas [21], J.D. Long [43], I. Longarini [163], R. Longo [167], A. Lopez Solis [49], N.A. Lopez-canelas [7], N. Lorenzo Martinez [4], A.M. Lory [111], M. Losada [119a], G. Löschcke Centeno [150], O. Loseva [39], X. Lou [48a,48b], X. Lou [14,114c],

A. Lounis[67], P.A. Love[93], G. Lu[14,114c], M. Lu[67], S. Lu[131], Y.J. Lu[152], H.J. Lubatti[142],
C. Luci[76a,76b], F.L. Lucio Alves[114a], F. Luehring[69], B.S. Lunday[131], O. Lundberg[148],
B. Lund-Jensen[148,*], N.A. Luongo[6], M.S. Lutz[37], A.B. Lux[26], D. Lynn[30], R. Lysak[134],
E. Lytken[100], V. Lyubushkin[40], T. Lyubushkina[40], M.M. Lyukova[149], M.Firdaus M. Soberi[53],
H. Ma[30], K. Ma[63a], L.L. Ma[63b], W. Ma[63a], Y. Ma[124], J.C. MacDonald[102],
P.C. Machado De Abreu Farias[84e], R. Madar[42], T. Madula[98], J. Maeda[86], T. Maeno[30],
P.T. Mafa[34c,k], H. Maguire[143], V. Maiboroda[138], A. Maio[133a,133b,133d], K. Maj[87a],
O. Majersky[49], S. Majewski[126], R. Makhmanazarov[39], N. Makovec[67], V. Maksimovic[16],
B. Malaescu[130], Pa. Malecki[88], V.P. Maleev[39], F. Malek[61,p], M. Mali[95], D. Malito[97],
U. Mallik[81,*], S. Maltezos[10], S. Malyukov[40], J. Mamuzic[13], G. Mancini[54], M.N. Mancini[27],
G. Manco[74a,74b], J.P. Mandalia[96], S.S. Mandarry[150], I. Mandić[95],
L. Manhaes de Andrade Filho[84a], I.M. Maniatis[174], J. Manjarres Ramos[91], D.C. Mankad[174],
A. Mann[111], S. Manzoni[37], L. Mao[63c], X. Mapekula[34c], A. Marantis[156,u], G. Marchiori[5],
M. Marcisovsky[134], C. Marcon[72a], M. Marinescu[21], S. Marium[49], M. Marjanovic[123],
A. Markhoos[55], M. Markovitch[67], M.K. Maroun[105], E.J. Marshall[93], Z. Marshall[18a],
S. Marti-Garcia[168], J. Martin[98], T.A. Martin[137], V.J. Martin[53], B. Martin dit Latour[17],
L. Martinelli[76a,76b], M. Martinez[13,x], P. Martinez Agullo[168], V.I. Martinez Outschoorn[105],
P. Martinez Suarez[13], S. Martin-Haugh[137], G. Martinovicova[136], V.S. Martoiu[28b],
A.C. Martyniuk[98], A. Marzin[37], D. Mascione[79a,79b], L. Masetti[102], J. Masik[103],
A.L. Maslennikov[40], S.L. Mason[43], P. Massarotti[73a,73b], P. Mastrandrea[75a,75b],
A. Mastroberardino[45b,45a], T. Masubuchi[127], T.T. Mathew[126], J. Matousek[136], D.M. Mattern[50],
J. Maurer[28b], T. Maurin[60], A.J. Maury[67], B. Maček[95], D.A. Maximov[39], A.E. May[103],
E. Mayer[42], R. Mazini[34g], I. Maznas[118], M. Mazza[109], S.M. Mazza[139], E. Mazzeo[72a,72b],
J.P. Mc Gowan[170], S.P. Mc Kee[108], C.A. Mc Lean[6], C.C. McCracken[169], E.F. McDonald[107],
A.E. McDougall[117], L.F. Mcelhinney[93], J.A. Mcfayden[150], R.P. McGovern[131],
R.P. Mckenzie[34g], T.C. Mclachlan[49], D.J. Mclaughlin[98], S.J. McMahon[137],
C.M. Mcpartland[94], R.A. McPherson[170,ac], S. Mehlhase[111], A. Mehta[94], D. Melini[168],
B.R. Mellado Garcia[34g], A.H. Melo[56], F. Meloni[49], A.M. Mendes Jacques Da Costa[103],
H.Y. Meng[159], L. Meng[93], S. Menke[112], M. Mentink[37], E. Meoni[45b,45a], G. Mercado[118],
S. Merianos[156], C. Merlassino[70a,70c], C. Meroni[72a,72b], J. Metcalfe[6], A.S. Mete[6],
E. Meuser[102], C. Meyer[69], J-P. Meyer[138], R.P. Middleton[137], L. Mijović[53],
G. Mikenberg[174], M. Mikestikova[134], M. Mikuž[95], H. Mildner[102], A. Milic[37],
D.W. Miller[41], E.H. Miller[147], L.S. Miller[35], A. Milov[174], D.A. Milstead[48a,48b], T. Min[114a],
A.A. Minaenko[39], I.A. Minashvili[153b], A.I. Mincer[120], B. Mindur[87a], M. Mineev[40],
Y. Mino[89], L.M. Mir[13], M. Miralles Lopez[60], M. Mironova[18a], M.C. Missio[116], A. Mitra[172],
V.A. Mitsou[168], Y. Mitsumori[113], O. Miu[159], P.S. Miyagawa[96], T. Mkrtchyan[64a],
M. Mlinarevic[98], T. Mlinarevic[98], M. Mlynarikova[37], S. Mobius[20], P. Mogg[111],
M.H. Mohamed Farook[115], A.F. Mohammed[14,114c], S. Mohapatra[43], S. Mohiuddin[124],
G. Mokgatitswane[34g], L. Moleri[174], B. Mondal[145], S. Mondal[135], K. Mönig[49],
E. Monnier[104], L. Monsonis Romero[168], J. Montejo Berlingen[13], A. Montella[48a,48b],
M. Montella[122], F. Montereali[78a,78b], F. Monticelli[92], S. Monzani[70a,70c], A. Morancho Tarda[44],
N. Morange[67], A.L. Moreira De Carvalho[49], M. Moreno Llácer[168], C. Moreno Martinez[57],
J.M. Moreno Perez[23b], P. Morettini[58b], S. Morgenstern[37], M. Morii[62], M. Morinaga[157],
M. Moritsu[90], F. Morodei[76a,76b], P. Moschovakos[37], B. Moser[129], M. Mosidze[153b],
T. Moskalets[46], P. Moskvitina[116], J. Moss[32,m], P. Moszkowicz[87a], A. Moussa[36d],
Y. Moyal[174], E.J.W. Moyse[105], O. Mtintsilana[34g], S. Muanza[104], J. Mueller[132], R. Müller[37],
G.A. Mullier[166], A.J. Mullin[33], J.J. Mullin[131], A.E. Mulski[62], D.P. Mungo[159],

D. Munoz Perez [168], F.J. Munoz Sanchez [103], M. Murin [103], W.J. Murray [172,137], M. Muškinja [95], C. Mwewa [30], A.G. Myagkov [39,a], A.J. Myers [8], G. Myers [108], M. Myska [135], B.P. Nachman [18a], K. Nagai [129], K. Nagano [85], R. Nagasaka[157], J.L. Nagle [30,aj], E. Nagy [104], A.M. Nairz [37], Y. Nakahama [85], K. Nakamura [85], K. Nakkalil [5], H. Nanjo [127], E.A. Narayanan [46], Y. Narukawa [157], I. Naryshkin [39], L. Nasella [72a,72b], S. Nasri [119b], C. Nass [25], G. Navarro [23a], J. Navarro-Gonzalez [168], A. Nayaz [19], P.Y. Nechaeva [39], S. Nechaeva [24b,24a], F. Nechansky [134], L. Nedic [129], T.J. Neep [21], A. Negri [74a,74b], M. Negrini [24b], C. Nellist [117], C. Nelson [106], K. Nelson [108], S. Nemecek [134], M. Nessi [37,h], M.S. Neubauer [167], F. Neuhaus [102], J. Newell [94], P.R. Newman [21], Y.W.Y. Ng [167], B. Ngair [119a], H.D.N. Nguyen [110], R.B. Nickerson [129], R. Nicolaidou [138], J. Nielsen [139], M. Niemeyer [56], J. Niermann [37], N. Nikiforou [37], V. Nikolaenko [39,a], I. Nikolic-Audit [130], P. Nilsson [30], I. Ninca [49], G. Ninio [155], A. Nisati [76a], N. Nishu [2], R. Nisius [112], N. Nitika [70a,70c], J-E. Nitschke [51], E.K. Nkadimeng [34g], T. Nobe [157], T. Nommensen [151], M.B. Norfolk [143], B.J. Norman [35], M. Noury [36a], J. Novak [95], T. Novak [95], R. Novotny [115], L. Nozka [125], K. Ntekas [163], N.M.J. Nunes De Moura Junior [84b], J. Ocariz [130], A. Ochi [86], I. Ochoa [133a], S. Oerdek [49,y], J.T. Offermann [41], A. Ogrodnik [136], A. Oh [103], C.C. Ohm [148], H. Oide [85], R. Oishi [157], M.L. Ojeda [37], Y. Okumura [157], L.F. Oleiro Seabra [133a], I. Oleksiyuk [57], S.A. Olivares Pino [140d], G. Oliveira Correa [13], D. Oliveira Damazio [30], J.L. Oliver [163], Ö.O. Öncel [55], A.P. O'Neill [20], A. Onofre [133a,133e], P.U.E. Onyisi [11], M.J. Oreglia [41], D. Orestano [78a,78b], R.S. Orr [159], L.M. Osojnak [131], Y. Osumi [113], G. Otero y Garzon [31], H. Otono [90], G.J. Ottino [18a], M. Ouchrif [36d], F. Ould-Saada [128], T. Ovsiannikova [142], M. Owen [60], R.E. Owen [137], V.E. Ozcan [22a], F. Ozturk [88], N. Ozturk [8], S. Ozturk [83], H.A. Pacey [129], K. Pachal [160a], A. Pacheco Pages [13], C. Padilla Aranda [13], G. Padovano [76a,76b], S. Pagan Griso [18a], G. Palacino [69], A. Palazzo [71a,71b], J. Pampel [25], J. Pan [177], T. Pan [65a], D.K. Panchal [11], C.E. Pandini [117], J.G. Panduro Vazquez [137], H.D. Pandya [1], H. Pang [138], P. Pani [49], G. Panizzo [70a,70c], L. Panwar [130], L. Paolozzi [57], S. Parajuli [167], A. Paramonov [6], C. Paraskevopoulos [54], D. Paredes Hernandez [65b], A. Pareti [74a,74b], K.R. Park [43], T.H. Park [112], F. Parodi [58b,58a], J.A. Parsons [43], U. Parzefall [55], B. Pascual Dias [42], L. Pascual Dominguez [101], E. Pasqualucci [76a], S. Passaggio [58b], F. Pastore [97], P. Patel [88], U.M. Patel [52], J.R. Pater [103], T. Pauly [37], F. Pauwels [136], C.I. Pazos [162], M. Pedersen [128], R. Pedro [133a], S.V. Peleganchuk [39], O. Penc [37], E.A. Pender [53], S. Peng [15], G.D. Penn [177], K.E. Penski [111], M. Penzin [39], B.S. Peralva [84d], A.P. Pereira Peixoto [142], L. Pereira Sanchez [147], D.V. Perepelitsa [30,aj], G. Perera [105], E. Perez Codina [160a], M. Perganti [10], H. Pernegger [37], S. Perrella [76a,76b], O. Perrin [42], K. Peters [49], R.F.Y. Peters [103], B.A. Petersen [37], T.C. Petersen [44], E. Petit [104], V. Petousis [135], C. Petridou [156,e], T. Petru [136], A. Petrukhin [145], M. Pettee [18a], A. Petukhov [83], K. Petukhova [37], R. Pezoa [140f], L. Pezzotti [24b,24a], G. Pezzullo [177], L. Pfaffenbichler [37], A.J. Pfleger [37], T.M. Pham [175], T. Pham [107], P.W. Phillips [137], G. Piacquadio [149], E. Pianori [18a], F. Piazza [126], R. Piegaia [31], D. Pietreanu [28b], A.D. Pilkington [103], M. Pinamonti [70a,70c], J.L. Pinfold [2], B.C. Pinheiro Pereira [133a], J. Pinol Bel [13], A.E. Pinto Pinoargote [138], L. Pintucci [70a,70c], K.M. Piper [150], A. Pirttikoski [57], D.A. Pizzi [35], L. Pizzimento [65b], T. Plehn[z], M.-A. Pleier [30], V. Pleskot [136], E. Plotnikova[40], G. Poddar [96], R. Poettgen [100], L. Poggioli [130], S. Polacek [136], G. Polesello [74a], A. Poley [146,160a], A. Polini [24b], C.S. Pollard [172], Z.B. Pollock [122], E. Pompa Pacchi [123], N.I. Pond [98], D. Ponomarenko [69], L. Pontecorvo [37], S. Popa [28a], G.A. Popeneciu [28d], A. Poreba [37], D.M. Portillo Quintero [160a], S. Pospisil [135], M.A. Postill [143], P. Postolache [28c], K. Potamianos [172], P.A. Potepa [87a], I.N. Potrap [40], C.J. Potter [33], H. Potti [151], J. Poveda [168], M.E. Pozo Astigarraga [37], A. Prades Ibanez [77a,77b], J. Pretel [170], D. Price [103], M. Primavera [71a], L. Primomo [70a,70c], M.A. Principe Martin [101],

R. Privara [125], T. Procter [60], M.L. Proffitt [142], N. Proklova [131], K. Prokofiev [65c], G. Proto [112],
J. Proudfoot [6], M. Przybycien [87a], W.W. Przygoda [87b], A. Psallidas [47], J.E. Puddefoot [143],
D. Pudzha [55], D. Pyatiizbyantseva [116], J. Qian [108], R. Qian [109], D. Qichen [103], Y. Qin [13],
T. Qiu [53], A. Quadt [56], M. Queitsch-Maitland [103], G. Quetant [57], R.P. Quinn [169],
G. Rabanal Bolanos [62], D. Rafanoharana [55], F. Raffaeli [77a,77b], F. Ragusa [72a,72b], J.L. Rainbolt [41],
J.A. Raine [57], S. Rajagopalan [30], E. Ramakoti [39], L. Rambelli [58b,58a], I.A. Ramirez-Berend [35],
K. Ran [49,114c], D.S. Rankin [131], N.P. Rapheeha [34g], H. Rasheed [28b], V. Raskina [130],
D.F. Rassloff [64a], A. Rastogi [18a], S. Rave [102], S. Ravera [58b,58a], B. Ravina [37], I. Ravinovich [174],
M. Raymond [37], A.L. Read [128], N.P. Readioff [143], D.M. Rebuzzi [74a,74b], A.S. Reed [112],
K. Reeves [27], J.A. Reidelsturz [176], D. Reikher [126], A. Rej [50], C. Rembser [37], H. Ren [63a],
M. Renda [28b], F. Renner [49], A.G. Rennie [163], A.L. Rescia [49], S. Resconi [72a],
M. Ressegotti [58b,58a], S. Rettie [37], W.F. Rettie [35], J.G. Reyes Rivera [109], E. Reynolds [18a],
O.L. Rezanova [40], P. Reznicek [136], H. Riani [36d], N. Ribaric [52], E. Ricci [79a,79b], R. Richter [112],
S. Richter [48a,48b], E. Richter-Was [87b], M. Ridel [130], S. Ridouani [36d], P. Rieck [120], P. Riedler [37],
E.M. Riefel [48a,48b], J.O. Rieger [117], M. Rijssenbeek [149], M. Rimoldi [37], L. Rinaldi [24b,24a],
P. Rincke [56,166], G. Ripellino [166], I. Riu [13], J.C. Rivera Vergara [170], F. Rizatdinova [124],
E. Rizvi [96], B.R. Roberts [18a], S.S. Roberts [139], D. Robinson [33], M. Robles Manzano [102],
A. Robson [60], A. Rocchi [77a,77b], C. Roda [75a,75b], S. Rodriguez Bosca [37], Y. Rodriguez Garcia [23a],
A.M. Rodríguez Vera [118], S. Roe [37], J.T. Roemer [37], O. Røhne [128], C.P.A. Roland [130], J. Roloff [30],
A. Romaniouk [80], E. Romano [74a,74b], M. Romano [24b], A.C. Romero Hernandez [167],
N. Rompotis [94], L. Roos [130], S. Rosati [76a], B.J. Rosser [41], E. Rossi [129], E. Rossi [73a,73b],
L.P. Rossi [62], L. Rossini [55], R. Rosten [122], M. Rotaru [28b], B. Rottler [55], D. Rousseau [67],
D. Rousso [49], S. Roy-Garand [159], A. Rozanov [104], Z.M.A. Rozario [60], Y. Rozen [154],
A. Rubio Jimenez [168], V.H. Ruelas Rivera [19], T.A. Ruggeri [1], A. Ruggiero [129],
A. Ruiz-Martinez [168], A. Rummler [37], Z. Rurikova [55], N.A. Rusakovich [40], H.L. Russell [170],
G. Russo [76a,76b], J.P. Rutherfoord [7], S. Rutherford Colmenares [33], M. Rybar [136], E.B. Rye [128],
A. Ryzhov [46], J.A. Sabater Iglesias [57], H.F-W. Sadrozinski [139], F. Safai Tehrani [76a], S. Saha [1],
M. Sahinsoy [83], A. Saibel [168], B.T. Saifuddin [123], M. Saimpert [138], M. Saito [157], T. Saito [157],
A. Sala [72a,72b], D. Salamani [37], A. Salnikov [147], J. Salt [168], A. Salvador Salas [155],
D. Salvatore [45b,45a], F. Salvatore [150], A. Salzburger [37], D. Sammel [55], E. Sampson [93],
D. Sampsonidis [156,e], D. Sampsonidou [126], J. Sánchez [168], V. Sanchez Sebastian [168],
H. Sandaker [128], C.O. Sander [49], J.A. Sandesara [105], M. Sandhoff [176], C. Sandoval [23b],
L. Sanfilippo [64a], D.P.C. Sankey [137], T. Sano [89], A. Sansoni [54], L. Santi [37], C. Santoni [42],
H. Santos [133a,133b], A. Santra [174], E. Sanzani [24b,24a], K.A. Saoucha [165], J.G. Saraiva [133a,133d],
J. Sardain [7], O. Sasaki [85], K. Sato [161], C. Sauer [37], E. Sauvan [4], P. Savard [159,ah], R. Sawada [157],
C. Sawyer [137], L. Sawyer [99], C. Sbarra [24b], A. Sbrizzi [24b,24a], T. Scanlon [98],
J. Schaarschmidt [142], U. Schäfer [102], A.C. Schaffer [67,46], D. Schaile [111], R.D. Schamberger [149],
C. Scharf [19], M.M. Schefer [20], V.A. Schegelsky [39], D. Scheirich [136], M. Schernau [140e],
C. Scheulen [57], C. Schiavi [58b,58a], M. Schioppa [45b,45a], B. Schlag [147], S. Schlenker [37],
J. Schmeing [176], M.A. Schmidt [176], K. Schmieden [102], C. Schmitt [102], N. Schmitt [102],
S. Schmitt [49], L. Schoeffel [138], A. Schoening [64b], P.G. Scholer [35], E. Schopf [145], M. Schott [25],
S. Schramm [57], T. Schroer [57], H-C. Schultz-Coulon [64a], M. Schumacher [55], B.A. Schumm [139],
Ph. Schune [138], H.R. Schwartz [139], A. Schwartzman [147], T.A. Schwarz [108], Ph. Schwemling [138],
R. Schwienhorst [109], F.G. Sciacca [20], A. Sciandra [30], G. Sciolla [27], F. Scuri [75a],
C.D. Sebastiani [37], K. Sedlaczek [118], S.C. Seidel [115], A. Seiden [139], B.D. Seidlitz [43],
C. Seitz [49], J.M. Seixas [84b], G. Sekhniaidze [73a], L. Selem [61], N. Semprini-Cesari [24b,24a],
A. Semushin [178,39], D. Sengupta [57], V. Senthilkumar [168], L. Serin [67], M. Sessa [77a,77b],

H. Severini [123], F. Sforza [58b,58a], A. Sfyrla [57], Q. Sha [14], E. Shabalina [56], H. Shaddix [118],
A.H. Shah [33], R. Shaheen [148], J.D. Shahinian [131], D. Shaked Renous [174], M. Shamim [37],
L.Y. Shan [14], M. Shapiro [18a], A. Sharma [37], A.S. Sharma [169], P. Sharma [30], P.B. Shatalov [39],
K. Shaw [150], S.M. Shaw [103], Q. Shen [63c], D.J. Sheppard [146], P. Sherwood [98], L. Shi [98],
X. Shi [14], S. Shimizu [85], C.O. Shimmin [177], I.P.J. Shipsey [129,*], S. Shirabe [90],
M. Shiyakova [40,aa], M.J. Shochet [41], D.R. Shope [128], B. Shrestha [123], S. Shrestha [122,al],
I. Shreyber [39], M.J. Shroff [170], P. Sicho [134], A.M. Sickles [167], E. Sideras Haddad [34g,164],
A.C. Sidley [117], A. Sidoti [24b], F. Siegert [51], Dj. Sijacki [16], F. Sili [92], J.M. Silva [53],
I. Silva Ferreira [84b], M.V. Silva Oliveira [30], S.B. Silverstein [48a], S. Simion [67], R. Simoniello [37],
E.L. Simpson [103], H. Simpson [150], L.R. Simpson [108], S. Simsek [83], S. Sindhu [56], P. Sinervo [159],
S.N. Singh [27], S. Singh [30], S. Sinha [49], S. Sinha [103], M. Sioli [24b,24a], K. Sioulas [9], I. Siral [37],
E. Sitnikova [49], J. Sjölin [48a,48b], A. Skaf [56], E. Skorda [21], P. Skubic [123], M. Slawinska [88],
I. Slazyk [17], V. Smakhtin [174], B.H. Smart [137], S.Yu. Smirnov [140b], Y. Smirnov [39],
L.N. Smirnova [39,a], O. Smirnova [100], A.C. Smith [43], D.R. Smith [163], E.A. Smith [41], J.L. Smith [103],
M.B. Smith [35], R. Smith [147], H. Smitmanns [102], M. Smizanska [93], K. Smolek [135],
A.A. Snesarev [40], H.L. Snoek [117], S. Snyder [30], R. Sobie [170,ac], A. Soffer [155],
C.A. Solans Sanchez [37], E.Yu. Soldatov [39], U. Soldevila [168], A.A. Solodkov [34g], S. Solomon [27],
A. Soloshenko [40], K. Solovieva [55], O.V. Solovyanov [42], P. Sommer [51], A. Sonay [13],
W.Y. Song [160b], A. Sopczak [135], A.L. Sopio [53], F. Sopkova [29b], J.D. Sorenson [115],
I.R. Sotarriva Alvarez [141], V. Sothilingam [64a], O.J. Soto Sandoval [140c,140b], S. Sottocornola [69],
R. Soualah [165], Z. Soumaimi [36e], D. South [49], N. Soybelman [174], S. Spagnolo [71a,71b],
M. Spalla [112], D. Sperlich [55], B. Spisso [73a,73b], D.P. Spiteri [60], M. Spousta [136], E.J. Staats [35],
R. Stamen [64a], E. Stanecka [88], W. Stanek-Maslouska [49], M.V. Stange [51], B. Stanislaus [18a],
M.M. Stanitzki [49], B. Stapf [49], E.A. Starchenko [39], G.H. Stark [139], J. Stark [91], P. Staroba [134],
P. Starovoitov [165], R. Staszewski [88], G. Stavropoulos [47], A. Stefl [37], P. Steinberg [30],
B. Stelzer [146,160a], H.J. Stelzer [132], O. Stelzer-Chilton [160a], H. Stenzel [59], T.J. Stevenson [150],
G.A. Stewart [37], J.R. Stewart [124], M.C. Stockton [37], G. Stoicea [28b], M. Stolarski [133a],
S. Stonjek [112], A. Straessner [51], J. Strandberg [148], S. Strandberg [48a,48b], M. Stratmann [176],
M. Strauss [123], T. Strebler [104], P. Strizenec [29b], R. Ströhmer [171], D.M. Strom [126],
R. Stroynowski [46], A. Strubig [48a,48b], S.A. Stucci [30], B. Stugu [17], J. Stupak [123], N.A. Styles [49],
D. Su [147], S. Su [63a], W. Su [63d], X. Su [63a], D. Suchy [29a], K. Sugizaki [131], V.V. Sulin [39],
M.J. Sullivan [94], D.M.S. Sultan [129], L. Sultanaliyeva [39], S. Sultansoy [3b], S. Sun [175], W. Sun [14],
O. Sunneborn Gudnadottir [166], N. Sur [104], M.R. Sutton [150], H. Suzuki [161], M. Svatos [134],
M. Swiatlowski [160a], T. Swirski [171], I. Sykora [29a], M. Sykora [136], T. Sykora [136], D. Ta [102],
K. Tackmann [49,y], A. Taffard [163], R. Tafirout [160a], J.S. Tafoya Vargas [67], Y. Takubo [85],
M. Talby [104], A.A. Talyshev [39], K.C. Tam [65b], N.M. Tamir [155], A. Tanaka [157], J. Tanaka [157],
R. Tanaka [67], M. Tanasini [149], Z. Tao [169], S. Tapia Araya [140f], S. Tapprogge [102],
A. Tarek Abouelfadl Mohamed [109], S. Tarem [154], K. Tariq [14], G. Tarna [28b], G.F. Tartarelli [72a],
M.J. Tartarin [91], P. Tas [136], M. Tasevsky [134], E. Tassi [45b,45a], A.C. Tate [167], G. Tateno [157],
Y. Tayalati [36e,ab], G.N. Taylor [107], W. Taylor [160b], A.S. Tegetmeier [91], P. Teixeira-Dias [97],
J.J. Teoh [159], K. Terashi [157], J. Terron [101], S. Terzo [13], M. Testa [54], R.J. Teuscher [159,ac],
A. Thaler [80], O. Theiner [57], T. Theveneaux-Pelzer [104], O. Thielmann [176], D.W. Thomas [97],
J.P. Thomas [21], E.A. Thompson [18a], P.D. Thompson [21], E. Thomson [131], R.E. Thornberry [46],
C. Tian [63a], Y. Tian [57], V. Tikhomirov [39,a], Yu.A. Tikhonov [39], S. Timoshenko [39],
D. Timoshyn [136], E.X.L. Ting [1], P. Tipton [177], A. Tishelman-Charny [30], S.H. Tlou [34g],
K. Todome [141], S. Todorova-Nova [136], S. Todt [51], L. Toffolin [70a,70c], M. Togawa [85], J. Tojo [90],
S. Tokár [29a], O. Toldaiev [69], G. Tolkachev [104], M. Tomoto [85,113], L. Tompkins [147,o],

E. Torrence [126], H. Torres [91], E. Torró Pastor [168], M. Toscani [31], C. Tosciri [41], M. Tost [11], D.R. Tovey [143], T. Trefzger [171], A. Tricoli [30], I.M. Trigger [160a], S. Trincaz-Duvoid [130], D.A. Trischuk [27], A. Tropina [40], L. Truong [34c], M. Trzebinski [88], A. Trzupek [88], F. Tsai [149], M. Tsai [108], A. Tsiamis [156], P.V. Tsiareshka [40], S. Tsigaridas [160a], A. Tsirigotis [156,u], V. Tsiskaridze [159], E.G. Tskhadadze [153a], M. Tsopoulou [156], Y. Tsujikawa [89], I.I. Tsukerman [39], V. Tsulaia [18a], S. Tsuno [85], K. Tsuri [121], D. Tsybychev [149], Y. Tu [65b], A. Tudorache [28b], V. Tudorache [28b], S. Turchikhin [58b,58a], I. Turk Cakir [3a], R. Turra [72a], T. Turtuvshin [40], P.M. Tuts [43], S. Tzamarias [156,e], E. Tzovara [102], F. Ukegawa [161], P.A. Ulloa Poblete [140c,140b], E.N. Umaka [30], G. Unal [37], A. Undrus [30], G. Unel [163], J. Urban [29b], P. Urrejola [140a], G. Usai [8], R. Ushioda [158], M. Usman [110], F. Ustuner [53], Z. Uysal [83], V. Vacek [135], B. Vachon [106], T. Vafeiadis [37], A. Vaitkus [98], C. Valderanis [111], E. Valdes Santurio [48a,48b], M. Valente [160a], S. Valentinetti [24b,24a], A. Valero [168], E. Valiente Moreno [168], A. Vallier [91], J.A. Valls Ferrer [168], D.R. Van Arneman [117], T.R. Van Daalen [142], A. Van Der Graaf [50], H.Z. Van Der Schyf [34g], P. Van Gemmeren [6], M. Van Rijnbach [37], S. Van Stroud [98], I. Van Vulpen [117], P. Vana [136], M. Vanadia [77a,77b], U.M. Vande Voorde [148], W. Vandelli [37], E.R. Vandewall [124], D. Vannicola [155], L. Vannoli [54], R. Vari [76a], E.W. Varnes [7], C. Varni [18b], D. Varouchas [67], L. Varriale [168], K.E. Varvell [151], M.E. Vasile [28b], L. Vaslin [85], A. Vasyukov [40], L.M. Vaughan [124], R. Vavricka [136], T. Vazquez Schroeder [13], J. Veatch [32], V. Vecchio [103], M.J. Veen [105], P.M. Velie [z], I. Veliscek [30], L.M. Veloce [159], F. Veloso [133a,133c], S. Veneziano [76a], A. Ventura [71a,71b], S. Ventura Gonzalez [138], A. Verbytskyi [112], M. Verducci [75a,75b], C. Vergis [96], M. Verissimo De Araujo [84b], W. Verkerke [117], J.C. Vermeulen [117], C. Vernieri [147], M. Vessella [163], M.C. Vetterli [146,ah], A. Vgenopoulos [102], N. Viaux Maira [140f], T. Vickey [143], O.E. Vickey Boeriu [143], G.H.A. Viehhauser [129], L. Vigani [64b], M. Vigl [112], M. Villa [24b,24a], M. Villaplana Perez [168], E.M. Villhauer [53], E. Vilucchi [54], M.G. Vincter [35], A. Visibile [117], C. Vittori [37], I. Vivarelli [24b,24a], E. Voevodina [112], F. Vogel [111], L. Vogel [z], J.C. Voigt [51], P. Vokac [135], Yu. Volkotrub [87b], E. Von Toerne [25], B. Vormwald [37], K. Vorobev [39], M. Vos [168], K. Voss [145], M. Vozak [37], L. Vozdecky [123], N. Vranjes [16], M. Vranjes Milosavljevic [16], M. Vreeswijk [117], N.K. Vu [63d,63c], R. Vuillermet [37], O. Vujinovic [102], I. Vukotic [41], I.K. Vyas [35], S. Wada [161], C. Wagner [147], J.M. Wagner [18a], W. Wagner [176], S. Wahdan [176], H. Wahlberg [92], C.H. Waits [123], J. Walder [137], R. Walker [111], W. Walkowiak [145], A. Wall [131], E.J. Wallin [100], T. Wamorkar [18a], A.Z. Wang [139], C. Wang [102], C. Wang [11], H. Wang [18a], J. Wang [65c], P. Wang [103], P. Wang [98], R. Wang [62], R. Wang [6], S.M. Wang [152], S. Wang [14], T. Wang [63a], W.T. Wang [81], W. Wang [14], X. Wang [167], X. Wang [63c], Y. Wang [114a], Y. Wang [63a], Z. Wang [108], Z. Wang [63d,52,63c], Z. Wang [108], C. Wanotayaroj [85], A. Warburton [106], R.J. Ward [21], A.L. Warnerbring [145], N. Warrack [60], S. Waterhouse [97], A.T. Watson [21], H. Watson [53], M.F. Watson [21], E. Watton [60], G. Watts [142], B.M. Waugh [98], J.M. Webb [55], C. Weber [30], H.A. Weber [19], M.S. Weber [20], S.M. Weber [64a], C. Wei [63a], Y. Wei [55], A.R. Weidberg [129], E.J. Weik [120], J. Weingarten [50], C. Weiser [55], C.J. Wells [49], T. Wenaus [30], B. Wendland [50], T. Wengler [37], N.S. Wenke [112], N. Wermes [25], M. Wessels [64a], A.M. Wharton [93], A.S. White [62], A. White [8], M.J. White [1], D. Whiteson [163], L. Wickremasinghe [127], W. Wiedenmann [175], M. Wielers [137], C. Wiglesworth [44], D.J. Wilbern [123], H.G. Wilkens [37], J.J.H. Wilkinson [33], D.M. Williams [43], H.H. Williams [131], S. Williams [33], S. Willocq [105], B.J. Wilson [103], D.J. Wilson [103], P.J. Windischhofer [41], F.I. Winkel [31], F. Winklmeier [126], B.T. Winter [55], M. Wittgen [147], M. Wobisch [99], T. Wojtkowski [61], Z. Wolffs [117], J. Wollrath [37], M.W. Wolter [88], H. Wolters [133a,133c], M.C. Wong [139], E.L. Woodward [43], S.D. Worm [49], B.K. Wosiek [88], K.W. Woźniak [88], S. Wozniewski [56], K. Wraight [60], C. Wu [21], M. Wu [114b], M. Wu [116], S.L. Wu [175], X. Wu [57], X. Wu [63a], Y. Wu [63a], Z. Wu [4], J. Wuerzinger [112,af],

T.R. Wyatt [103], B.M. Wynne [53], S. Xella [44], L. Xia [114a], M. Xia [15], M. Xie [63a], A. Xiong [126], J. Xiong [18a], D. Xu [14], H. Xu [63a], L. Xu [63a], R. Xu [131], T. Xu [108], Y. Xu [142], Z. Xu [53], Z. Xu [114a], B. Yabsley [151], S. Yacoob [34a], Y. Yamaguchi [85], E. Yamashita [157], H. Yamauchi [161], T. Yamazaki [18a], Y. Yamazaki [86], S. Yan [60], Z. Yan [105], H.J. Yang [63c,63d], H.T. Yang [63a], S. Yang [63a], T. Yang [65c], X. Yang [37], X. Yang [14], Y. Yang [46], Y. Yang [63a], W-M. Yao [18a], H. Ye [56], J. Ye [14], S. Ye [30], X. Ye [63a], Y. Yeh [98], I. Yeletskikh [40], B. Yeo [18b], M.R. Yexley [98], T.P. Yildirim [129], P. Yin [43], K. Yorita [173], S. Younas [28b], C.J.S. Young [37], C. Young [147], N.D. Young [126], Y. Yu [63a], J. Yuan [14,114c], M. Yuan [108], R. Yuan [63d,63c], L. Yue [98], M. Zaazoua [63a], B. Zabinski [88], I. Zahir [36a], Z.K. Zak [88], T. Zakareishvili [168], S. Zambito [57], J.A. Zamora Saa [140d,140b], J. Zang [157], D. Zanzi [55], R. Zanzottera [72a,72b], O. Zaplatilek [135], C. Zeitnitz [176], H. Zeng [14], J.C. Zeng [167], D.T. Zenger Jr [27], O. Zenin [39], T. Ženiš [29a], S. Zenz [96], S. Zerradi [36a], D. Zerwas [67], M. Zhai [14,114c], D.F. Zhang [143], J. Zhang [63b], J. Zhang [6], K. Zhang [14,114c], L. Zhang [63a], L. Zhang [114a], P. Zhang [14,114c], R. Zhang [175], S. Zhang [91], T. Zhang [157], X. Zhang [63c], Y. Zhang [142], Y. Zhang [98], Y. Zhang [63a], Y. Zhang [114a], Z. Zhang [18a], Z. Zhang [63b], Z. Zhang [67], H. Zhao [142], T. Zhao [63b], Y. Zhao [35], Z. Zhao [63a], Z. Zhao [63a], A. Zhemchugov [40], J. Zheng [114a], K. Zheng [167], X. Zheng [63a], Z. Zheng [147], D. Zhong [167], B. Zhou [108], H. Zhou [7], N. Zhou [63c], Y. Zhou [15], Y. Zhou [114a], Y. Zhou [7], C.G. Zhu [63b], J. Zhu [108], X. Zhu [63d], Y. Zhu [63c], Y. Zhu [63a], X. Zhuang [14], K. Zhukov [69], N.I. Zimine [40], J. Zinsser [64b], M. Ziolkowski [145], L. Živković [16], A. Zoccoli [24b,24a], K. Zoch [62], T.G. Zorbas [143], O. Zormpa [47], W. Zou [43], L. Zwalinski [37].

[1] Department of Physics, University of Adelaide, Adelaide; Australia.

[2] Department of Physics, University of Alberta, Edmonton AB; Canada.

[3] (a) Department of Physics, Ankara University, Ankara; (b) Division of Physics, TOBB University of Economics and Technology, Ankara; Türkiye.

[4] LAPP, Université Savoie Mont Blanc, CNRS/IN2P3, Annecy; France.

[5] APC, Université Paris Cité, CNRS/IN2P3, Paris; France.

[6] High Energy Physics Division, Argonne National Laboratory, Argonne IL; United States of America.

[7] Department of Physics, University of Arizona, Tucson AZ; United States of America.

[8] Department of Physics, University of Texas at Arlington, Arlington TX; United States of America.

[9] Physics Department, National and Kapodistrian University of Athens, Athens; Greece.

[10] Physics Department, National Technical University of Athens, Zografou; Greece.

[11] Department of Physics, University of Texas at Austin, Austin TX; United States of America.

[12] Institute of Physics, Azerbaijan Academy of Sciences, Baku; Azerbaijan.

[13] Institut de Física d'Altes Energies (IFAE), Barcelona Institute of Science and Technology, Barcelona; Spain.

[14] Institute of High Energy Physics, Chinese Academy of Sciences, Beijing; China.

[15] Physics Department, Tsinghua University, Beijing; China.

[16] Institute of Physics, University of Belgrade, Belgrade; Serbia.

[17] Department for Physics and Technology, University of Bergen, Bergen; Norway.

[18] (a) Physics Division, Lawrence Berkeley National Laboratory, Berkeley CA; (b) University of California, Berkeley CA; United States of America.

[19] Institut für Physik, Humboldt Universität zu Berlin, Berlin; Germany.

[20] Albert Einstein Center for Fundamental Physics and Laboratory for High Energy Physics, University of Bern, Bern; Switzerland.

[21] School of Physics and Astronomy, University of Birmingham, Birmingham; United Kingdom.

[22] (a) Department of Physics, Bogazici University, Istanbul; (b) Department of Physics Engineering,

Gaziantep University, Gaziantep;[c]Department of Physics, Istanbul University, Istanbul; Türkiye.

23[a]Facultad de Ciencias y Centro de Investigaciónes, Universidad Antonio Nariño, Bogotá;[b]Departamento de Física, Universidad Nacional de Colombia, Bogotá; Colombia.

24[a]Dipartimento di Fisica e Astronomia A. Righi, Università di Bologna, Bologna;[b]INFN Sezione di Bologna; Italy.

25Physikalisches Institut, Universität Bonn, Bonn; Germany.

26Department of Physics, Boston University, Boston MA; United States of America.

27Department of Physics, Brandeis University, Waltham MA; United States of America.

28[a]Transilvania University of Brasov, Brasov;[b]Horia Hulubei National Institute of Physics and Nuclear Engineering, Bucharest;[c]Department of Physics, Alexandru Ioan Cuza University of Iasi, Iasi;[d]National Institute for Research and Development of Isotopic and Molecular Technologies, Physics Department, Cluj-Napoca;[e]National University of Science and Technology Politechnica, Bucharest;[f]West University in Timisoara, Timisoara;[g]Faculty of Physics, University of Bucharest, Bucharest; Romania.

29[a]Faculty of Mathematics, Physics and Informatics, Comenius University, Bratislava;[b]Department of Subnuclear Physics, Institute of Experimental Physics of the Slovak Academy of Sciences, Kosice; Slovak Republic.

30Physics Department, Brookhaven National Laboratory, Upton NY; United States of America.

31Universidad de Buenos Aires, Facultad de Ciencias Exactas y Naturales, Departamento de Física, y CONICET, Instituto de Física de Buenos Aires (IFIBA), Buenos Aires; Argentina.

32California State University, CA; United States of America.

33Cavendish Laboratory, University of Cambridge, Cambridge; United Kingdom.

34[a]Department of Physics, University of Cape Town, Cape Town;[b]iThemba Labs, Western Cape;[c]Department of Mechanical Engineering Science, University of Johannesburg, Johannesburg;[d]National Institute of Physics, University of the Philippines Diliman (Philippines);[e]University of South Africa, Department of Physics, Pretoria;[f]University of Zululand, KwaDlangezwa;[g]School of Physics, University of the Witwatersrand, Johannesburg; South Africa.

35Department of Physics, Carleton University, Ottawa ON; Canada.

36[a]Faculté des Sciences Ain Chock, Université Hassan II de Casablanca;[b]Faculté des Sciences, Université Ibn-Tofail, Kénitra;[c]Faculté des Sciences Semlalia, Université Cadi Ayyad, LPHEA-Marrakech;[d]LPMR, Faculté des Sciences, Université Mohamed Premier, Oujda;[e]Faculté des sciences, Université Mohammed V, Rabat;[f]Institute of Applied Physics, Mohammed VI Polytechnic University, Ben Guerir; Morocco.

37CERN, Geneva; Switzerland.

38Affiliated with an institute formerly covered by a cooperation agreement with CERN.

39Affiliated with an institute covered by a cooperation agreement with CERN.

40Affiliated with an international laboratory covered by a cooperation agreement with CERN.

41Enrico Fermi Institute, University of Chicago, Chicago IL; United States of America.

42LPC, Université Clermont Auvergne, CNRS/IN2P3, Clermont-Ferrand; France.

43Nevis Laboratory, Columbia University, Irvington NY; United States of America.

44Niels Bohr Institute, University of Copenhagen, Copenhagen; Denmark.

45[a]Dipartimento di Fisica, Università della Calabria, Rende;[b]INFN Gruppo Collegato di Cosenza, Laboratori Nazionali di Frascati; Italy.

46Physics Department, Southern Methodist University, Dallas TX; United States of America.

47National Centre for Scientific Research "Demokritos", Agia Paraskevi; Greece.

48[a]Department of Physics, Stockholm University;[b]Oskar Klein Centre, Stockholm; Sweden.

49Deutsches Elektronen-Synchrotron DESY, Hamburg and Zeuthen; Germany.

50Fakultät Physik , Technische Universität Dortmund, Dortmund; Germany.

[51]Institut für Kern- und Teilchenphysik, Technische Universität Dresden, Dresden; Germany.

[52]Department of Physics, Duke University, Durham NC; United States of America.

[53]SUPA - School of Physics and Astronomy, University of Edinburgh, Edinburgh; United Kingdom.

[54]INFN e Laboratori Nazionali di Frascati, Frascati; Italy.

[55]Physikalisches Institut, Albert-Ludwigs-Universität Freiburg, Freiburg; Germany.

[56]II. Physikalisches Institut, Georg-August-Universität Göttingen, Göttingen; Germany.

[57]Département de Physique Nucléaire et Corpusculaire, Université de Genève, Genève; Switzerland.

[58][a]Dipartimento di Fisica, Università di Genova, Genova;[b]INFN Sezione di Genova; Italy.

[59]II. Physikalisches Institut, Justus-Liebig-Universität Giessen, Giessen; Germany.

[60]SUPA - School of Physics and Astronomy, University of Glasgow, Glasgow; United Kingdom.

[61]LPSC, Université Grenoble Alpes, CNRS/IN2P3, Grenoble INP, Grenoble; France.

[62]Laboratory for Particle Physics and Cosmology, Harvard University, Cambridge MA; United States of America.

[63][a]Department of Modern Physics and State Key Laboratory of Particle Detection and Electronics, University of Science and Technology of China, Hefei;[b]Institute of Frontier and Interdisciplinary Science and Key Laboratory of Particle Physics and Particle Irradiation (MOE), Shandong University, Qingdao;[c]School of Physics and Astronomy, Shanghai Jiao Tong University, Key Laboratory for Particle Astrophysics and Cosmology (MOE), SKLPPC, Shanghai;[d]Tsung-Dao Lee Institute, Shanghai;[e]School of Physics, Zhengzhou University; China.

[64][a]Kirchhoff-Institut für Physik, Ruprecht-Karls-Universität Heidelberg, Heidelberg;[b]Physikalisches Institut, Ruprecht-Karls-Universität Heidelberg, Heidelberg; Germany.

[65][a]Department of Physics, Chinese University of Hong Kong, Shatin, N.T., Hong Kong;[b]Department of Physics, University of Hong Kong, Hong Kong;[c]Department of Physics and Institute for Advanced Study, Hong Kong University of Science and Technology, Clear Water Bay, Kowloon, Hong Kong; China.

[66]Department of Physics, National Tsing Hua University, Hsinchu; Taiwan.

[67]IJCLab, Université Paris-Saclay, CNRS/IN2P3, 91405, Orsay; France.

[68]Centro Nacional de Microelectrónica (IMB-CNM-CSIC), Barcelona; Spain.

[69]Department of Physics, Indiana University, Bloomington IN; United States of America.

[70][a]INFN Gruppo Collegato di Udine, Sezione di Trieste, Udine;[b]ICTP, Trieste;[c]Dipartimento Politecnico di Ingegneria e Architettura, Università di Udine, Udine; Italy.

[71][a]INFN Sezione di Lecce;[b]Dipartimento di Matematica e Fisica, Università del Salento, Lecce; Italy.

[72][a]INFN Sezione di Milano;[b]Dipartimento di Fisica, Università di Milano, Milano; Italy.

[73][a]INFN Sezione di Napoli;[b]Dipartimento di Fisica, Università di Napoli, Napoli; Italy.

[74][a]INFN Sezione di Pavia;[b]Dipartimento di Fisica, Università di Pavia, Pavia; Italy.

[75][a]INFN Sezione di Pisa;[b]Dipartimento di Fisica E. Fermi, Università di Pisa, Pisa; Italy.

[76][a]INFN Sezione di Roma;[b]Dipartimento di Fisica, Sapienza Università di Roma, Roma; Italy.

[77][a]INFN Sezione di Roma Tor Vergata;[b]Dipartimento di Fisica, Università di Roma Tor Vergata, Roma; Italy.

[78][a]INFN Sezione di Roma Tre;[b]Dipartimento di Matematica e Fisica, Università Roma Tre, Roma; Italy.

[79][a]INFN-TIFPA;[b]Università degli Studi di Trento, Trento; Italy.

[80]Universität Innsbruck, Department of Astro and Particle Physics, Innsbruck; Austria.

[81]University of Iowa, Iowa City IA; United States of America.

[82]Department of Physics and Astronomy, Iowa State University, Ames IA; United States of America.

[83]Istinye University, Sariyer, Istanbul; Türkiye.

[84][a]Departamento de Engenharia Elétrica, Universidade Federal de Juiz de Fora (UFJF), Juiz de Fora;[b]Universidade Federal do Rio De Janeiro COPPE/EE/IF, Rio de Janeiro;[c]Instituto de Física,

Universidade de São Paulo, São Paulo;[(d)]Rio de Janeiro State University, Rio de Janeiro;[(e)]Federal University of Bahia, Bahia; Brazil.

[85]KEK, High Energy Accelerator Research Organization, Tsukuba; Japan.

[86]Graduate School of Science, Kobe University, Kobe; Japan.

[87][(a)]AGH University of Krakow, Faculty of Physics and Applied Computer Science, Krakow;[(b)]Marian Smoluchowski Institute of Physics, Jagiellonian University, Krakow; Poland.

[88]Institute of Nuclear Physics Polish Academy of Sciences, Krakow; Poland.

[89]Faculty of Science, Kyoto University, Kyoto; Japan.

[90]Research Center for Advanced Particle Physics and Department of Physics, Kyushu University, Fukuoka ; Japan.

[91]L2IT, Université de Toulouse, CNRS/IN2P3, UPS, Toulouse; France.

[92]Instituto de Física La Plata, Universidad Nacional de La Plata and CONICET, La Plata; Argentina.

[93]Physics Department, Lancaster University, Lancaster; United Kingdom.

[94]Oliver Lodge Laboratory, University of Liverpool, Liverpool; United Kingdom.

[95]Department of Experimental Particle Physics, Jožef Stefan Institute and Department of Physics, University of Ljubljana, Ljubljana; Slovenia.

[96]School of Physics and Astronomy, Queen Mary University of London, London; United Kingdom.

[97]Department of Physics, Royal Holloway University of London, Egham; United Kingdom.

[98]Department of Physics and Astronomy, University College London, London; United Kingdom.

[99]Louisiana Tech University, Ruston LA; United States of America.

[100]Fysiska institutionen, Lunds universitet, Lund; Sweden.

[101]Departamento de Física Teorica C-15 and CIAFF, Universidad Autónoma de Madrid, Madrid; Spain.

[102]Institut für Physik, Universität Mainz, Mainz; Germany.

[103]School of Physics and Astronomy, University of Manchester, Manchester; United Kingdom.

[104]CPPM, Aix-Marseille Université, CNRS/IN2P3, Marseille; France.

[105]Department of Physics, University of Massachusetts, Amherst MA; United States of America.

[106]Department of Physics, McGill University, Montreal QC; Canada.

[107]School of Physics, University of Melbourne, Victoria; Australia.

[108]Department of Physics, University of Michigan, Ann Arbor MI; United States of America.

[109]Department of Physics and Astronomy, Michigan State University, East Lansing MI; United States of America.

[110]Group of Particle Physics, University of Montreal, Montreal QC; Canada.

[111]Fakultät für Physik, Ludwig-Maximilians-Universität München, München; Germany.

[112]Max-Planck-Institut für Physik (Werner-Heisenberg-Institut), München; Germany.

[113]Graduate School of Science and Kobayashi-Maskawa Institute, Nagoya University, Nagoya; Japan.

[114][(a)]Department of Physics, Nanjing University, Nanjing;[(b)]School of Science, Shenzhen Campus of Sun Yat-sen University;[(c)]University of Chinese Academy of Science (UCAS), Beijing; China.

[115]Department of Physics and Astronomy, University of New Mexico, Albuquerque NM; United States of America.

[116]Institute for Mathematics, Astrophysics and Particle Physics, Radboud University/Nikhef, Nijmegen; Netherlands.

[117]Nikhef National Institute for Subatomic Physics and University of Amsterdam, Amsterdam; Netherlands.

[118]Department of Physics, Northern Illinois University, DeKalb IL; United States of America.

[119][(a)]New York University Abu Dhabi, Abu Dhabi;[(b)]United Arab Emirates University, Al Ain; United Arab Emirates.

[120]Department of Physics, New York University, New York NY; United States of America.

[121]Ochanomizu University, Otsuka, Bunkyo-ku, Tokyo; Japan.

[122]Ohio State University, Columbus OH; United States of America.

[123]Homer L. Dodge Department of Physics and Astronomy, University of Oklahoma, Norman OK; United States of America.

[124]Department of Physics, Oklahoma State University, Stillwater OK; United States of America.

[125]Palacký University, Joint Laboratory of Optics, Olomouc; Czech Republic.

[126]Institute for Fundamental Science, University of Oregon, Eugene, OR; United States of America.

[127]Graduate School of Science, Osaka University, Osaka; Japan.

[128]Department of Physics, University of Oslo, Oslo; Norway.

[129]Department of Physics, Oxford University, Oxford; United Kingdom.

[130]LPNHE, Sorbonne Université, Université Paris Cité, CNRS/IN2P3, Paris; France.

[131]Department of Physics, University of Pennsylvania, Philadelphia PA; United States of America.

[132]Department of Physics and Astronomy, University of Pittsburgh, Pittsburgh PA; United States of America.

[133][a]Laboratório de Instrumentação e Física Experimental de Partículas - LIP, Lisboa;[b]Departamento de Física, Faculdade de Ciências, Universidade de Lisboa, Lisboa;[c]Departamento de Física, Universidade de Coimbra, Coimbra;[d]Centro de Física Nuclear da Universidade de Lisboa, Lisboa;[e]Departamento de Física, Universidade do Minho, Braga;[f]Departamento de Física Teórica y del Cosmos, Universidad de Granada, Granada (Spain);[g]Departamento de Física, Instituto Superior Técnico, Universidade de Lisboa, Lisboa; Portugal.

[134]Institute of Physics of the Czech Academy of Sciences, Prague; Czech Republic.

[135]Czech Technical University in Prague, Prague; Czech Republic.

[136]Charles University, Faculty of Mathematics and Physics, Prague; Czech Republic.

[137]Particle Physics Department, Rutherford Appleton Laboratory, Didcot; United Kingdom.

[138]IRFU, CEA, Université Paris-Saclay, Gif-sur-Yvette; France.

[139]Santa Cruz Institute for Particle Physics, University of California Santa Cruz, Santa Cruz CA; United States of America.

[140][a]Departamento de Física, Pontificia Universidad Católica de Chile, Santiago;[b]Millennium Institute for Subatomic physics at high energy frontier (SAPHIR), Santiago;[c]Instituto de Investigación Multidisciplinario en Ciencia y Tecnología, y Departamento de Física, Universidad de La Serena;[d]Universidad Andres Bello, Department of Physics, Santiago;[e]Instituto de Alta Investigación, Universidad de Tarapacá, Arica;[f]Departamento de Física, Universidad Técnica Federico Santa María, Valparaíso; Chile.

[141]Department of Physics, Institute of Science, Tokyo; Japan.

[142]Department of Physics, University of Washington, Seattle WA; United States of America.

[143]Department of Physics and Astronomy, University of Sheffield, Sheffield; United Kingdom.

[144]Department of Physics, Shinshu University, Nagano; Japan.

[145]Department Physik, Universität Siegen, Siegen; Germany.

[146]Department of Physics, Simon Fraser University, Burnaby BC; Canada.

[147]SLAC National Accelerator Laboratory, Stanford CA; United States of America.

[148]Department of Physics, Royal Institute of Technology, Stockholm; Sweden.

[149]Departments of Physics and Astronomy, Stony Brook University, Stony Brook NY; United States of America.

[150]Department of Physics and Astronomy, University of Sussex, Brighton; United Kingdom.

[151]School of Physics, University of Sydney, Sydney; Australia.

[152]Institute of Physics, Academia Sinica, Taipei; Taiwan.

[153][a]E. Andronikashvili Institute of Physics, Iv. Javakhishvili Tbilisi State University, Tbilisi;[b]High

Energy Physics Institute, Tbilisi State University, Tbilisi; [(c)]University of Georgia, Tbilisi; Georgia.

[154]Department of Physics, Technion, Israel Institute of Technology, Haifa; Israel.

[155]Raymond and Beverly Sackler School of Physics and Astronomy, Tel Aviv University, Tel Aviv; Israel.

[156]Department of Physics, Aristotle University of Thessaloniki, Thessaloniki; Greece.

[157]International Center for Elementary Particle Physics and Department of Physics, University of Tokyo, Tokyo; Japan.

[158]Graduate School of Science and Technology, Tokyo Metropolitan University, Tokyo; Japan.

[159]Department of Physics, University of Toronto, Toronto ON; Canada.

[160][(a)]TRIUMF, Vancouver BC;[(b)]Department of Physics and Astronomy, York University, Toronto ON; Canada.

[161]Division of Physics and Tomonaga Center for the History of the Universe, Faculty of Pure and Applied Sciences, University of Tsukuba, Tsukuba; Japan.

[162]Department of Physics and Astronomy, Tufts University, Medford MA; United States of America.

[163]Department of Physics and Astronomy, University of California Irvine, Irvine CA; United States of America.

[164]University of West Attica, Athens; Greece.

[165]University of Sharjah, Sharjah; United Arab Emirates.

[166]Department of Physics and Astronomy, University of Uppsala, Uppsala; Sweden.

[167]Department of Physics, University of Illinois, Urbana IL; United States of America.

[168]Instituto de Física Corpuscular (IFIC), Centro Mixto Universidad de Valencia - CSIC, Valencia; Spain.

[169]Department of Physics, University of British Columbia, Vancouver BC; Canada.

[170]Department of Physics and Astronomy, University of Victoria, Victoria BC; Canada.

[171]Fakultät für Physik und Astronomie, Julius-Maximilians-Universität Würzburg, Würzburg; Germany.

[172]Department of Physics, University of Warwick, Coventry; United Kingdom.

[173]Waseda University, Tokyo; Japan.

[174]Department of Particle Physics and Astrophysics, Weizmann Institute of Science, Rehovot; Israel.

[175]Department of Physics, University of Wisconsin, Madison WI; United States of America.

[176]Fakultät für Mathematik und Naturwissenschaften, Fachgruppe Physik, Bergische Universität Wuppertal, Wuppertal; Germany.

[177]Department of Physics, Yale University, New Haven CT; United States of America.

[178]Yerevan Physics Institute, Yerevan; Armenia.

[a] Also Affiliated with an institute covered by a cooperation agreement with CERN.

[b] Also at An-Najah National University, Nablus; Palestine.

[c] Also at Borough of Manhattan Community College, City University of New York, New York NY; United States of America.

[d] Also at Center for High Energy Physics, Peking University; China.

[e] Also at Center for Interdisciplinary Research and Innovation (CIRI-AUTH), Thessaloniki; Greece.

[f] Also at CERN, Geneva; Switzerland.

[g] Also at CMD-AC UNEC Research Center, Azerbaijan State University of Economics (UNEC); Azerbaijan.

[h] Also at Département de Physique Nucléaire et Corpusculaire, Université de Genève, Genève; Switzerland.

[i] Also at Departament de Fisica de la Universitat Autonoma de Barcelona, Barcelona; Spain.

[j] Also at Department of Financial and Management Engineering, University of the Aegean, Chios; Greece.

[k] Also at Department of Mathematical Sciences, University of South Africa, Johannesburg; South Africa.

[l] Also at Department of Physics, Bolu Abant Izzet Baysal University, Bolu; Türkiye.

[m] Also at Department of Physics, California State University, Sacramento; United States of America.

[n] Also at Department of Physics, King's College London, London; United Kingdom.

[o] Also at Department of Physics, Stanford University, Stanford CA; United States of America.

[p] Also at Department of Physics, Stellenbosch University; South Africa.

[q] Also at Department of Physics, University of Fribourg, Fribourg; Switzerland.

[r] Also at Department of Physics, University of Thessaly; Greece.

[s] Also at Department of Physics, Westmont College, Santa Barbara; United States of America.

[t] Also at Faculty of Physics, Sofia University, 'St. Kliment Ohridski', Sofia; Bulgaria.

[u] Also at Hellenic Open University, Patras; Greece.

[v] Also at Henan University; China.

[w] Also at Imam Mohammad Ibn Saud Islamic University; Saudi Arabia.

[x] Also at Institucio Catalana de Recerca i Estudis Avancats, ICREA, Barcelona; Spain.

[y] Also at Institut für Experimentalphysik, Universität Hamburg, Hamburg; Germany.

[z] Associated at Institut für Theoretische Physik, Universität Heidelberg; Germany.

[aa] Also at Institute for Nuclear Research and Nuclear Energy (INRNE) of the Bulgarian Academy of Sciences, Sofia; Bulgaria.

[ab] Also at Institute of Applied Physics, Mohammed VI Polytechnic University, Ben Guerir; Morocco.

[ac] Also at Institute of Particle Physics (IPP); Canada.

[ad] Also at Institute of Physics, Azerbaijan Academy of Sciences, Baku; Azerbaijan.

[ae] Also at National Institute of Physics, University of the Philippines Diliman (Philippines); Philippines.

[af] Also at Technical University of Munich, Munich; Germany.

[ag] Also at The Collaborative Innovation Center of Quantum Matter (CICQM), Beijing; China.

[ah] Also at TRIUMF, Vancouver BC; Canada.

[ai] Also at Università di Napoli Parthenope, Napoli; Italy.

[aj] Also at University of Colorado Boulder, Department of Physics, Colorado; United States of America.

[ak] Also at University of the Western Cape; South Africa.

[al] Also at Washington College, Chestertown, MD; United States of America.

[am] Also at Yeditepe University, Physics Department, Istanbul; Türkiye.

[*] Deceased