# Peer review of "Precision calibration of calorimeter signals in the ATLAS experiment using an uncertainty-aware neural network"

_SciPost Physics_

## Round 1 · Referee Report · Anonymous (Referee 1) · 2025-3-7

Strengths

1- First application of BNNs to calorimeter calibration in ATLAS. 2- A key advantage over standard deep learning-based calibrations. 3- Performance comparisons with existing methods (LCW and DNN) are well-structured. 4- Logical structure and good readability.

Weaknesses

1- The work does not yet demonstrate performance on real experimental data, making systematic uncertainties related to simulation inaccuracies a concern. 2- The discussion on Bayesian inference and loss function derivation is somewhat lengthy and should be streamlined. 3- It is not clear if the trained models and inference scripts will be provided for independent verification.

Report

This paper presents an application of Bayesian Neural Networks (BNNs) for the multi-dimensional calibration of calorimeter signals in the ATLAS experiment. The proposed method provides a continuous and smooth calibration function and estimates uncertainties associated with the calibrated energies. The results are compared with standard local hadronic calibration (LCW) and a deep neural network (DNN)-based approach. The BNN-calibrated energy is shown to improve upon previous techniques, with an additional advantage of providing well-characterized uncertainties.
The study is well motivated and relevant for calorimeter-based measurements with the ATLAS experiment. The application of uncertainty-aware machine learning techniques in energy calibration represents an important forward. Its importance, however, has not been demonstrated.
The methodology provides a foundation for future applications of uncertainty-aware deep learning in detector calibrations beyond calorimetry, potentially extending to other domains such as jet reconstruction.
The paper is well structured, with minimal jargon and a clear motivation. The methodology is described in sufficient detail, although some explanations of Bayesian techniques could be made more concise. The methodology, including dataset composition, feature selection, training procedures, and evaluation metrics, is clearly documented. The paper also references an external repository for further details on implementation.
It meets the journal acceptance criteria.

Requested changes

1- The discussion on Bayesian inference and loss function derivation is somewhat lengthy and should be streamlined. 2- "They potentially represent important contributions to the nuisances characterising the overall local systematic uncertainties.". This statement deserves a justification, i.e. a test which shows that this is indeed the case.

Recommendation

Publish (meets expectations and criteria for this Journal)

---

## Round 1 · Referee Report · Anonymous (Referee 2) · 2025-3-13

Report

The ATLAS Collaboration presents a detailed study of Bayesian neural networks (BNN) for the energy calibration of clusters in the ATLAS calorimeters at the LHC. The study is novel in that it is one of the first applications of BNNs under realistic conditions at a collider experiment. It is also an application in a highly-relevant field of LHC physics, as energy clusters in calorimeters are used in almost every data analysis at the LHC. This work could hence very well open new pathways in the analysis of LHC data, as deep neural networks are frequently used for calibrations but the associated with network predictions have not gained much attention so far. It is hence plausible that the presented work contributes to improving the precision at the ATLAS experiment and possibly other high-energy physics experiments.

The presented work contains a detailed study of the achievable mean energy regression and the associated energy resolution, in comparison to uncalibrated clusters (EM scale), to an ATLAS standard calibration, and to a previously published regression with a deep neural network (DNN). In addition, the uncertainty predictions of the BNN are studied in detail and are compared to the predictions from repulsive neural networks (RE), which provide an independent uncertainty measure. The BNN is shown to provide similar (and sometimes even better) calibration improvements (over the standard technique ) as the DNN. The uncertainties of the BNN and the RE are found to be consistent and conservative, which gives confidence in their robustness.

The paper is in general well written and contains interesting results that are worth publishing in this journal. My main criticism is two-fold: 1) The discussion of uncertainties is not consistent throughout the paper. As the main purpose of this study are the uncertainties, this should be improved. The main difficulty arises from the different nomenclature of uncertainties in data science and high-energy physics, and hence the interpretation of the BNN and RE uncertainties. Please refer to my comments below. 2) The difference in performance between the BNN and the DNN is not discussed in detail. In principle, the BNN should have the same performance of the DNN, unless there are differences in these approaches beyond the BNN’s estimate of the weight variances. One difference is the use of the 3 Gaussians for the loss term in the BNN, which is also mentioned in the paper, but is not discussed in detail. Other differences could be due to the self-regularizing nature of BNNs or to differences in the networks’ input features, pre-processing, network widths, depths etc. Also here, please refer to comments below.

I recommend this paper to be published in SciPost Physics once my comments below are answered.

Uncertainties:

page 12, last paragraph of Section 3.2: “They provide measures of the ultimate accuracy achieved with the trained calibration network.” - This is a strong statement that would benefit from a more detailed discussion of what predictive uncertainties are, how they are defined and what they are intended to cover. I suggest to introduce the relevant concepts much earlier and with more clarity rather than in Section 5 (see also below).

page 18, last paragraph of Section 3.3.2: “They potentially represent important contributions to the nuisances characterising the overall local systematic uncertainties.” - Also these statement are potentially strong. It is important for the reader what is understood by “local systematic uncertainties” in this context and this would again benefit from more clarify in the discussion of what BNN uncertainties are intended to cover and what is only introduced very late in Section 5.

Figure 2: introduces statistical and systematic uncertainties without further context of whether these correspond to the HEP understanding of statistical and systematic uncertainties. Again, information from Section 5 is necessary to understand this.

Section 5: Only this section introduces the concepts of episematic and aleatoric uncertainties. As discussed in the three points above, it is necessary to discuss this earlier in the paper, so that no confusion arises for readers that were not already familiar with BNNs. It is necessary to be very clear in how epistemic and aleatoric uncertainties are linked to statistical and systematic uncertainties. I find the discussion in Section 5 not very clear. First it is indicated in Section 5.1 that epistemic uncertainties may be understood as statistical uncertainties and aleatoric uncertainties as systematic uncertainties. At the end of Section 5.1, it is mentioned that both (epistemic and aleatoric uncertainties) give rise to nuisance parameters (= systematic uncertainties in the HEP physicists’ understanding). And then, in Section 5.2, it is discussed that both, systematic and statistical uncertainties, have epistemic and aleatoric “components” without specifying further how large these components could be or whether these components are even well-defined. In addition to making the discussion clearer here, it is important to make the link to HEP physicists’ understanding of statistical and systematic uncertainties in calibrations or measurements.

Comparison of BNN to DNN:

Section 3.3.2: Do you have evidence that 3 Gaussians are appropriate for approximating the likelihood? How do you choose this number? How do the results differ if you change it?

Figures 6, 7, 9: What is the origin of the differences in performance between BNN and DNN? Wouldn’t one expect that the mean predictions of the BNN reproduce those of the DNN if they use the same inputs and are both expressive enough for this regression task? How does the depth and width of the BNN and DNN compare? How is the DNN regularized compared to the BNN (which typically does not require regularization)? Or is this coming from the Gaussian mixture model? If it is the Gaussian mixture model, you may be able to show this by reducing the likelihood to a single Gaussian in the BNN.

Other clarifications and questions:

page 11, last paragraph: I suggest to clarify in in Eq. (3) which of these inputs are “all of the topo-cluster observables employed by the classification and the hadronic calibration in the LCW sequence”. I assume that these are those from zeta_clus^EM to f_emc, but it would be helpful for the reader to have this explicitly stated, so that the reader does not feel the need to look this up in the literature referenced earlier.

page 14, discussion of R^EM_clus: I am missing a discussion in terms of physics of the peak at ~50 in Figure 1c) in the EM and mixed components. Why do the effects that reduce the true deposited energy (denominator) not also reduce the measured energy at the EM scale (numerator)? Why is there a peak in particular at 50? Is this related to selection cuts in the measured energy (numerator) and if yes: how?

Table 3: Why do you use a batch size for inference? In general, I would expect inference on the full validation/test dataset.

page 16, footnote: The footnote claims that there is no “observable effects on the accuracy of the predictions” when using other functions than Gaussians for the representation of the distribution over the network weights. The statement reads as if this was tested in the context of this work, while I would have expected this to be a general statement about BNNs. If it is the former, please give more information. If it is the latter, please give proof with a reference.

page 21: Why do you say “|Delta_E^EM|>=Delta^kappa_E “and not “-Delta_E^EM > |Delta^kappa_E|”? Delta_E^EM should be mostly negative, as the EM scale should be further away from the hadronic scale than the calibrated scales. However, the hadronic scale (as in Delta^kappa_E in the formula) can vary around the true scale and would need the magnitude sign.

page 21, footnote 11: Why does the EM scale correspond to the hadronic scale (i.e. E^dep_clus) at high energies?

page 21, last paragraph before Section 4.1.3: What is the difference between a kinematic bias and a kinematic shift?

Figure 5: Why do you cut the y-axis at ~7, while there are relevant features up to ~100? Is the second peak around 50 (Fig. 1c) reproduced by the BNN and the RE?

page 23. first paragraph: I did not understand the explanation for the “upper ridge”. What does the following sentence mean? “The upper ridge populated by topo-clusters with rising REM for decreasing femc is likely introduced by E^EM_clus at EM scale that overestimates energy deposits more and more dominated by ionisations in hadronic showers extending into the Tile calorimeter.” Can you please rephrase for clarity?

Section 4.2.2: Please clarify how the choice of the loss function is connected to the fact that the mean response is described worse than the median response? Isn’t the loss function based on means rather than medians?

page 24, last paragraph: Which input features are used to tag the resistant topo-clusters? Please provide more detail to the readers.

Figure 13: What does an uncertainty of ~10 mean? An uncertainty of 10 on R or on log(R)? (Also related to the next question.)

page 33, Eq. (20): I did not understand this formula (and hence Figure 12). If the target is log10(R), why isn’t the pull then just (log10R_prediction - log10R_target)/sigma_prediction. Why do you multiply by R^BNN_clus? Does sigma_prediction represent the variation in log10R (the training target) or in R itself? I may be missing something here that may need clarification in the text of the paper.

page 35: Do all clusters in the gap region show these large variations or only a small fraction? If it is only a small fraction, can you point to what causes those few topo-clusters to be badly regressed? (This is also related to the next question.)

page 35, last paragraph: “appropriate and traceable total uncertainty” - What is meant by “appropriate” and “traceable” and how do you come to this conclusion? Did you check that an uncertainty of 10 is indeed the correct value? (For example by a pull plot for these badly estimated topo-clusters?)

page 37, last sentence of the conclusions: I did not follow the argument why high-level variables are better than low-level variables for the retraining of a neural network. Please consider rephrasing for more clarity.

Typos and minor suggestions:

page 10, numbered list, item 3: “it not all” -> “if not all”

page 19, last paragraph: I suggest to rephrase that the median “is better defined than the statistical mean”. The choice of median/mode/arithmetic mean/… is only a choice. You argue why use the median here, which is fine, but I would refrain from a statement saying that one or another choice is “better defined”. Please consider rephrasing.

page 29: “the use of a Gaussian mixture model”

page 34, first line: I suggest to rephrase “confirms that” to “is consistent with the observation that”.

Recommendation

Ask for minor revision

  • validity: top
  • significance: high
  • originality: top
  • clarity: high
  • formatting: perfect
  • grammar: perfect

Author:  ATLAS Collaboration  on 2025-05-19  [id 5496]

(in reply to Report 2 on 2025-03-13)

SciPost referee report #1: Report #1 by Anonymous (Referee 1) on 2025-3-7 (Invited Report)

Strengths 1- First application of BNNs to calorimeter calibration in ATLAS. 2- A key advantage over standard deep learning-based calibrations. 3- Performance comparisons with existing methods (LCW and DNN) are well-structured. 4- Logical structure and good readability.

Weaknesses 1- The work does not yet demonstrate performance on real experimental data, making systematic uncertainties related to simulation inaccuracies a concern.

— This paper is intentionally limited to a proof-of-principle study, and as such is considered the foundation of the performance studies that are now underway in ATLAS for all final states in both MC simulations and data.

2- The discussion on Bayesian inference and loss function derivation is somewhat lengthy and should be streamlined.

— This paper tries to address a still somewhat diverse audience. We believe it will find interested readers both from the detector (calorimeter) and performance domain and in the machine-learning community. We tried to serve both communities in the sense that not too many external sources would have to be considered to understand the findings reported here. We understand that we may not have found the optimal level here, but we would really like to keep this as is.

3- It is not clear if the trained models and inference scripts will be provided for independent verification.

— We will follow the ATLAS on policy on open data on this issue. We are working on making all relevant software and data available for the greater public.

Report This paper presents an application of Bayesian Neural Networks (BNNs) for the multi-dimensional calibration of calorimeter signals in the ATLAS experiment. The proposed method provides a continuous and smooth calibration function and estimates uncertainties associated with the calibrated energies. The results are compared with standard local hadronic calibration (LCW) and a deep neural network (DNN)-based approach. The BNN-calibrated energy is shown to improve upon previous techniques, with an additional advantage of providing well-characterized uncertainties. The study is well motivated and relevant for calorimeter-based measurements with the ATLAS experiment. The application of uncertainty-aware machine learning techniques in energy calibration represents an important forward. Its importance, however, has not been demonstrated. The methodology provides a foundation for future applications of uncertainty-aware deep learning in detector calibrations beyond calorimetry, potentially extending to other domains such as jet reconstruction. The paper is well structured, with minimal jargon and a clear motivation. The methodology is described in sufficient detail, although some explanations of Bayesian techniques could be made more concise. The methodology, including dataset composition, feature selection, training procedures, and evaluation metrics, is clearly documented. The paper also references an external repository for further details on implementation. It meets the journal acceptance criteria.

Requested changes 1- The discussion on Bayesian inference and loss function derivation is somewhat lengthy and should be streamlined.

— See initial response above. In addition what is said there, the present text is also the result of an in-depth editorial process involving both detector and ML experts..

2- "They potentially represent important contributions to the nuisances characterising the overall local systematic uncertainties.". This statement deserves a justification, i.e. a test which shows that this is indeed the case.

— We believe this is a very general statement. Whether or not this is the case depends on how the calibrated topo-clusters enter the final state reconstruction. If they contribute to a jet, the cluster uncertainties may not enter at all because the jet itself has a better constraint calibration that is not fed back to its constituent. Yet, if the topo-cluster is part of the (soft) hadronic recoil, outside of a jet or reconstructed particle context, its calibration uncertainty may well be a component contributing to the nuisances of the recoil reconstruction. We change the text here to soften the statement and add the other important use case (topo-cluster selection based on calibration accuracy prior to particle, jet and recoil reconstruction).

— Thank you very much for your helpful review. We hope you find our responses satisfactory

Recommendation Publish (meets expectations and criteria for this Journal)

validity: good significance: good originality: good clarity: high formatting: good grammar: excellent

SciPost referee report #2:

---

## Round 1 · Referee Report · Anonymous (Referee 1) · 2025-3-13

The presented work contains a detailed study of the achievable mean energy regression and the associated energy resolution, in comparison to uncalibrated clusters (EM scale), to an ATLAS standard calibration, and to a previously published regression with a deep neural network (DNN). In addition, the uncertainty predictions of the BNN are studied in detail and are compared to the predictions from repulsive neural networks (RE), which provide an independent uncertainty measure. The BNN is shown to provide similar (and sometimes even better) calibration improvements (over the standard technique ) as the DNN. The uncertainties of the BNN and the RE are found to be consistent and conservative, which gives confidence in their robustness.

The paper is in general well written and contains interesting results that are worth publishing in this journal. My main criticism is two-fold: 1) The discussion of uncertainties is not consistent throughout the paper. As the main purpose of this study are the uncertainties, this should be improved. The main difficulty arises from the different nomenclature of uncertainties in data science and high-energy physics, and hence the interpretation of the BNN and RE uncertainties. Please refer to my comments below. 2) The difference in performance between the BNN and the DNN is not discussed in detail. In principle, the BNN should have the same performance of the DNN, unless there are differences in these approaches beyond the BNN’s estimate of the weight variances. One difference is the use of the 3 Gaussians for the loss term in the BNN, which is also mentioned in the paper, but is not discussed in detail. Other differences could be due to the self-regularizing nature of BNNs or to differences in the networks’ input features, pre-processing, network widths, depths etc. Also here, please refer to comments below.

I recommend this paper to be published in SciPost Physics once my comments below are answered.

Uncertainties:

page 12, last paragraph of Section 3.2: “They provide measures of the ultimate accuracy achieved with the trained calibration network.” - This is a strong statement that would benefit from a more detailed discussion of what predictive uncertainties are, how they are defined and what they are intended to cover. I suggest to introduce the relevant concepts much earlier and with more clarity rather than in Section 5 (see also below).

— We like to keep the detailed discussion of the uncertainties in Section 5.1, close to the presentation of the corresponding findings. We added a reference to this section, and rephrased the sentence a bit.

page 18, last paragraph of Section 3.3.2: “They potentially represent important contributions to the nuisances characterising the overall local systematic uncertainties.” - Also these statement are potentially strong. It is important for the reader what is understood by “local systematic uncertainties” in this context and this would again benefit from more clarity in the discussion of what BNN uncertainties are intended to cover and what is only introduced very late in Section 5.

— In the spirit of our response to the previous comment, we like to keep the detailed uncertainty discussion in Section 5. In our opinion the statement about the contribution of nuisances to total uncertainties is rather general, but we rephrased to include the aspect that this contribution depends on the calibration model and context, in particular when applying the MC-derived calibration to experimental data.

Figure 2: introduces statistical and systematic uncertainties without further context of whether these correspond to the HEP understanding of statistical and systematic uncertainties. Again, information from Section 5 is necessary to understand this.

— We added a reference to the discussion in Section 5 to the captions of Figure 2 (and Figure 3). We believe that having the detailed discussion close to the presentation of the corresponding findings is an appropriate way to present those.

Section 5: Only this section introduces the concepts of episematic and aleatoric uncertainties. As discussed in the three points above, it is necessary to discuss this earlier in the paper, so that no confusion arises for readers that were not already familiar with BNNs. It is necessary to be very clear in how epistemic and aleatoric uncertainties are linked to statistical and systematic uncertainties. I find the discussion in Section 5 not very clear. First it is indicated in Section 5.1 that epistemic uncertainties may be understood as statistical uncertainties and aleatoric uncertainties as systematic uncertainties. At the end of Section 5.1, it is mentioned that both (epistemic and aleatoric uncertainties) give rise to nuisance parameters (= systematic uncertainties in the HEP physicists’ understanding). And then, in Section 5.2, it is discussed that both, systematic and statistical uncertainties, have epistemic and aleatoric “components” without specifying further how large these components could be or whether these components are even well-defined. In addition to making the discussion clearer here, it is important to make the link to HEP physicists’ understanding of statistical and systematic uncertainties in calibrations or measurements.

— The principal definition of systematic and statistical uncertainties in the context of the study presented in this paper is already given in the introduction. We added two more sentences there to make this more obvious, including a direct reference to “standard” sample-based calibration methods like the one for jets in ATLAS. As the concepts of aleatoric and epistemic uncertainties are highly descriptive and well understood in terms of categorization in the ML community, they are not too relevant for the application of this calibration in the HEP context. We tried to link these ML uncertainties with the experimental uncertainties in a qualitative manner in Section 5. For this, we added a small table making the relation between these two categories more clear, and changed the text accordingly. As said before, we like to keep the discussion in Section 5, though.

Comparison of BNN to DNN:

Section 3.3.2: Do you have evidence that 3 Gaussians are appropriate for approximating the likelihood? How do you choose this number? How do the results differ if you change it?

— The number of Gaussians used for the studies in the paper (3) is the result of studies with 1 to 9[a][b] Gaussians. We found that N_mix >= 3 yields the same results on the metrics we use. We feel this does not need more discussions, we will just add one sentence.

Figures 6, 7, 9: What is the origin of the differences in performance between BNN and DNN? Wouldn’t one expect that the mean predictions of the BNN reproduce those of the DNN if they use the same inputs and are both expressive enough for this regression task? How does the depth and width of the BNN and DNN compare? How is the DNN regularized compared to the BNN (which typically does not require regularization)? Or is this coming from the Gaussian mixture model? If it is the Gaussian mixture model, you may be able to show this by reducing the likelihood to a single Gaussian in the BNN.

— The DNN is discussed in detail in the referred ATLAS PubNote. It is regularized using a leaky Gaussian kernel (see https://cds.cern.ch/record/2866591/files/ATL-PHYS-PUB-2023-019.pdf for the details) in a first round of training, which is followed by a second round without regularization that starts from the weights fitted in the first. The focus of this paper are indeed the predictive uncertainties, not a detailed comparison with previous ML-based calibration attempts. We understand and share the curiosity about the reason for the differences, but it is also non-trivial to directly project these back to network (model) designs, loss functions and hyper-parameter settings. We believe that main reasons for the only nearly always improvements when using the BNN are in the (1) loss function definition accommodating asymmetric shapes of probability density, and (2) the numerically less restrictive choice of activation functions supporting all predictions >=0 while in case of the DNN the (last layer) activation function only constructs predictions in ]0,4].

Other clarifications and questions:

page 11, last paragraph: I suggest to clarify in in Eq. (3) which of these inputs are “all of the topo-cluster observables employed by the classification and the hadronic calibration in the LCW sequence”. I assume that these are those from zeta_clus^EM to f_emc, but it would be helpful for the reader to have this explicitly stated, so that the reader does not feel the need to look this up in the literature referenced earlier.

— We added a column to Table 1 indicating the features used by LCW.

page 14, discussion of R^EM_clus: I am missing a discussion in terms of physics of the peak at ~50 in Figure 1c) in the EM and mixed components. Why do the effects that reduce the true deposited energy (denominator) not also reduce the measured energy at the EM scale (numerator)? Why is there a peak in particular at 50? Is this related to selection cuts in the measured energy (numerator) and if yes: how?

— The very large responses are discussed in the text, but we added a bit more explanation. In general, the large responses can be associated with topo-clusters located in the Tile gap region (those are the same with the large uncertainties). This region is not instrumented by a calorimeter, rather it has scintillating counters that are supposed to generate signals proportional to the energy losses in the material around them. The energy deposited in this counter (which is looked at as a calorimeter cell without an absorber) is relatively small. The EM-scale calibration that enters R^EM_clus (the target) was set to correspond to some expectation for an average energy loss in the surrounding materials, and thus the numerator reflects a much larger true (deposited) energy than is given in the denominator. The peak (on the log-R scale) arises from the non-sampling character and the rather linear relation between the TileGap signal and the energy deposited in it (both are near directly proportional, the signal is just about ~50 times too big at EM scale), folded with the relatively small fluctuations in the ionisation energy loss . Similar arguments yield for topo-clusters in the pre-samplers in the LAr calorimeters. In addition, in-time pile-up signal contributions add to the signal but not to the truth (denominator). We changed the text along these lines.

Table 3: Why do you use a batch size for inference? In general, I would expect inference on the full validation/test dataset.

— This is introduced by computational limitations, in particular the lack of residual memory. We infer on the full test and validation dataset, just do it in batches.

page 16, footnote: The footnote claims that there is no “observable effects on the accuracy of the predictions” when using other functions than Gaussians for the representation of the distribution over the network weights. The statement reads as if this was tested in the context of this work, while I would have expected this to be a general statement about BNNs. If it is the former, please give more information. If it is the latter, please give proof with a reference.

— We added the reference.

page 21: Why do you say “|Delta_E^EM|>=Delta^kappa_E “and not “-Delta_E^EM > |Delta^kappa_E|”? Delta_E^EM should be mostly negative, as the EM scale should be further away from the hadronic scale than the calibrated scales. However, the hadronic scale (as in Delta^kappa_E in the formula) can vary around the true scale and would need the magnitude sign.

— Thanks for finding this oversight. We removed the equation and changed the text a bit to highlight that E^EM_clus is not a calibrated scale. In our definition in eq 16, the signal on EM scale is naively expected to be too small, but as Figures 7-8 show, due to pile-up this is not the case for the full phasespace.

page 21, footnote 11: Why does the EM scale correspond to the hadronic scale (i.e. E^dep_clus) at high energies?

— The fraction of energy deposited by photons increases (slightly) for high topo-cluster energies (Figures 1(a),(b)), and the intrinsic EM component in hadronic showers increases with energy as well, e/pi gets closer to unity (how close depends on the models). Nevertheless, with the change of text due to the previous comment we removed this footnote.

page 21, last paragraph before Section 4.1.3: What is the difference between a kinematic bias and a kinematic shift?

— We removed the shifts.

Figure 5: Why do you cut the y-axis at ~7, while there are relevant features up to ~100? Is the second peak around 50 (Fig. 1c) reproduced by the BNN and the RE?

— These qualitative comparisons focus on the complex shape in the highly populated areas. Considering the log(z) scale, we decided to cut on the y-axis as seen. The direct correlation between the prediction and the target are available in the auxiliary material and cover the full response range. Both BNN and RE actually learn the high responses, albeit with a large uncertainty. Please check https://atlas.web.cern.ch/Atlas/GROUPS/PHYSICS/PAPERS/JETM-2024-01/ , Figure 03a in the auxiliary material section.

page 23. first paragraph: I did not understand the explanation for the “upper ridge”. What does the following sentence mean? “The upper ridge populated by topo-clusters with rising REM for decreasing femc is likely introduced by E^EM_clus at EM scale that overestimates energy deposits more and more dominated by ionisations in hadronic showers extending into the Tile calorimeter.” Can you please rephrase for clarity?

— It largely is an artifact from e/mu < 1 in the Tile calorimeter. A descreasing f_emc indicates topo-clusters more and more exclusively in the hadronic Tile calorimeters, which is sensitive enough to have energy deposits from ionization losses by charged hadrons (behave like muons) generating topo-clusters. The fact that these charged tracks leave a larger signal than electrons depositing the same energy is clearly visible in this upper ridge (response > 1). We rephrased this paragraph to emphasize this fact.

Section 4.2.2: Please clarify how the choice of the loss function is connected to the fact that the mean response is described worse than the median response? Isn’t the loss function based on means rather than medians?

— For stability and algorithmic simplicity the median of our typical response distributions (in bins of any evaluation scale) is used for performance evaluations. The prediction is actually the mode of the Gaussian mixture PDF (one learned for each topo-cluster). For our case, the mean may show too much dependence on statistics due to asymmetric PDFs, to provide a stable calibration target.

page 24, last paragraph: Which input features are used to tag the resistant topo-clusters? Please provide more detail to the readers.

— This information is available in Table 01 of the auxiliary material at https://atlas.web.cern.ch/Atlas/GROUPS/PHYSICS/PAPERS/JETM-2024-01/ There are also more results on these topo-clusters.

Figure 13: What does an uncertainty of ~10 mean? An uncertainty of 10 on R or on log(R)? (Also related to the next question.)

— These are absolute uncertainties on R, to illustrate the effect better. The large uncertainties in the Tile Gap around 10 mean relative uncertainties of about 20% (10/50). If axis labels do not indicate otherwise, all predictions are in direct (non-transformed) observable space.

page 33, Eq. (20): I did not understand this formula (and hence Figure 12). If the target is log10(R), why isn’t the pull then just (log10R_prediction - log10R_target)/sigma_prediction. Why do you multiply by R^BNN_clus? Does sigma_prediction represent the variation in log10R (the training target) or in R itself? I may be missing something here that may need clarification in the text of the paper.

— The pull in log10(R) space is indeed given as you said, with sigma_prediction determined at this transformed scale as well. The original formula relates the log10(R) space response and uncertainty predictions to the corresponding ones in linear space, which is useful for calibration. We rephrased this and modified Equation 20 to highlight the relation between the linear and log pull.

page 35: Do all clusters in the gap region show these large variations or only a small fraction? If it is only a small fraction, can you point to what causes those few topo-clusters to be badly regressed? (This is also related to the next question.)

— We can only conclude (for this study) that with very few exceptions all topo-clusters in the selected uncertainty range can be found in the Tile gap region (see Figure 14(b)). We believe the uncertainty is considerable because some features lack expressiveness in this region. The fact that the center of mass of the affected topo-clusters are inside or very close to the thin scintillators indicate that cells in the active calorimeter volumes in front and behind it do not contribute much to the topo-cluster energy and thus to e.g., longitudinal (along the particle direction of flight) energy distributions and related features. Also, the Tile gap scintillators are large paddles with little granularity in rapidity (3 sections) and coarse in azimuth (64/32 panels around). Features of showers starting around them, or shower particles traversing them, are not too relevant for the signal in them.

page 35, last paragraph: “appropriate and traceable total uncertainty” - What is meant by “appropriate” and “traceable” and how do you come to this conclusion? Did you check that an uncertainty of 10 is indeed the correct value? (For example by a pull plot for these badly estimated topo-clusters?)

— Appropriate means large, while traceable means “traceable source/understandable reason”. We rephrased to make this more clear and removed both “appropriate” and “traceable”.

page 37, last sentence of the conclusions: I did not follow the argument why high-level variables are better than low-level variables for the retraining of a neural network. Please consider rephrasing for more clarity.

— This remake reflects on a practicality aspect. We are aware that in ML unconditioned features are preferred, and potential perform even better than what we can show with our choices. Yet, in case of changing beam conditions or other effects relevant for the calibration, a (relatively) quick training and inference is of advantage. Any “raw data” (cell-base) calibration would require a full reprocessing of data and MC campaigns, which usually can only happen once or twice per run year. The model here can be applied as often as needed during the run year.

Typos and minor suggestions:

page 10, numbered list, item 3: “it not all” -> “if not all”

— Thank you, fixed!

page 19, last paragraph: I suggest to rephrase that the median “is better defined than the statistical mean”. The choice of median/mode/arithmetic mean/… is only a choice. You argue why use the median here, which is fine, but I would refrain from a statement saying that one or another choice is “better defined”. Please consider rephrasing.

— We changed the text and removed “better defined”.

page 29: “the use of a Gaussian mixture model”

— Thanks, fixed.

page 34, first line: I suggest to rephrase “confirms that” to “is consistent with the observation that”.

— We rephrased along these lines.

— Thank you very much for your helpful review. We hope you find our responses satisfactory.

Recommendation Ask for minor revision

validity: top significance: high originality: top clarity: high formatting: perfect grammar: perfect

[a]we used up to 9 (not 6) Gaussians [b]Danke!

---

## Editorial Decision

resubmitted